# Reverse-Engineering Model Editing on Language Models

Zhiyu Sun [* 1 2]   Minrui Luo [* 3 1]   Yu Wang [5 1]   Zhili Chen [2]   Tianxing He [3 1 4 †]

## Abstract

Large language models (LLMs) are pretrained on corpora containing trillions of tokens and, therefore, inevitably memorize sensitive information. Locate-then-edit methods, as a mainstream paradigm of model editing, offer a promising solution by modifying model parameters without retraining. However, in this work, we reveal a critical vulnerability of this paradigm: the parameter updates inadvertently serve as a side channel, enabling attackers to recover the edited data. We propose a two-stage reverse-engineering attack named *KSTER* (**K**ey**S**pace**R**econs**T**ruction-then-**E**ntropy**R**eduction) that leverages the low-rank structure of these updates. First, we theoretically show that the row space of the update matrix encodes a "fingerprint" of the edited subjects, enabling accurate subject recovery via spectral analysis. Second, we introduce an entropy-based prompt recovery attack that reconstructs the semantic context of the edit. Extensive experiments on multiple LLMs demonstrate that our attacks can recover edited data with high success rates. Furthermore, we propose *subspace camouflage*, a defense strategy that obfuscates the update fingerprint with semantic decoys. This approach effectively mitigates reconstruction risks without compromising editing utility. Our code is available at https://github.com/reanatom/EditingAttack.

---

[*]Equal contribution [1]Shanghai Qi Zhi Institute [2]Software Engineering Institute, East China Normal University, Shanghai, China [3]Institute for Interdisciplinary Information Sciences, Tsinghua University [4]Xiongan AI Institute [5]Institute of Information Engineering, Chinese Academy of Sciences, Beijing, China. [†]Corresponding author: Tianxing He <hetianxing@mail.tsinghua.edu.cn>.

*Proceedings of the 43rd International Conference on Machine Learning*, Seoul, South Korea. PMLR 306, 2026. Copyright 2026 by the author(s).

## 1. Introduction

Large language models (LLMs) are pretrained on corpora containing trillions of tokens (Allen-Zhu & Li, 2024; Wang et al., 2024a; Kong et al., 2025), which poses a risk of memorizing sensitive information (Nasr et al., 2023; Cheng et al., 2025), such as personal data. Model editing (Wang et al., 2024c) has emerged as a promising solution to mitigate this issue by modifying specific parameters without retraining (Meng et al., 2022; Mitchell et al., 2022a). Among various editing approaches, the locate-then-edit paradigm (Meng et al., 2022; 2023; Fang et al., 2025) has drawn wide research interest. It first identifies the parameters that store specific knowledge, then edits them. This paradigm offers strong interpretability and zero inference overhead, making it a leading candidate for cost-effective privacy protection (Hossain & Kagal, 2025; Li et al., 2025).

While prior research has primarily focused on editing effectiveness and generalization, a critical safety question remains underexplored: *Does the editing process itself create a leakage side-channel*? Consider a scenario in which malicious attackers gain access to the model parameters before and after editing. Although the sensitive information resides in the pre-edit model, locating it within billions of parameters is computationally prohibitive. However, attackers can efficiently pinpoint and extract the knowledge that is erased by analyzing the parameter differences induced by editing, paradoxically turning the safety mechanism into a vulnerability.

In this work, we address this question and reveal a latent vulnerability in the locate-then-edit paradigm, demonstrating that the act of editing can serve as a side channel, exposing the very information it aims to edit. Specifically, through analyzing the algebraic structure of widely used editing algorithms, we theoretically prove that the row space of the parameter difference matrix encodes a unique mathematical "fingerprint" of the edited subjects. With this insight, we propose a two-stage reverse-engineering attack framework. First, we employ spectral analysis on the update matrix to execute subject inference, accurately pinpointing the edited subject (e.g., `Alice`). Second, we introduce an entropy-based prompt recovery method to reconstruct the semantic context associated with the edit (e.g., `{}'s phone number is {}`).

We conduct comprehensive experiments to evaluate the performance of our framework across three models and three editing methods. The results demonstrate the effectiveness of our approach, showing that we can accurately recover both the subject and the prompt. For instance, our attack method achieves a subject recall rate of over 99% and a semantic similarity of 88% when applied to Llama3-8B-Instruct (Dubey et al., 2024) on the CounterFact (Meng et al., 2022) dataset.

Moreover, to mitigate the edit-leakage risk, we propose a defense strategy named *subspace camouflage*. By injecting semantic decoys during the update process, this method obfuscates the spectral fingerprint of the target subjects, thereby effectively misleading algebraic inversion without compromising editing efficacy.

Our contributions are summarized as follows:

- We propose a reverse-engineering attack *KSTER* (**K**ey**S**paceRecons**T**ruction-then-**E**ntropy**R**eduction) against model editing, which is a two-stage framework that inverts parameter updates to recover both subjects and associated semantic prompts. We provide a theoretical analysis of the weight update in subject attack, exhibiting the reconstruction of the linear subspace spanned by the key vectors of edited knowledge.

- We introduce subspace camouflage, a defense strategy that injects semantic decoys into the update subspace to obscure algebraic fingerprints while preserving editing utility. We provide a theoretical result for the robustness of this defense strategy against a broad class of white-box attacks.

- Experiments across multiple LLMs show that our attacks recover edited data with high success rates, while our defense effectively mitigates leakage with negligible performance degradation.

## 2. Related Work

This work investigates the reverse-engineering risks of model editing. Due to the space limitation, we provide a brief overview of related work here and a more comprehensive discussion in Appendix B.

Prior work reveals various vulnerabilities in LLMs, ranging from security attacks like jailbreaking (Wei et al., 2023; Shen et al., 2024; Deng et al., 2023) and backdoors (Cai et al., 2022; Shi et al., 2023) to privacy risks such as membership inference (Kaneko et al., 2025; Feng et al., 2025) and personally identifiable information leakage (Nakka et al., 2024; Kim et al., 2023). To mitigate these risks, model editing has emerged as a surgical solution (Hossain & Kagal, 2025; Li et al., 2025). While current editing techniques span from external memory (Mitchell et al., 2022b; Huang et al., 2023; Hartvigsen et al., 2023; Yu et al., 2024) to meta-

learning (De Cao et al., 2021; Mitchell et al., 2022a; Zhang et al., 2024), the locate-then-edit paradigm (Meng et al., 2022; 2023; Fang et al., 2025) has emerged as a popular approach due to its parameter efficiency and interpretability. However, research on the privacy risks of this paradigm remains scarce.

Related to our approach, Youssef et al. (2025) recovers the pre-edit behavior of models, but cannot reconstruct the editing prompts and original outputs. Patil et al. (2024) detects original outputs by analyzing intermediate layer logits, while relying on the relatively strong assumption that the attacker has access to the original prompts. In contrast, our work accurately recovers all editing information by analyzing parameter updates, without requiring original prompts.

## 3. Preliminaries

### 3.1. Model Editing in LLMs

Let $f_\theta$ be an LLM parameterized by weights $\theta$. We define a batch of knowledge triples as $\mathcal{D}^t = \{d_1^t, \ldots, d_N^t\}$, where each tuple $d_i^t = (s_i, r_i, o_i)$. Here, $t$ represents knowledge triples, $r_i$ is a prompt template containing two placeholders: the first for the subject $s_i$, which together form the query context, and the second for the target output $o_i$ (e.g., $s_i$: Alice, $r_i$: {} lives in {}, $o_i$: New York).

In this work, we focus on the locate-then-edit paradigm, specifically ROME (Meng et al., 2022) and its variants (Meng et al., 2023; Fang et al., 2025). These methods implement knowledge editing by updating specific feed-forward networks (FFNs) that store factual associations. We focus on the single-time editing setting in the main text and leave sequential editing results in Appendix F.

For a given edit input, let key vector $\mathbf{k}_*$ denote the hidden state of the subject's last token at the target layer, where $\mathbf{k}_*$ is mapped to a value vector $\mathbf{v} = \mathbf{W}\mathbf{k}_*$ by the FFN weight matrix $\mathbf{W}$. The objective is to find an update $\Delta\mathbf{W}$ that remaps $\mathbf{k}_*$ to a desired target value $\mathbf{v}_*$, i.e., $(\mathbf{W} + \Delta\mathbf{W})\mathbf{k}_* = \mathbf{v}_*$. This is typically formulated as a (constrained) least-squares problem, aiming to satisfy the edit requirement without compromising the preserved (general) knowledge $\mathbf{K}_p$.

For single-fact editing ($N = 1$), ROME (Meng et al., 2022) defines the *residual vector* as $\mathbf{r}_* = \mathbf{v}_* - \mathbf{v}$ and derives a rank-one analytical solution:

$$\Delta\mathbf{W}_{\text{ROME}} = \frac{\mathbf{r}_*\mathbf{k}_*^\top\mathbf{C}^{-1}}{\mathbf{k}_*^\top\mathbf{C}^{-1}\mathbf{k}_*}, \tag{1}$$

where $\mathbf{C} \triangleq \mathbf{K}_p\mathbf{K}_p^\top$ is a pre-cached matrix, typically estimated using key vectors sampled from Wikipedia.

To support batch editing ($N > 1$), MEMIT (Meng et al., 2023) generalizes this approach: Let $\mathbf{K} \in \mathbb{R}^{d_{\text{in}} \times N}$ denote the stacked key matrix containing key vectors for the batch

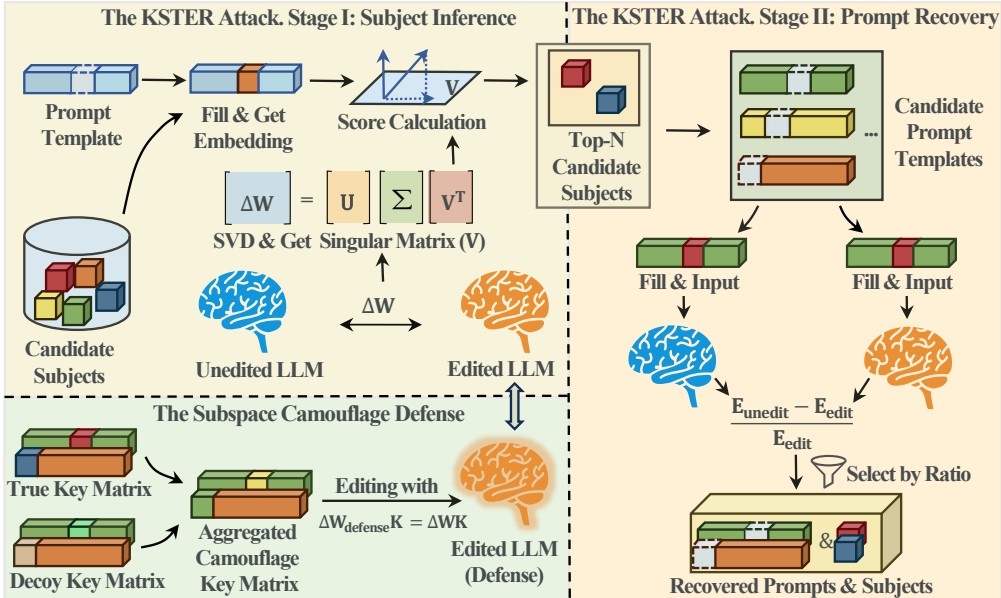

*Figure 1.* Overview of our proposed two-stage reverse-engineering attack framework *KSTER* in a white-box setting, along with an associated defense strategy *subspace camouflage*.

of samples, and $\mathbf{R} \in \mathbb{R}^{d_{\text{out}} \times N}$ denote the corresponding residual matrix. Its update is:

$$\Delta\mathbf{W}_{\text{MEMIT}} = \mathbf{R}\mathbf{K}^\top \left(\mathbf{C} + \mathbf{K}\mathbf{K}^\top\right)^{-1}. \qquad (2)$$

To mitigate model collapse[1] in sequential editing, AlphaEdit (Fang et al., 2025) maintains general model capabilities by projecting updates onto the null space of $\mathbf{K}_p$:

$$\Delta\mathbf{W}_{\text{AlphaEdit}} = \mathbf{R}\mathbf{K}^\top \mathbf{P} \left(\mathbf{K}\mathbf{K}^\top \mathbf{P} + \mathbf{I}\right)^{-1}, \qquad (3)$$

where $\mathbf{P}$ is the orthogonal projection matrix onto the null space of $\mathbf{K}_p$.

### 3.2. Threat Model

The attacker's goal is to reverse-engineer the knowledge $\mathcal{D}^t$ that has been edited. We consider two attacker settings: (1) *White-box*: The attacker has access to (i) the model parameter weights before ($\theta$) and after ($\theta'$) the edit; and (ii) the editing algorithm (e.g., MEMIT) and its hyperparameters. (2) *Gray-box*: The attacker has access only to the output logits of the model before and after editing, and the model parameters are inaccessible. In this work, we mainly explore the white-box setting, and we also include a gray-box baseline.

In both settings, we assume the attacker cannot access $\mathcal{D}^t$ but (1) has access to the covariance matrix $\mathbf{C}$ (we will show this assumption can be relaxed with an ablation study in §4.4); (2) has access to a joint candidate pool $\mathcal{S}_{\text{cand}} \times \mathcal{R}_{\text{cand}}$

---
[1]Refers to the severe degradation of general model capabilities as sequential edits accumulate.

containing the ground-truth edit(s) $(s, r)$. This assumption is reasonable as attackers can filter candidates based on public trends or specific domains, making the search space feasible to construct.

## 4. The KSTER Attack

In this section, we show how to reverse-engineer the edited data using the parameter difference $\Delta\theta$. It has two stages: (1) *Subject inference*: Given a candidate set of subjects $\mathcal{S}_{\text{cand}}$, identify the specific subject $\hat{s} \in \mathcal{S}_{\text{cand}}$ that is involved in the edit. (2) *Prompt recovery*: Given a candidate set of relations $\mathcal{R}_{\text{cand}}$, recover the prompt $\hat{r} \in \mathcal{R}_{\text{cand}}$ used in the edit context. With these components, we can obtain the original output $o^{\text{ori}}$ by querying the initial model $f_\theta$.

Below, we begin with a key observation on subject invariance, which provides the empirical basis for our method. We then present the proposed two-stage reverse-engineering attack framework.

### 4.1. Subject Invariance in LM Model Editing

In locate-then-edit algorithms such as MEMIT (Meng et al., 2023), the key matrix $\mathbf{K}$ is constructed by stacking the last-token hidden states of a subject batch at a specific FFN layer. A critical question for subject inference is whether the key matrix $\mathbf{K}$ is strongly affected by the prompt contexts used during editing.

We first analyze the attention distribution in Figure 2a, which reveals that the model's attention is primarily focused on subject tokens across different prompt templates. Motivated by this, we use cosine similarity to compare the

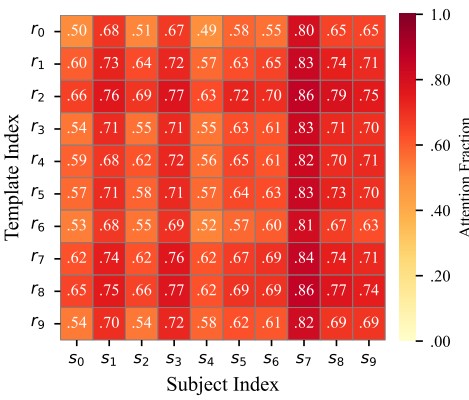

*(a)* Attention distribution across subjects and templates.

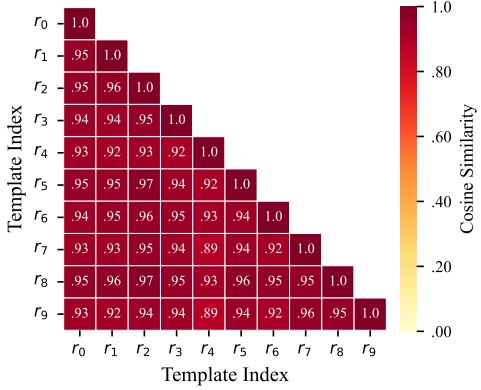

*(b)* Activation similarity across subjects and templates.

*Figure 2.* Subject invariance for MEMIT (Llama3-8B-Instruct). (a) Attention consistently focuses ($>50\%$) on subject tokens across different prompt templates $\{r_j\}$ for each subject $\{s_i\}$. (b) Different prompt templates filled with the same subject exhibit high similarity in their hidden states at the shallowest edited layer.

hidden states of the same subject at the shallowest edited layer across different prompt templates. As shown in Figure 2b, the results show a surprising consistency: the activations are highly similar across all templates. Detailed lists of the subjects and prompt templates are provided in Appendix E.3.

This observation gives us a key insight: **K primarily encodes unique features of the subjects and is largely decoupled from the prompt context**. Consequently, an attacker does not need to reconstruct the exact prompts to infer the edited subjects. Instead, the attacker can simply query the model with a generic template. If a candidate subject's activation vector aligns strongly with **K**, it is likely that the subject belongs to the edited batch.

### 4.2. Attack Stage I: Subject Inference

The first stage aims to recover the edited subject set $\{s_i\}_{i=1}^N$ from the update matrix $\Delta\mathbf{W}$ of the shallowest edited layer.

---

**Algorithm 1** The KSTER Attack (MEMIT).

**Input:** Pre-edit model $\theta$, post-edit model $\theta'$, weight update matrix $\Delta\mathbf{W}$, covariance $\mathbf{C}$, subject candidates $\mathcal{S}_{\mathrm{cand}}$, prompt candidates $\mathcal{R}_{\mathrm{cand}}$, generic template $\mathcal{T}_{\mathrm{gen}}$, activation extraction function $\mathcal{F}_\theta(\cdot)$, score function $\mathrm{Score}(\cdot \mid \cdot)$, number of recovered prompts per subject $N_r$.

**Output:** Predicted subject-prompts set $\hat{\mathcal{R}}$ (initially $\emptyset$).

```
// Stage I: Subject Inference.
```
1   $N \leftarrow \mathrm{Rank}(\Delta\mathbf{W})$.
2   $\mathbf{M} \leftarrow \Delta\mathbf{W}\mathbf{C}$.
3   $[\mathbf{U}, \mathbf{\Sigma}, \mathbf{V}^\top] \leftarrow \text{Singular-value-decomposition}(\mathbf{M})$.
4   $\mathbf{V}_N \leftarrow \mathbf{V}[:, 1:N]$.
5   $\mathcal{J} \leftarrow \emptyset$.
6   **for** $s_i^c \in \mathcal{S}_{\mathrm{cand}}$ **do**
7      $\mathbf{k}_i^c \leftarrow \mathcal{F}_\theta(s_i^c, \mathcal{T}_{\mathrm{gen}})$.
8      $\rho_i^c \leftarrow \dfrac{\left\|\mathbf{V}_N^\top \mathbf{k}_i^c\right\|_2}{\left\|\mathbf{k}_i^c\right\|_2}$.
9      $\mathcal{J} \leftarrow \mathcal{J} \cup \{(s_i^c, \rho_i^c)\}$.
10   $\hat{\mathcal{S}} \leftarrow \text{Top-}N(\{s_i^c \mid (s_i^c, \rho_i^c) \in \mathcal{J}\})$.
```
// Stage II: Prompt Recovery.
```
11   **for** $\hat{s}_i \in \hat{\mathcal{S}}$ **do**
12      $\mathcal{Q} \leftarrow \emptyset$.
13      **for** $r_j^c \in \mathcal{R}_{\mathrm{cand}}$ **do**
14          $\gamma_{i,j}^c \leftarrow \mathrm{Score}(r_j^c \mid \hat{s}_i, \theta, \theta')$.
15          $\mathcal{Q} \leftarrow \mathcal{Q} \cup \{(r_j^c, \gamma_{i,j}^c)\}$.
16      $\hat{\mathcal{R}}_i \leftarrow \text{Top-}N_r(\{r_j^c \mid (r_j^c, \gamma_{i,j}^c) \in \mathcal{Q}\})$.
17      $\hat{\mathcal{R}} \leftarrow \hat{\mathcal{R}} \cup \{(\hat{s}_i, \hat{\mathcal{R}}_i)\}$.
18   **return** $\hat{\mathcal{R}}$.

---

Although it is hard to recover $\mathbf{K}$ directly because of the unknown residual matrix $\mathbf{R}$ (or residual vector $\mathbf{r}_*$), we can indeed reconstruct the column space of $\mathbf{K}$ (namely key space) through a carefully designed mathematical transformation. The intuition here is to apply the Woodbury matrix identity (Lemma G.9) to rewrite the weight update $\Delta\mathbf{W}$.

Here we take MEMIT as an example. By applying the Woodbury matrix identity, the weight update in Eq. (2) can be written as:

$$\Delta\mathbf{W} = \mathbf{R}\left(\mathbf{I} + \mathbf{K}^\top \mathbf{C}^{-1} \mathbf{K}\right)^{-1} \mathbf{K}^\top \mathbf{C}^{-1}. \quad (4)$$

The detailed derivations can be found in Eq. (24), Appendix G.2. This reveals an important property that the row space of $\Delta\mathbf{W}\mathbf{C}$ lies within the column space of $\mathbf{K}$:

$$\mathrm{RowSpace}(\Delta\mathbf{W}\mathbf{C}) \subseteq \mathrm{ColSpace}(\mathbf{K}). \quad (5)$$

Specifically, the signature of the edited subjects is preserved in the row space of $\Delta\mathbf{W}\mathbf{C}$.

Based on the geometric property in Eq. (5), we propose our subject inference attack for MEMIT, as shown in lines 1–10 of Algorithm 1. The procedure has the following two steps:

**Step 1: Subspace Reconstruction.** To recover the subspace associated with the target subjects, we first determine the number of edited knowledge $N$ by analyzing the rank of weight update $\Delta\mathbf{W}$: since generally $\mathbf{R}$ and $\mathbf{K}$ are full column rank, the rank of $\Delta\mathbf{W}$ is exactly $N$, which is detailed in Lemma G.10 and Remark G.11 in the appendix. Then, we eliminate the geometric distortion induced by the covariance matrix: by computing $\mathbf{M} = \Delta\mathbf{W}\mathbf{C}$ and its singular value decomposition $\mathbf{M} = \mathbf{U}\mathbf{\Sigma}\mathbf{V}^{\top}$, the subspace spanned by the edited key vectors (equivalently the column space of $\mathbf{K}$) is revealed by $\mathbf{V}_N$, whose columns are the top-$N$ right singular vectors of $\mathbf{M}$.

**Step 2: Projection-based Ranking.** In the second step, we score each candidate subject $s_i^c \in \mathcal{S}_{\text{cand}}$ (see Appendix E.1 for the detailed construction of $\mathcal{S}_{\text{cand}}$) by generating a proxy key vector $\mathbf{k}_i^c = \mathcal{F}_\theta(s_i^c, \mathcal{T}_{\text{gen}})$ using a generic template (see Appendix E.4 for details). Here, $\mathcal{F}_\theta(s_i^c, \mathcal{T}_{\text{gen}})$ denotes the activation vector at the shallowest edited layer of the pre-edit model $\theta$, corresponding to the subject's last token. This vector is obtained by a forward pass with the subject-filled template. We then compute the projection coefficient of $\mathbf{k}_i^c$ onto $\text{ColSpace}(\mathbf{V}_N)$, which is $\rho_i^c = \frac{\|\mathbf{V}_N^{\top}\mathbf{k}_i^c\|_2}{\|\mathbf{k}_i^c\|_2}$. A higher score indicates that the candidate is likely one of the edited targets. Finally, we obtain the top-$N$ predicted subjects $\hat{\mathcal{S}}$. We further perform an ablation study on the selection of templates in Appendix F.

The application of our techniques to other locate-then-edit methods, such as ROME (Meng et al., 2022) and AlphaEdit (Fang et al., 2025), is provided in Appendix C.1 and C.2. Notably, the scoring of AlphaEdit differs slightly from MEMIT due to the projection matrix $\mathbf{P}$, as explained in Lemma G.18 and Remark G.19 (Appendix G.4).

Theoretical Guarantees. In Appendix G, we provide theoretical guarantees for the correctness and robustness of the subject inference. Under appropriate assumptions established in Appendix G.1, we prove that Algorithm 1 (MEMIT) exactly recovers the correct $N$ edited subjects in Theorem G.13. A similar guarantee for the attack strategy of AlphaEdit (Algorithm 3, which differs from Algorithm 1 in its score function $\rho^c$) is provided in Theorem G.21. Furthermore, to address practical scenarios where the covariance matrix $\mathbf{C}$ is noisily estimated, we also prove that Algorithm 1 is robust to perturbation on $\mathbf{C}$ in Theorem G.16, by utilizing subspace perturbation bounds.

### 4.3. Attack Stage II: Prompt Recovery

After inferring the target subject set $\hat{\mathcal{S}}$ in Stage I, we now aim to recover the corresponding prompt for each subject $\hat{s}_i \in \hat{\mathcal{S}}$. We formulate this part as an *entropy reduction analysis* between the pre-edit model ($\theta$) and the post-edit model ($\theta'$).

**Entropy Reduction Analysis.** A key consequence of model editing is a sharp reduction in uncertainty for the edited fact. When queried with the target subject and prompt, the pre-edit model ($\theta$) may exhibit high entropy, whereas the edited model ($\theta'$) is optimized to output the desired answer with near-zero entropy (i.e., extreme confidence) (Meng et al., 2022; 2023; Fang et al., 2025). Since this optimization does not apply to non-target prompts, the entropy change for them is much smaller. Leveraging this, we employ an entropy-based approach to identify the most likely prompt.

We evaluate a set of candidate prompts $\mathcal{R}_{\text{cand}}$, which contains the ground-truth target prompts (see Appendix E.1 for the detailed construction of $\mathcal{R}_{\text{cand}}$). For each candidate $r_j^c \in \mathcal{R}_{\text{cand}}$, we compute the Shannon entropy of the next-token distribution conditioned on subject $\hat{s}_i$:

$$H(\hat{s}_i, r_j^c; \theta) = -\sum_{t \in \mathcal{V}} \mathbb{P}_\theta(t \mid \hat{s}_i, r_j^c) \log \mathbb{P}_\theta(t \mid \hat{s}_i, r_j^c), \quad (6)$$

where $\mathcal{V}$ denotes the vocabulary and $\mathbb{P}_\theta(\cdot \mid \hat{s}_i, r_j^c)$ denotes next-token probability predicted by the model $\theta$. This formulation is designed to capture the model's *prediction confidence* regarding a specific prompt.

We use the *relative entropy reduction* as our discriminator. This metric measures the entropy shift from the pre-edit to the post-edit model, normalized by the post-edit prediction confidence:

$$\text{Score}(r_j^c \mid \hat{s}_i, \theta, \theta') = \frac{H(\hat{s}_i, r_j^c; \theta) - H(\hat{s}_i, r_j^c; \theta')}{H(\hat{s}_i, r_j^c; \theta') + \epsilon}, \quad (7)$$

where $\epsilon$ is set to $10^{-9}$ for numerical stability. As illustrated in lines 11–18 of Algorithm 1, for each subject $\hat{s}_i \in \hat{\mathcal{S}}$, we predict the top-$N_r$ prompts $\hat{\mathcal{R}}_i$ as the most likely original prompts. The denominator ensures that prompts gaining over-confidence (where $H(\hat{s}_i, r_j^c; \theta') \to 0$) receive higher scores, precisely identifying the edited prompt.

Finally, note that if the inference of the subject and the prompt is correct, it should then be easy to get the target object. Therefore, we also provide the end-to-end target object extraction results in our evaluation.

### 4.4. Attack Evaluation

In this section, we conduct experiments to address the following research questions:

- **RQ1:** How reliably can our attack pinpoint the edited subjects across different editing methods?

*Table 1.* Performance comparison of subject inference attack for batch-edit tasks. We report top-$N$ recall.

| Model | $N$ | CounterFact | | | | zsRE | | | |
|---|---|---|---|---|---|---|---|---|---|
| | | MEMIT | | AlphaEdit | | MEMIT | | AlphaEdit | |
| | | Ours (White-box) | Gray-box | Ours (White-box) | Gray-box | Ours (White-box) | Gray-box | Ours (White-box) | Gray-box |
| GPT-J | 10 | **0.98** ± 0.04 | 0.96 ± 0.05 | **0.98** ± 0.04 | 0.96 ± 0.05 | **0.98** ± 0.04 | **0.98** ± 0.04 | **0.98** ± 0.04 | **0.98** ± 0.04 |
| | 50 | **0.96** ± 0.01 | 0.87 ± 0.06 | **0.96** ± 0.03 | 0.85 ± 0.10 | **0.99** ± 0.01 | 0.92 ± 0.04 | **0.99** ± 0.01 | 0.96 ± 0.03 |
| | 100 | **0.95** ± 0.01 | 0.88 ± 0.03 | **0.96** ± 0.01 | 0.86 ± 0.04 | **0.99** ± 0.01 | 0.92 ± 0.02 | **0.99** ± 0.01 | 0.96 ± 0.01 |
| Llama-3 -Instruct | 10 | **1.00** ± 0.00 | 0.96 ± 0.05 | **1.00** ± 0.00 | 0.76 ± 0.11 | **0.98** ± 0.04 | 0.82 ± 0.08 | **0.98** ± 0.04 | 0.70 ± 0.14 |
| | 50 | **0.99** ± 0.01 | 0.79 ± 0.05 | **0.99** ± 0.02 | 0.52 ± 0.05 | **0.99** ± 0.01 | 0.74 ± 0.04 | **0.99** ± 0.01 | 0.44 ± 0.07 |
| | 100 | **0.99** ± 0.01 | 0.68 ± 0.04 | **0.99** ± 0.01 | 0.45 ± 0.03 | **0.99** ± 0.01 | 0.69 ± 0.02 | **0.99** ± 0.01 | 0.43 ± 0.07 |
| Qwen-2.5 -Instruct | 10 | **1.00** ± 0.00 | 0.88 ± 0.11 | **1.00** ± 0.00 | 0.80 ± 0.10 | **0.98** ± 0.04 | 0.90 ± 0.10 | **0.98** ± 0.04 | 0.84 ± 0.16 |
| | 50 | **0.93** ± 0.06 | 0.63 ± 0.15 | **0.93** ± 0.07 | 0.54 ± 0.13 | **0.98** ± 0.01 | 0.68 ± 0.02 | **0.99** ± 0.01 | 0.61 ± 0.06 |
| | 100 | **0.94** ± 0.03 | 0.59 ± 0.11 | **0.95** ± 0.03 | 0.51 ± 0.06 | **0.98** ± 0.01 | 0.58 ± 0.03 | **0.99** ± 0.01 | 0.55 ± 0.04 |

*Table 2.* Performance of prompt recovery attack of MEMIT. We report different recall (Top-$N_r$) and semantic similarity (Sim.).

| Model | $N$ | CounterFact | | | | zsRE | | | |
|---|---|---|---|---|---|---|---|---|---|
| | | Top-1 | Top-5 | Top-20 | Sim. | Top-1 | Top-5 | Top-20 | Sim. |
| GPT-J | 10 | 0.57 ± 0.07 | 0.86 ± 0.05 | 0.96 ± 0.09 | 0.88 ± 0.02 | 0.28 ± 0.05 | 0.70 ± 0.16 | 0.86 ± 0.09 | 0.83 ± 0.03 |
| | 50 | 0.51 ± 0.06 | 0.90 ± 0.02 | 0.99 ± 0.02 | 0.88 ± 0.01 | 0.33 ± 0.03 | 0.64 ± 0.02 | 0.85 ± 0.03 | 0.82 ± 0.01 |
| | 100 | 0.50 ± 0.05 | 0.92 ± 0.02 | 0.99 ± 0.01 | 0.88 ± 0.01 | 0.32 ± 0.07 | 0.61 ± 0.06 | 0.83 ± 0.06 | 0.82 ± 0.01 |
| Llama-3 -Instruct | 10 | 0.53 ± 0.07 | 0.90 ± 0.01 | 0.96 ± 0.05 | 0.88 ± 0.02 | 0.40 ± 0.10 | 0.64 ± 0.09 | 0.90 ± 0.07 | 0.86 ± 0.01 |
| | 50 | 0.49 ± 0.03 | 0.82 ± 0.04 | 0.94 ± 0.03 | 0.88 ± 0.01 | 0.40 ± 0.04 | 0.60 ± 0.05 | 0.76 ± 0.04 | 0.84 ± 0.02 |
| | 100 | 0.51 ± 0.05 | 0.81 ± 0.03 | 0.94 ± 0.02 | 0.88 ± 0.01 | 0.33 ± 0.04 | 0.57 ± 0.06 | 0.76 ± 0.05 | 0.83 ± 0.01 |
| Qwen-2.5 -Instruct | 10 | 0.35 ± 0.22 | 0.83 ± 0.17 | 0.96 ± 0.09 | 0.85 ± 0.03 | 0.04 ± 0.05 | 0.42 ± 0.21 | 0.60 ± 0.17 | 0.81 ± 0.03 |
| | 50 | 0.42 ± 0.05 | 0.81 ± 0.07 | 0.95 ± 0.06 | 0.86 ± 0.01 | 0.13 ± 0.07 | 0.34 ± 0.16 | 0.60 ± 0.07 | 0.79 ± 0.03 |
| | 100 | 0.47 ± 0.06 | 0.86 ± 0.02 | 0.97 ± 0.03 | 0.86 ± 0.01 | 0.08 ± 0.06 | 0.29 ± 0.12 | 0.55 ± 0.10 | 0.78 ± 0.03 |

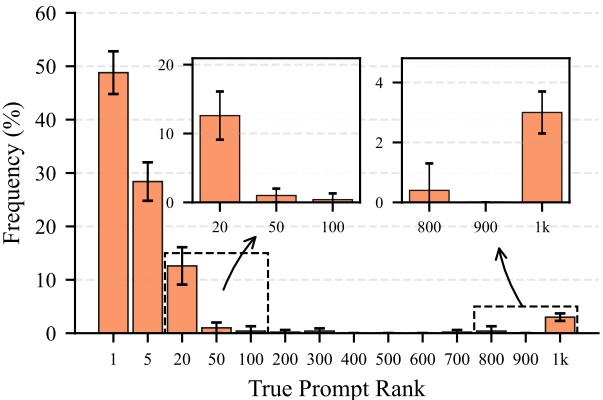

*Figure 3.* Distribution of true prompt ranks for Llama3-8B-Instruct on the CounterFact dataset. The x-axis labels denote the upper bound of each rank interval (e.g., 50 represents the range $(20, 50]$).

- **RQ2:** How robust is our attack against estimation noise and distribution shifts in the covariance matrix?
- **RQ3:** How effectively can our attack recover the semantic essence of the original prompts and original outputs?

- **RQ4:** What are the failure cases of our attack, and what explains these failures?

**Experimental Setup.** We employ three LLMs: GPT-J (6B) (Wang & Komatsuzaki, 2021), Llama3-8B-Instruct (Dubey et al., 2024), and Qwen2.5-7B-Instruct (Hui et al., 2024), to evaluate our attack framework, covering ROME (Meng et al., 2022) for single-edit tasks and batch editing methods like MEMIT (Meng et al., 2023) and AlphaEdit (Fang et al., 2025). Experiments are conducted on two standard benchmarks, CounterFact (Meng et al., 2022), and zsRE (Levy et al., 2017), where we construct specific candidate libraries for subjects and prompts to simulate realistic attack scenarios. For full details, please refer to Appendix E.1.

**Baseline & Metrics.** Since reverse-engineering attacks on model editing are under-explored, we establish a gray-box baseline (Appendix E.1) based on the divergence of output distributions to compare with our white-box approach for subject inference. We evaluate the attack performance using the following metrics: subject recall and projection coefficient for subject inference; Prompt recall and semantic similarity for prompt recovery (Reimers & Gurevych, 2019).

Please refer to Appendix E.1 for details.

**RQ1: Effectiveness of Subject Inference.** As shown in Table 1, our white-box attack maintains high accuracy across all batch sizes, with recall consistently staying between 0.94 and 1.00. In contrast, while the gray-box baseline matches our performance when $N = 10$, its performance drops notably as $N$ increases, falling to 0.43–0.55 at $N = 100$. This indicates that while logit changes are sufficient to identify the subject in simple cases, they become entangled and overlapped in larger batches, making it difficult to distinguish specific targets through logits alone. For the impact of candidate set size, sequential editing setting on the results, and additional experiments on ROME, please refer to Tables 12, 13, 6 in Appendix F.

**RQ2: Robustness to Covariance Estimation.** To evaluate robustness, we vary both the data source and sample size for covariance estimation. While our default $\mathbf{C}$ is computed from 100,000 Wikipedia samples, Figures 9a and 9b (Appendix F) show that our attack remains highly effective with far less data. Attack performance against MEMIT converges at 100 samples across Wikipedia (Lhoest et al., 2021), WikiText (Merity et al., 2017), and Pile (Gao et al., 2020). Notably, our attack on AlphaEdit exhibits superior stability, maintaining peak performance even at just 10 samples. These results confirm that our attack is robust to distribution shifts and feasible in limited-data scenarios. We further provide a theoretical analysis for the case of MEMIT in Appendix G.3.

**RQ3: Performance of Prompt and Output Recovery.** Table 2 presents the performance of our prompt recovery attack. On the CounterFact dataset, the attack achieves high-fidelity recovery: top-20 recall consistently reaches 0.94–0.99, and top-1 recall often exceeds 50%. This indicates that for completion-style edits, our method effectively compresses the search space to a verifiable range. On the more challenging zsRE benchmark, although recall decreases due to the complex syntax of the QA format, the attack still maintains a stable semantic similarity (0.78–0.88). These results highlight the robustness of our attack across diverse settings, underscoring critical information leakage risks in model editing. For the impact of candidate set size, the comparison between our method and the naive method (only observe the logit of post-edit model, see Appendix E.1), as well as additional experimental results for ROME and AlphaEdit, please refer to Tables 14, 15, 7 and 5 of Appendix F. Furthermore, Table 3 illustrates the end-to-end results for extracting original outputs. On the CounterFact dataset, our approach recovers over 70% of the original outputs at Top-20, showing the effectiveness of KSTER.

**RQ4: Failure Cases of Our Attack.** Although our prompt recovery attack achieves high recovery rates in most cases, the rank distribution in Figure 3 shows a bimodal pattern of

*Table 3.* End-to-end attack recall using MEMIT in $N = 100$.

| Dataset | Model | Top-1 | Top-5 | Top-20 |
|---------|-------|-------|-------|--------|
| **CounterFact** | GPT-J | 0.35 | 0.64 | 0.78 |
| | Llama-3-Instruct | 0.32 | 0.61 | 0.74 |
| | Qwen-2.5-Instruct | 0.33 | 0.64 | 0.81 |
| **zsRE** | GPT-J | 0.28 | 0.47 | 0.62 |
| | Llama-3-Instruct | 0.07 | 0.21 | 0.42 |
| | Qwen-2.5-Instruct | 0.25 | 0.53 | 0.65 |

failures. Our analysis of the main reasons is as follows:

- **Semantic Generalization Error (Rank 6–100):** The intermediate rankings reveal *semantic ambiguity* in edit generalization. When the model fails to identify a subject's specific type, it may activate prompts from broader or incorrect categories. As illustrated by the case of "Anaal Nathrakh" in Figure 5 (Appendix F), the model treats the subject as a "location-associated entity" rather than a "musical group." Consequently, generic location prompts (e.g., `{} was developed in {}, {} was formulated in {}`) are activated with the ground truth (`{} was created in {}`). These mismatched prompts induce a competitive entropy drop that pushes the true prompt into the 6–100 range.

- **Optimization Constraint Conflict (Rank 700–1k):** The tail of the distribution exposes *confused edits*, where the editing objective conflicts with statistical priors. Locate-then-edit algorithms solve a least-squares problem to map subjects to target outputs, constrained by $\mathbf{C}$. In these outlier cases, the required modification contradicts the statistics encoded in $\mathbf{C}$. Although the edit increases the target token's probability, the mismatch between the actual activations and the optimized target $\mathbf{R}$ prevents the model from reaching a low-entropy regime. Instead, the output entropy rises, rendering our metric ineffective and pushing the true prompt into the tail of the ranking. This notably compromises the specificity of the edit.

## 5. Defense Strategy: Subspace Camouflage

Our attack (§4) exhibits that the weight update $\Delta\mathbf{W}$ leaks the key space $\mathrm{ColSpace}(\mathbf{K})$, providing a backdoor for the attacker to accurately obtain the edited knowledge. To address this problem, we propose our defense strategy named *subspace camouflage* in Algorithm 2, thwarting the attacker's attempt to extract the subspace by perturbing $\mathbf{K}$ in the update calculation.

### 5.1. Subspace Camouflage via Semantic Decoy

The underlying mathematical principle for our subject inference attack (Algorithm 1) is the reconstruction of the key space $\mathrm{ColSpace}(\mathbf{K})$, through the (distorted) row space

**Algorithm 2** The Subspace Camouflage Defense (MEMIT).

**Input:** Pre-edit model $\theta$, true key matrix $\mathbf{K}$, original weight update $\Delta\mathbf{W}$, covariance $\mathbf{C}$, decoy subjects set $\mathcal{S}_{\text{decoy}}$, camouflage strength $\alpha$, activation extraction function $\mathcal{F}_\theta(\cdot)$, generic template $\mathcal{T}_{\text{gen}}$, ridge parameter $\lambda = 10^{-8}$.

**Output:** Defensive weight update $\Delta\mathbf{W}_{\text{defense}}$.

1  $\mathbf{K}_{\text{decoy}} \leftarrow \mathcal{F}_\theta(\mathcal{S}_{\text{decoy}}, \mathcal{T}_{\text{gen}})$.
2  $\tilde{\mathbf{K}} \leftarrow \mathbf{K} + \alpha \frac{\|\mathbf{K}\|_2}{\|\mathbf{K}_{\text{decoy}}\|_2} \mathbf{K}_{\text{decoy}}$.
3  $\mathbf{G} \leftarrow \tilde{\mathbf{K}}^\top \mathbf{C}^{-1} \mathbf{K}$.
4  $\mathbf{G}_{\text{reg}} \leftarrow \mathbf{G} + \lambda\mathbf{I}$.
5  $\Delta\mathbf{W}_{\text{defense}} \leftarrow \Delta\mathbf{W}\mathbf{K}(\mathbf{G}_{\text{reg}})^{-1}\tilde{\mathbf{K}}^\top\mathbf{C}^{-1}$.
6  **return** $\Delta\mathbf{W}_{\text{defense}}$.

of the weight update $\Delta\mathbf{W}$. Since the attacker's inference depends exclusively on this subspace, a successful defense only needs to (i) alter this subspace from $\text{ColSpace}(\mathbf{K})$ to a camouflaged subspace $\text{ColSpace}(\tilde{\mathbf{K}})$ while (ii) preserving the update's effect on the original keys. Intuitively, we aim to modify the update so that it behaves identically on the true edited subjects, while presenting a misleading low-rank structure to any subspace-based analysis.

To obtain this camouflaged subspace, we construct the *aggregated camouflage key matrix* $\tilde{\mathbf{K}}$ by perturbing the true key matrix $\mathbf{K}$ with a decoy key matrix $\mathbf{K}_{\text{decoy}} \in \mathbb{R}^{d_{\text{in}} \times N}$. This decoy matrix is sampled from a set of decoy subjects $\mathcal{S}_{\text{decoy}}$, which consists of real but unrelated subjects (see Appendix E.2 for details):

$$\tilde{\mathbf{K}} = \mathbf{K} + \alpha \cdot \frac{\|\mathbf{K}\|_2}{\|\mathbf{K}_{\text{decoy}}\|_2} \mathbf{K}_{\text{decoy}}. \quad (8)$$

Unlike adding random noise on $\mathbf{K}$, this *semantic decoy* approach corresponds to real key vectors and thus actively competes with the target keys in the recovered subspace. Here $\alpha \in \mathbb{R}_+$ controls the magnitude of the injected decoys: a larger $\alpha$ enhances protection by making the reconstructed subspace more decoy-dominant, while introducing risk of degradation of the model's general capabilities.

Below we take MEMIT as an example, while the specific defense strategies for ROME and AlphaEdit are provided in Appendix D.1 and D.2. Formally, we replace the weight update $\Delta\mathbf{W}$ by a defense weight $\Delta\mathbf{W}_{\text{defense}}$ which satisfies:

$$\begin{cases} \text{RowSpace}(\Delta\mathbf{W}_{\text{defense}}\mathbf{C}) & \subseteq \text{ColSpace}(\tilde{\mathbf{K}}), \\ \Delta\mathbf{W}_{\text{defense}}\mathbf{K} & = \Delta\mathbf{W}\mathbf{K}. \end{cases} \quad (9)$$

By solving the constraints above (detailed in Theorem H.1, Appendix H), $\Delta\mathbf{W}_{\text{defense}}$ is uniquely determined as:

$$\Delta\mathbf{W}_{\text{defense}} = \Delta\mathbf{W}\mathbf{K}\left(\tilde{\mathbf{K}}^\top\mathbf{C}^{-1}\mathbf{K}\right)^{-1}\tilde{\mathbf{K}}^\top\mathbf{C}^{-1}. \quad (10)$$

To improve numerical stability, we apply a small ridge regularization to $\tilde{\mathbf{K}}^\top\mathbf{C}^{-1}\mathbf{K}$ before inversion. The complete pipeline of our defense strategy is detailed in Algorithm 2. Notice that our defense weight update reduces to the original weight update as $\alpha \to 0^+$, as declared in Remark H.2.

Although our defense strategy is motivated by the *KSTER* attack, it is theoretically robust against a broader class of white-box attacks that rely solely on the singular structure of $\Delta\mathbf{W}$. Specifically, one cannot (i) identify whether this subspace camouflage defense strategy is applied to the weight change nor (ii) recover any meaningful information of the key matrix $\mathbf{K}$ beyond what is revealed by $\tilde{\mathbf{K}}$, without additional knowledge of the residual matrix $\mathbf{R}$.[2] Formal statements and proofs are provided in Theorem H.3 and H.4 in Appendix H, respectively.

### 5.2. Defense Evaluation

We evaluate our defense mechanism to answer the following research questions:

- **RQ1:** How does the scaling of the camouflage coefficient $\alpha$ affect editing performance?
- **RQ2:** How protection impacts model utility and the hallucinations of decoy subjects?

**Experimental Setup & Metrics.** We evaluate our defense on Llama3-8B-Instruct (Dubey et al., 2024) using CounterFact (Meng et al., 2022) and zsRE (Levy et al., 2017), with our white-box subject inference attack (§4.2) serving as the attacker. The evaluation primarily focuses on the following three dimensions: editing performance, editing protection, and model utility. For detailed hyperparameter settings and metric definitions, please refer to Appendix E.2.

**RQ1: Impact of $\alpha$ on Editing Performance.** We evaluate the impact of the camouflage coefficient $\alpha$ on editing performance and protection using the CounterFact dataset. As illustrated in Table 4, with $\alpha$ increasing from 0 to 5, the average rank of the true subjects rises substantially from

---

[2]Crucially, the construction of residual matrix $\mathbf{R}$ in model editing algorithms requires the subtraction of target value vector/matrix and the original target value vector/matrix, thus requiring *knowledge of the original prompts and subjects both before and after editing*, which are inherently tied to the model's original and target outputs.

This creates a circular dependency: the information needed to recover $\mathbf{R}$ is equivalent to obtaining the attack's ultimate goal, that is, extracting protected knowledge from the edited model. Therefore, attempting to bypass the defense by recovering $\mathbf{R}$ is, in practice, a logically circular, or no easier than directly breaking the defense itself.

*Table 4.* Performance comparison on CounterFact dataset across camouflage scales $\alpha \in \{0, 1, 3, 5, 7\}$.

| Method | $\alpha$ | Rank | Efficacy | Generalization | Specificity | Fluency | Consistency |
|--------|----------|------|----------|----------------|-------------|---------|-------------|
| **MEMIT** | 0 | $50.83 \pm 0.21$ | $0.95 \pm 0.01$ | $0.52 \pm 0.03$ | $0.24 \pm 0.03$ | $6.33 \pm 0.01$ | $0.31 \pm 0.01$ |
| | 1 | $148.62 \pm 1.39$ | $0.98 \pm 0.01$ | $0.49 \pm 0.01$ | $0.22 \pm 0.02$ | $6.33 \pm 0.01$ | $0.30 \pm 0.01$ |
| | 3 | $206.47 \pm 14.64$ | $0.96 \pm 0.02$ | $0.52 \pm 0.03$ | $0.24 \pm 0.03$ | $6.34 \pm 0.01$ | $0.31 \pm 0.01$ |
| | 5 | $394.12 \pm 35.40$ | $0.96 \pm 0.00$ | $0.53 \pm 0.02$ | $0.23 \pm 0.03$ | $6.32 \pm 0.01$ | $0.31 \pm 0.00$ |
| | 7 | $634.39 \pm 48.24$ | $0.91 \pm 0.03$ | $0.42 \pm 0.05$ | $0.19 \pm 0.01$ | $6.26 \pm 0.03$ | $0.30 \pm 0.01$ |
| **AlphaEdit** | 0 | $50.79 \pm 0.30$ | $0.94 \pm 0.02$ | $0.51 \pm 0.06$ | $0.24 \pm 0.03$ | $6.33 \pm 0.01$ | $0.30 \pm 0.01$ |
| | 1 | $150.28 \pm 0.48$ | $0.99 \pm 0.00$ | $0.56 \pm 0.04$ | $0.23 \pm 0.03$ | $6.33 \pm 0.01$ | $0.32 \pm 0.00$ |
| | 3 | $234.89 \pm 13.54$ | $1.00 \pm 0.00$ | $0.65 \pm 0.06$ | $0.23 \pm 0.02$ | $6.32 \pm 0.01$ | $0.33 \pm 0.01$ |
| | 5 | $398.87 \pm 19.52$ | $1.00 \pm 0.00$ | $0.64 \pm 0.03$ | $0.20 \pm 0.01$ | $6.29 \pm 0.02$ | $0.32 \pm 0.01$ |
| | 7 | $613.96 \pm 46.71$ | $0.46 \pm 0.33$ | $0.16 \pm 0.19$ | $0.04 \pm 0.04$ | $6.07 \pm 0.08$ | $0.30 \pm 0.01$ |

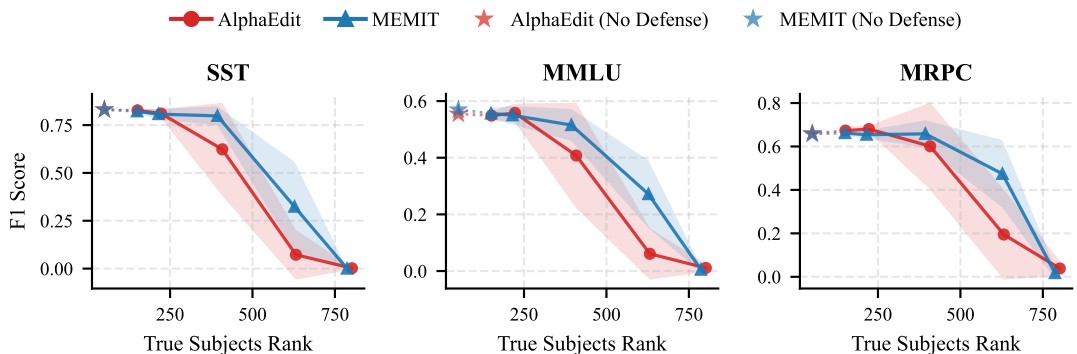

*Figure 4.* Protection-utility trade-off for Llama3-8B on CounterFact ($N = 100$). Shaded regions indicate standard deviation over 5 runs.

50.83 to 394.12 for MEMIT, indicating stronger protection, while efficacy and generalization remain high within this stable range. However, performance degrades sharply at excessive scales (e.g., $\alpha = 7$), where AlphaEdit's efficacy drops to 0.46. This occurs because an excessively large $\alpha$ causes the model update to deviate too far from the original least-squares solution. This destroys the model's feature extraction capability for other tokens, eventually leading to a decline in editing performance and the model's general capabilities. For our protection performance against the gray-box baseline as well as additional experimental results regarding ROME and the zsRE dataset, please refer to Tables 16, 8, 9, and 10 in Appendix F.

**RQ2: The Protection-Utility Trade-off.** Figure 4 shows the protection-utility trade-off, confirming that stronger subject concealment generally sacrifices the model's general capability. However, we can still achieve an effective balance: with $\alpha = 5$, the model maintains high general utility while providing strong protection. In addition, the shaded regions (standard deviations) show that AlphaEdit is more unstable than MEMIT. This stems from AlphaEdit's reliance on the projection matrix $\mathbf{P}$, which typically has smaller eigenvalues than MEMIT's $\mathbf{C}^{-1}$. When computing the weight update matrix $\Delta\mathbf{W}_{\text{defense}}$, inverting a matrix with these near-zero eigenvalues renders AlphaEdit's $\Delta\mathbf{W}_{\text{defense}}$ more ill-conditioned, which amplifies its sensitivity to sam-

pling noise and leads to greater variance across different runs. More comprehensive evaluation results are detailed in Figures 6, 7, and 8 of Appendix F.

Moreover, as illustrated by Table 17 in Appendix F, when $\alpha \geq 3$, the hallucination level of decoy subjects increases gradually. Specifically, the TFR declines from 0.51 to below 0.43, and the fluency drops from 6.23 to below 6.18. Overall, these results show a clear trade-off between the protection level and the model's general utility.

# 6. Conclusion

This work reveals that model editing inadvertently leaks the data it aims to edit. By analyzing the algebraic structure of parameter updates, we propose a reverse-engineering attack (named KSTER) that accurately recovers sensitive data. To mitigate this, we introduce *subspace camouflage*, a defense mechanism that obscures the fingerprint of subjects within the editing updates. Extensive evaluations show that our attack achieves high success rates, while our defense effectively protects the editing process without compromising editing efficacy. Furthermore, we identify the phenomena of *semantic ambiguity* and *confused edits* in existing methods, offering insights for the community to better investigate this technology. Detailed discussions on the limitations of our approach are provided in Appendix A.

## Impact Statement

In this work, we propose a reverse-engineering attack targeting the locate-then-edit paradigm, along with a defensive strategy to mitigate the vulnerabilities it reveals. Model editing has long been regarded as an efficient, retraining-free approach to erase sensitive data inadvertently memorized by LLMs. However, our research reveals that the parameter updates in the existing locate-then-edit paradigm can severely leak the very data intended for editing.

This discovery exposes a dual-sided impact. On the malicious side, attackers could exploit our proposed attack to reconstruct sensitive information directly from open-weights updates, thereby posing a formidable challenge to the open-source community. To mitigate such threats, we offer a camouflage-based defense as a proactive solution to help developers safeguard their edits. In contrast, on the beneficial side, our attack functions as a "security ruler" for model editing auditing. It empowers regulatory authorities and third-party organizations to verify the intent of model updates, therefore ensuring that edits are strictly for claimed corrections and not for injecting backdoors or concealing malicious biases.

Furthermore, our work provides unique insights into how LLMs store and access information. We observe the property of *subject invariance* in the parameter space and, through our attack, reveal the phenomena of *semantic ambiguity* and *confused edits* inherent in existing editing methods. These findings deepen our understanding of the intrinsic constraints of model editing and help facilitate future developments in the field.

Finally, we look forward to further research on developing more secure editing frameworks. We encourage the community to investigate whether existing defensive techniques can be scaled to thwart the reverse-engineering attack without compromising model utility, and we suggest that policymakers and platform maintainers consider "edit auditability" as a standard compliance requirement.

## Acknowledgment.

We sincerely thank Professor Kaifeng Lyv from Tsinghua University for valuable discussions and insightful comments that helped improve this work. We sincerely thank Haoxin Li from The University of Science and Technology of China for valuable discussions on mathematical details, specifically Lemma G.14 and Remark H.5 in the Appendix.

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

# A. Limitation

First, our reverse-engineering attack currently relies on predefined candidate pools of subjects and prompt templates; future work may develop gradient-based, high-fidelity attack methods that operate without auxiliary data, which could pose a higher safety risk to model editing.

Second, while our camouflage-based defense effectively obscures editing subjects, increasing the protection level inevitably results in the hallucination of decoy subjects. Furthermore, its long-term behavior under large-scale, sequential editing settings remains to be investigated. As edits accumulate, repeatedly applying subspace camouflage may cause gradual performance degradation or distributional drift in the model's latent representations, potentially reducing overall utility. Future work could explore more scalable and theoretically grounded protection mechanisms.

# B. Related Work

### B.1. Security and Privacy Threats in LLMs.

LLMs face a variety of threats that compromise model safety and data privacy (Das et al., 2025). At the security level, prompt injection (Shin et al., 2020) and jailbreaking (Wei et al., 2023) manipulate inputs to override system instructions (Liu et al., 2023; Shi et al., 2024) or bypass safety alignment (Shen et al., 2024; Deng et al., 2023). Meanwhile, backdoor (Shi et al., 2023; Cai et al., 2022) and data poisoning attacks (Chen et al., 2024; Yan et al., 2024) embed malicious triggers during training to elicit harmful behaviors. At the privacy level, unintended memorization enables PII leakage (Nakka et al., 2024; Kim et al., 2023), membership inference (Kaneko et al., 2025; Feng et al., 2025) and gradient leakage (Guo et al., 2021; Balunovic et al., 2022). These risks necessitate efficient data erasure mechanisms. Model editing has thus emerged for its surgical precision in knowledge modification (Hossain & Kagal, 2025; Li et al., 2025).

### B.2. Model Editing Paradigms.

Existing model editing techniques generally fall into three categories. Parameter-preserving methods utilize external modules (Mitchell et al., 2022b; Huang et al., 2023; Hartvigsen et al., 2023; Yu et al., 2024), memory systems (Wang et al., 2024b), or input contexts (Madaan et al., 2022; Zheng et al., 2023) to implement editing, yet suffer from over-parameterization as model scales grow. Meta-learning strategies utilize hypernetworks to predict weight updates (De Cao et al., 2021; Mitchell et al., 2022a; Zhang et al., 2024), though they offer limited interpretability. In contrast, locate-then-edit algorithms offer a more transparent and parameter-efficient alternative. By localizing knowledge-related neurons via causal tracing, this approach implements editing by modifying these specific units. Representative methods include ROME (Meng et al., 2022) for single-fact editing and MEMIT (Meng et al., 2023) for batch editing. To mitigate model collapse, recent works introduce post-hoc regularization techniques (Gu et al., 2024; Ma et al., 2025), and AlphaEdit (Fang et al., 2025) maintains general model capabilities by projecting updates onto the null space of general knowledge. Furthermore, recent studies recognize the limitations of existing editing techniques. Methodologically, Liu et al. (2026a) enhances the editing performance of MEMIT using forward replay techniques. In terms of evaluation, Liu et al. (2026b) proposes a behavior distribution-based evaluation framework to measure editing locality.

### B.3. Leakage Risks in Model Editing.

Although model editing enables efficient knowledge modification, its risks remain unclear. Patil et al. (2024) first explored this issue and proposed a posterior evaluation framework to detect the residuals of erased answers by analyzing intermediate layer logit distributions. However, their attack relies on exact original prompts as triggers, which is a strong assumption that seldom holds in real-world scenarios. Youssef et al. (2025) recovers the pre-edit behavior of the model by analyzing the post-edit model, but they cannot reconstruct the editing prompt and original output. In contrast, our attack focuses on the mathematical fingerprints of parameter updates. It enables the accurate recovery of all editing information without original queries, revealing the inherent side-channel leakage risks in model editing mechanisms.

# C. Extension of Subject Inference Attack

In this section, we provide detailed derivations and discussions on how our subject inference attack framework extends to other locate-then-edit methods, specifically ROME and AlphaEdit.

## C.1. Extension to ROME

ROME treats model editing as a constrained optimization problem, where the goal is to satisfy the hard constraint $\mathbf{W}_{new}\mathbf{k}_* = \mathbf{v}_*$ while preserving general knowledge as much as possible (encoded by the covariance matrix $\mathbf{C}$). Let $\mathbf{r}_* = \mathbf{v}_* - \mathbf{W}_0\mathbf{k}_*$ denote the residual vector. The closed-form solution for the weight update is given by:

$$\Delta\mathbf{W}_{\text{ROME}} = \frac{\mathbf{r}_*\mathbf{k}_*^\top\mathbf{C}^{-1}}{\mathbf{k}_*^\top\mathbf{C}^{-1}\mathbf{k}_*}. \tag{11}$$

From the perspective of our attack, although the optimization objective differs from MEMIT, the geometric property of the update remains exploitable. The update $\Delta\mathbf{W}$ is strictly a rank-one matrix. Crucially, its row space is entirely spanned by the vector $\mathbf{k}_*^\top\mathbf{C}^{-1}$.

Consequently, we can directly employ Algorithm 1 by setting $N = 1$. It is worth noting that in this rank-one scenario, our projection-based scoring function $\rho_i^c$ degenerates into a cosine similarity calculation:

$$\rho_i^c = \frac{\left\|\mathbf{k}_*^\top\mathbf{k}_i^c\right\|_2}{\left\|\mathbf{k}_*\right\|_2\left\|\mathbf{k}_i^c\right\|_2} = \left|\text{CosSim}(\mathbf{k}_i^c, \mathbf{k}_*)\right|. \tag{12}$$

## C.2. Extension to AlphaEdit

Unlike MEMIT, which employs a soft constraint weighted by the covariance matrix $\mathbf{C}$, AlphaEdit constructs a projection matrix $\mathbf{P}$ to preserve the model's general capabilities by projecting the update onto the null space of general knowledge. Its closed-form solution is given by:

$$\Delta\mathbf{W}_{\text{AlphaEdit}} = \mathbf{R}\mathbf{K}^\top\mathbf{P}\left(\mathbf{K}\mathbf{K}^\top\mathbf{P} + \mathbf{I}\right)^{-1}. \tag{13}$$

Next, following the derivations in Appendix G.4, we apply the Woodbury matrix identity to rewrite the AlphaEdit update. The reformulated update is given by:

$$\Delta\mathbf{W}_{\text{AlphaEdit}} = \mathbf{R}\left(\mathbf{I} + \mathbf{K}^\top\mathbf{P}\mathbf{K}\right)^{-1}\mathbf{K}^\top\mathbf{P}. \tag{14}$$

Due to the null-space projection matrix $\mathbf{P}$, we cannot recover some of the information of $\text{ColSpace}(\mathbf{K})$ (see Lemma G.18 and Remark G.19 for details). Thus, we propose a different subject recovery algorithm for AlphaEdit in Algorithm 3, where $\rho_i^c$ represents the projection coefficient of the candidate vector projected by $\mathbf{P}$ onto the subspace $\text{ColSpace}(\mathbf{P}\mathbf{K})$.

---

**Algorithm 3** The KSTER Attack. Stage I: Subject Inference (AlphaEdit)

**Input:** Pre-edit model $\theta$, weight update matrix $\Delta\mathbf{W}$, projection matrix $\mathbf{P}$, subject candidates $\mathcal{S}_{\text{cand}}$, generic template $\mathcal{T}_{\text{gen}}$, activation extraction function $\mathcal{F}_\theta(\cdot)$.

**Output:** Predicted target set $\hat{\mathcal{S}}$.

1   $N \leftarrow \text{Rank}(\Delta\mathbf{W})$.
2   $\mathbf{M} \leftarrow \Delta\mathbf{W}$.
3   $[\mathbf{U}, \mathbf{\Sigma}, \mathbf{V}^\top] \leftarrow$ Singular value decomposition$(\mathbf{M})$.
4   $\mathbf{V}_N \leftarrow \mathbf{V}[:, 1:N]$.
5   $\mathcal{J} \leftarrow \emptyset$.
6   **for** $s_i^c \in \mathcal{S}_{\text{cand}}$ **do**
7     $\mathbf{k}_i'^c \leftarrow \mathbf{P} \cdot \mathcal{F}_\theta(s_i^c, \mathcal{T}_{\text{gen}})$.
8     $\rho_i^c \leftarrow \frac{\left\|\mathbf{V}_N^\top\mathbf{k}_i'^c\right\|_2}{\left\|\mathbf{k}_i'^c\right\|_2}$.
9     $\mathcal{J} \leftarrow \mathcal{J} \cup \{(s_i^c, \rho_i^c)\}$.
10   $\hat{\mathcal{S}} \leftarrow \text{Top-}N(\{s_i^c \mid (s_i^c, \rho_i^c) \in \mathcal{J}\})$.
11   **return** $\hat{\mathcal{S}}$.

---

# D. Extensions of Subspace Camouflage

In this section, we extend our defense strategy *subspace camouflage* to ROME and AlphaEdit by formalizing the constraint problems and providing their solutions. Here we still require the same construction of $\tilde{\mathbf{K}}$ (for ROME, $N = 1$, $\tilde{\mathbf{K}}$ reduces to the vector $\tilde{\mathbf{k}}_*$), while the mathematical forms of defense weight updates $\Delta\mathbf{W}_{\text{defense}}$ are slightly different.

We recall the constraint problem and its solution for MEMIT in §5.1. The constraint problem is formulated by

$$\begin{cases} \text{RowSpace}(\Delta\mathbf{W}_{\text{defense,MEMIT}}\mathbf{C}) & \subseteq \text{ColSpace}(\tilde{\mathbf{K}}), \\ \Delta\mathbf{W}_{\text{defense,MEMIT}}\mathbf{K} & = \Delta\mathbf{W}_{\text{MEMIT}}\mathbf{K}. \end{cases} \tag{15}$$

Its solution is (proven in Theorem H.1, Appendix H)

$$\Delta\mathbf{W}_{\text{defense,MEMIT}} = \Delta\mathbf{W}_{\text{MEMIT}}\mathbf{K} \left( \tilde{\mathbf{K}}^\top\mathbf{C}^{-1}\mathbf{K} \right)^{-1} \tilde{\mathbf{K}}^\top\mathbf{C}^{-1}. \tag{16}$$

## D.1. Extension to ROME

ROME is designed for a single-edit setting ($N = 1$). Following similar arguments in §5.1, now the constraints become:

$$\begin{cases} \text{RowSpace}(\Delta\mathbf{W}_{\text{defense,ROME}}\mathbf{C}) & \subseteq \text{Span}(\{\tilde{\mathbf{k}}_*\}), \\ \Delta\mathbf{W}_{\text{defense,ROME}}\mathbf{k}_* & = \Delta\mathbf{W}_{\text{ROME}}\mathbf{k}_*. \end{cases} \tag{17}$$

Here $\tilde{\mathbf{k}}_*$ is exactly the $N = 1$ case of $\tilde{\mathbf{K}}$.

By solving the constraints above (detailed in Theorem H.1, Appendix H), we obtain the new weight update as:

$$\Delta\mathbf{W}_{\text{defense,ROME}} = \frac{\Delta\mathbf{W}_{\text{ROME}}\mathbf{k}_*\tilde{\mathbf{k}}_*^\top\mathbf{C}^{-1}}{\tilde{\mathbf{k}}_*^\top\mathbf{C}^{-1}\mathbf{k}_*}. \tag{18}$$

## D.2. Extension to AlphaEdit

For AlphaEdit, since Algorithm 3 only reconstructs $\text{ColSpace}(\mathbf{P}\mathbf{K})$, we slightly modify the constraint on the linear subspaces (change the right side from $\text{ColSpace}(\tilde{\mathbf{K}})$ to $\text{ColSpace}(\mathbf{P}\tilde{\mathbf{K}})$), while retaining the consistency of $\Delta\mathbf{W}_{\text{defense,AlphaEdit}}$ and $\Delta\mathbf{W}_{\text{AlphaEdit}}$ on edited subjects $\mathbf{K}$:

$$\begin{cases} \text{RowSpace}(\Delta\mathbf{W}_{\text{defense,AlphaEdit}}) & \subseteq \text{ColSpace}(\mathbf{P}\tilde{\mathbf{K}}), \\ \Delta\mathbf{W}_{\text{defense,AlphaEdit}}\mathbf{K} & = \Delta\mathbf{W}_{\text{AlphaEdit}}\mathbf{K}. \end{cases} \tag{19}$$

By solving the constraints (detailed in Theorem H.1, Appendix H) we obtain

$$\Delta\mathbf{W}_{\text{defense,AlphaEdit}} = \Delta\mathbf{W}_{\text{AlphaEdit}}\mathbf{K} \left( \tilde{\mathbf{K}}^\top\mathbf{P}\mathbf{K} \right)^{-1} \tilde{\mathbf{K}}^\top\mathbf{P}. \tag{20}$$

Similar to MEMIT, in implementation, we apply a small ridge regularization on $\tilde{\mathbf{K}}^\top\mathbf{P}\mathbf{K}$ before inversion to ensure numerical stability.

# E. Implementation Details

## E.1. Detailed Experimental Setup for Attack

This section provides additional details regarding the experimental configurations, library construction, and metric definitions for our attack framework.

**Implementation Details.** For each dataset, we randomly sample $N$ knowledge tuples from the first 2,000 entries. We set $N = 1$ for the single-edit algorithm (ROME) and vary $N \in \{10, 50, 100\}$ for batched editors (MEMIT and AlphaEdit). To ensure statistical significance, all reported results are averaged over five independent runs. Furthermore, we follow the configurations of AlphaEdit (Fang et al., 2025) for the calculation of the covariance matrix $\mathbf{C}$ and the projection matrix $\mathbf{P}$.

**Candidate Library Construction.** To simulate a realistic attack scenario where the attacker does not have exact knowledge of the edited subjects and prompts, we construct the following candidate libraries:

- **Subject Library:** We aggregate the subjects from the first 2,000 examples of the target dataset (CounterFact or zsRE) to form the candidate pool.
- **Prompt Library:** We build a unified candidate pool of 1,000 prompts. This is achieved by aggregating and deduplicating prompts from the first 2,000 examples of both CounterFact and zsRE. Additional diverse prompts are added to reach the fixed size of 1,000.

**Gray-box Baseline Details for Subject Inference.** The gray-box baseline serves to quantify the advantage of accessing internal weights ($\Delta \mathbf{W}$) over observing external behavior. For each candidate subject $s_i^c$, we query both the pre-edit model $\theta$ and the post-edit model $\theta'$ with a generic template. We compute the Jensen–Shannon (JS) divergence between their next-token logit distributions: $D_{JS}(\mathbb{P}_\theta \| \mathbb{P}_{\theta'})$. The $N$ subjects with the highest divergence are identified as the predicted targets. Regarding the recovery of prompt templates, the procedure is identical to stage II of our white-box attack, as it relies solely on output behavior rather than internal parameter computations.

**Naive Method for Prompt Recovery.** The naive method simply observes the logit distribution of the edited model when candidate prompt templates are filled with the current subject; it then selects the prompt template with the minimum entropy (i.e., the sharpest distribution) as the final result.

**Detailed Metric Definitions.**

- **Subject Inference:** Subject recall@N measures the percentage of cases where the ground-truth subject is included in the top-$N$ predicted candidates. The average projection coefficient quantifies the spectral alignment between the subject's key vector and the principal linear subspace derived by parameter update $\Delta \mathbf{W}$.
- **Prompt Recovery:** Top-$N_r$ recall ($N_r \in \{1, 5, 20\}$) evaluates the exact-match success rate. Semantic similarity (Sim.) utilizes Sentence-BERT (SBERT) (Reimers & Gurevych, 2019) to compute the average cosine similarity between the ground-truth prompt and the top-5 recovered candidates, reflecting the recovery of semantic essence.

For all of these metrics, higher values indicate better performance.

Once the subject and the prompt are successfully recovered, the sensitive information can be easily extracted by querying the pre-edit model $\theta$.

### E.2. Detailed Experimental Setup for Defense

This section provides the implementation details and the set of metrics used to evaluate our defense mechanism.

**Implementation Details.** We evaluate the defense on Llama3-8B-Instruct. The defense is integrated into three representative editors: ROME (Meng et al., 2022), MEMIT (Meng et al., 2023), and AlphaEdit (Fang et al., 2025). For MEMIT and AlphaEdit, we adopt a batch editing setting with a batch size of $N = 100$, while for ROME, we use the default single-edit setting ($N = 1$). The strength of the defense is controlled by the camouflage coefficient $\alpha$, where $\alpha = 0$ represents the vanilla, unprotected model. To ensure a rigorous evaluation, we employ our white-box subject inference attack (described in §4) as the attacker. We construct the decoy key matrix by sampling $N$ decoy subjects from the first 2,000 examples for batch editing, whereas for single editing, the decoy key vector is obtained by averaging the key vectors of 10 sampled subjects. All results are the average of 5 independent runs.

**Editing Metrics.** Distinct from previous works (Fang et al., 2025) that often rely on relative probability (e.g., $P(y_{target}) > P(y_{original})$), we implement a stricter exact match standard based on greedy decoding to better reflect real-world editing quality:

- **Efficacy (Eff):** The rate at which the model's greedy generation exactly matches the target output $y_{\text{target}}$.
- **Generalization (Gen):** The exact match rate when the model is queried with semantically equivalent but lexically different rephrased prompts.
- **Specificity (Spe):** The exact match rate with the original ground-truth answer for neighborhood (unrelated) prompts,

ensuring no over-generalization.

- **Fluency (Flu):** Measured by the $n$-gram entropy of generated sequences to monitor for repetitive patterns or linguistic collapse.
- **Consistency (Consis):** The TF-IDF cosine similarity between generated text and a set of reference texts to measure semantic stability.

**Editing Protection Metrics.** To quantify the effectiveness of concealing edited subjects, we utilize the average rank of true edited subjects within the constructed candidate library. During an attack, the attacker ranks all candidates based on their inference scores; a higher average rank for the true subject indicates that it is effectively obscured among decoys, reflecting stronger editing protection.

**Model Utility Metrics.** To evaluate whether the defense mechanism compromises the model's general capabilities, we test the edited models on six standard benchmarks: SST, MMLU, MRPC, CoLA, RTE, and NLI (Hendrycks et al., 2020; Wang et al., 2018). We report the F1 scores across these tasks. A minimal performance drop compared to the pre-edit model indicates that the defense successfully preserves the model's general utility.

For all of these metrics, higher values indicate better performance.

### E.3. Subject and Prompt Template Examples for Figure 2

**Subject List**

```
Danielle Darrieux, Albert Einstein, Marie Curie, Isaac Newton, Galileo Galilei, Charles
Darwin, Nikola Tesla, Leonardo da Vinci, Ada Lovelace, Alan Turing.
```

**Prompt Templates ($\mathcal{T}_{\mathrm{gen}}$)**

**T1:** *Considering their family background and early upbringing, the mother tongue of {} is {}.*

**T2:** *According to historical records regarding their early life, the birthplace of {} is {}.*

**T3:** *Throughout their long and distinguished career, the primary occupation of {} is {}.*

**T4:** *Based on their legal citizenship and country of origin, the nationality of {} is {}.*

**T5:** *Regarding their personal spiritual beliefs and practices, the religion of {} is {}.*

**T6:** *If we look at their specific area of professional expertise, the field of work of {} is {}.*

**T7:** *At the end of their life journey, the recorded place of death of {} is {}.*

**T8:** *During the peak of their professional career, the main employer of {} is {}.*

**T9:** *Regarding their academic background and alma mater, the institution where {} was educated at is {}.*

**T10:** *Among the many accolades and honors bestowed upon them, the award received by {} is {}.*

### E.4. Generic Prompt Template Case

We adopt the first prompt template among the following options[3] as our generic template for all primary experiments. Furthermore, we conduct an ablation study in Table 11 (Appendix F) to evaluate the robustness of our method to the choice of different prompt templates.

**Generic Prompt Template**

```
The mother tongue of {} is {}.
The US dollar {} earns each year is {}.
The detailed address of {} is {}.
We know that {} is {}.
{}, who is {}.
```

---

[3]We did not cherry-pick these prompt templates.

## F. Additional Experiments

**Subject Inference Performance on ROME.** We evaluate the performance of our subject inference attack on ROME in Table 6. Results show that our white-box attack consistently achieves a 100% recall rate, while the logit-based baseline still faces the possibility of attack failure on Qwen2.5-7B-Instruct.

**Prompt Recovery Performance on ROME and AlphaEdit.** Tables 7 and 5 provide supplementary results for the prompt recovery performance of ROME and AlphaEdit, respectively. Consistent with the observations for MEMIT, our attack maintains high recall rates on the CounterFact dataset across all evaluated models. Specifically, for ROME (Table 7), the Top-20 recall for all three models consistently reaches $1.00$, with semantic similarity remaining above $0.84$. For the more complex AlphaEdit (Table 5), the recovery quality remains stable as the number of edits $N$ scales from 10 to 100, where both GPT-J and Qwen2.5-7B-Instruct achieve Top-20 recall rates exceeding $0.95$ on CounterFact. Although performance on zsRE shows a slight decline compared to CounterFact due to the QA format, the semantic similarity scores ($0.77$–$0.89$) confirm that the recovered prompts maintain a high degree of semantic correlation with the original prompts.

**Semantic Ambiguity in case of "Anaal Nathrakh."** Figure 5 illustrates the score ranking of candidate prompts for the subject "Anaal Nathrakh," revealing the phenomenon of *semantic ambiguity* during the editing process. In this case, although the original prompt (`{}, that was created in {}`) is successfully recovered, it ranks only 17th and is masked by several unrelated candidate prompts. The top-five prompts primarily consist of location-related templates, such as `The headquarters of {} is in {}` and `{} was developed in {}`. This indicates that existing editing methods fail to precisely identify the "musical group" entity category, instead treating it as an entity with location attributes (such as a company or software). This exposes the deficiencies of existing editing algorithms, consequently limiting the performance of our prompt recovery attack.

**Editing Performance under Subspace Camouflage.** Tables 8, 9, and 10 evaluate the impact of the camouflage coefficient $\alpha$ across different editing methods and datasets. The results demonstrate that as $\alpha$ increases, the protection strength improves notably, while editing utility remains stable within a specific range. For the ROME method, as shown in Tables 8 and 9, increasing $\alpha$ from 0 to 9 causes the average rank of the target subjects to increase from $1.00$ to $17.20$ (CounterFact) and $30.40$ (zsRE), respectively. Crucially, throughout the scaling process, the editing efficacy remains near $1.00$, and metrics such as fluency and consistency are virtually unaffected. On the more challenging zsRE dataset, MEMIT and AlphaEdit exhibit similar trends (Table 10). When $\alpha$ scales from 0 to 2, the rank increases sharply (e.g., from $47.42$ to $157.23$ for MEMIT). However, when $\alpha$ reaches 4, the efficacy of MEMIT drops to $0.59$, and the specificity of AlphaEdit also begins to decline. This is because an excessively large $\alpha$ increases the $L_2$ norm of the update matrix $\Delta W_{\text{defense}}$, thereby interfering with unrelated input-output mappings.

**Utility-Protection Trade-off under Subspace Camouflage.** Figures 6, 7, and 8 illustrate the trade-off between subject concealment and the model's general capability, confirming that stronger protection generally sacrifices general utility. For the batch edits ($N = 100$) shown in Figures 6 and 7, both MEMIT and AlphaEdit maintain relatively stable utility within a certain range but experience a performance decline at excessive scales. In contrast, the ROME method ($N = 1$) in Figure 8 exhibits remarkable stability. This reveals that batch editing faces a more severe risk of degrading general capability than single-instance editing.

*Table 5.* Performance of prompt recovery attack of AlphaEdit. We report different recall (Top-$N_r$) and semantic similarity (Sim.).

| Model | $N$ | CounterFact | | | | zsRE | | | |
|---|---|---|---|---|---|---|---|---|---|
| | | Top-1 | Top-5 | Top-20 | Sim. | Top-1 | Top-5 | Top-20 | Sim. |
| **GPT-J** | 10 | $0.48 \pm 0.11$ | $0.92 \pm 0.05$ | $0.98 \pm 0.05$ | $0.89 \pm 0.01$ | $0.28 \pm 0.13$ | $0.62 \pm 0.13$ | $0.88 \pm 0.11$ | $0.84 \pm 0.03$ |
| | 50 | $0.47 \pm 0.07$ | $0.88 \pm 0.03$ | $1.00 \pm 0.01$ | $0.89 \pm 0.01$ | $0.33 \pm 0.04$ | $0.67 \pm 0.02$ | $0.88 \pm 0.02$ | $0.83 \pm 0.01$ |
| | 100 | $0.52 \pm 0.05$ | $0.90 \pm 0.02$ | $1.00 \pm 0.00$ | $0.89 \pm 0.01$ | $0.31 \pm 0.03$ | $0.66 \pm 0.04$ | $0.87 \pm 0.04$ | $0.83 \pm 0.01$ |
| **Llama-3 -Instruct** | 10 | $0.54 \pm 0.13$ | $0.88 \pm 0.05$ | $0.94 \pm 0.09$ | $0.88 \pm 0.01$ | $0.34 \pm 0.09$ | $0.68 \pm 0.13$ | $0.84 \pm 0.11$ | $0.84 \pm 0.02$ |
| | 50 | $0.44 \pm 0.06$ | $0.79 \pm 0.04$ | $0.87 \pm 0.05$ | $0.89 \pm 0.02$ | $0.31 \pm 0.08$ | $0.55 \pm 0.05$ | $0.76 \pm 0.04$ | $0.83 \pm 0.01$ |
| | 100 | $0.45 \pm 0.02$ | $0.77 \pm 0.03$ | $0.87 \pm 0.04$ | $0.87 \pm 0.01$ | $0.25 \pm 0.04$ | $0.51 \pm 0.05$ | $0.71 \pm 0.04$ | $0.83 \pm 0.01$ |
| **Qwen-2.5 -Instruct** | 10 | $0.30 \pm 0.19$ | $0.82 \pm 0.13$ | $0.98 \pm 0.05$ | $0.85 \pm 0.02$ | $0.12 \pm 0.11$ | $0.34 \pm 0.24$ | $0.54 \pm 0.32$ | $0.81 \pm 0.04$ |
| | 50 | $0.36 \pm 0.04$ | $0.80 \pm 0.05$ | $0.96 \pm 0.05$ | $0.85 \pm 0.01$ | $0.08 \pm 0.06$ | $0.31 \pm 0.16$ | $0.57 \pm 0.21$ | $0.78 \pm 0.04$ |
| | 100 | $0.43 \pm 0.03$ | $0.85 \pm 0.04$ | $0.96 \pm 0.03$ | $0.85 \pm 0.01$ | $0.07 \pm 0.05$ | $0.21 \pm 0.15$ | $0.42 \pm 0.16$ | $0.77 \pm 0.03$ |

*Table 6.* Performance of subject inference attack for single-edit tasks (ROME).

| Model | CounterFact | | zsRE | |
|---|---|---|---|---|
| | Ours (White-box) | Gray-box | Ours (White-box) | Gray-box |
| GPT-J | **1.00** ± 0.00 | **1.00** ± 0.00 | **1.00** ± 0.00 | **1.00** ± 0.00 |
| Llama-3-Instruct | **1.00** ± 0.00 | **1.00** ± 0.00 | **1.00** ± 0.00 | **1.00** ± 0.00 |
| Qwen-2.5-Instruct | **1.00** ± 0.00 | **1.00** ± 0.00 | **1.00** ± 0.00 | 0.80 ± 0.44 |

*Table 7.* Performance of prompt recovery attack of ROME. We report different recall (Top-$N_r$) and semantic similarity (Sim.).

| Model | CounterFact | | | | zsRE | | | |
|---|---|---|---|---|---|---|---|---|
| | Top-1 | Top-5 | Top-20 | Sim. | Top-1 | Top-5 | Top-20 | Sim. |
| GPT-J | 0.60 ± 0.55 | 0.80 ± 0.45 | 1.00 ± 0.00 | 0.86 ± 0.09 | 0.20 ± 0.45 | 0.40 ± 0.55 | 1.00 ± 0.00 | 0.85 ± 0.06 |
| Llama-3-Instruct | 0.60 ± 0.55 | 0.80 ± 0.45 | 1.00 ± 0.00 | 0.89 ± 0.08 | 0.40 ± 0.55 | 0.60 ± 0.55 | 0.80 ± 0.45 | 0.87 ± 0.04 |
| Qwen-2.5-Instruct | 0.20 ± 0.45 | 0.40 ± 0.55 | 1.00 ± 0.00 | 0.84 ± 0.07 | 0.40 ± 0.55 | 0.40 ± 0.55 | 0.60 ± 0.55 | 0.80 ± 0.05 |

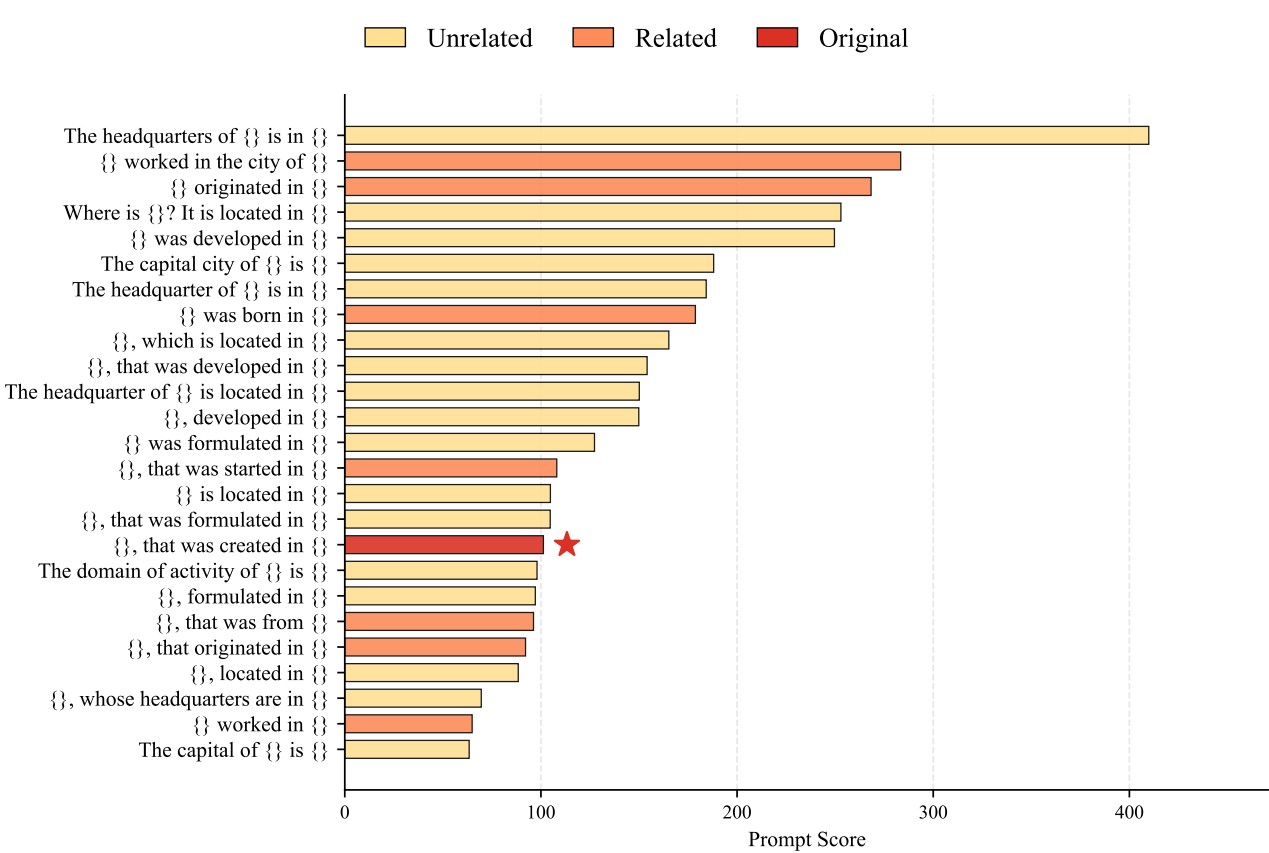

*Figure 5.* Ranking of Candidate Prompts by Score for Subject "Anaal Nathrakh".

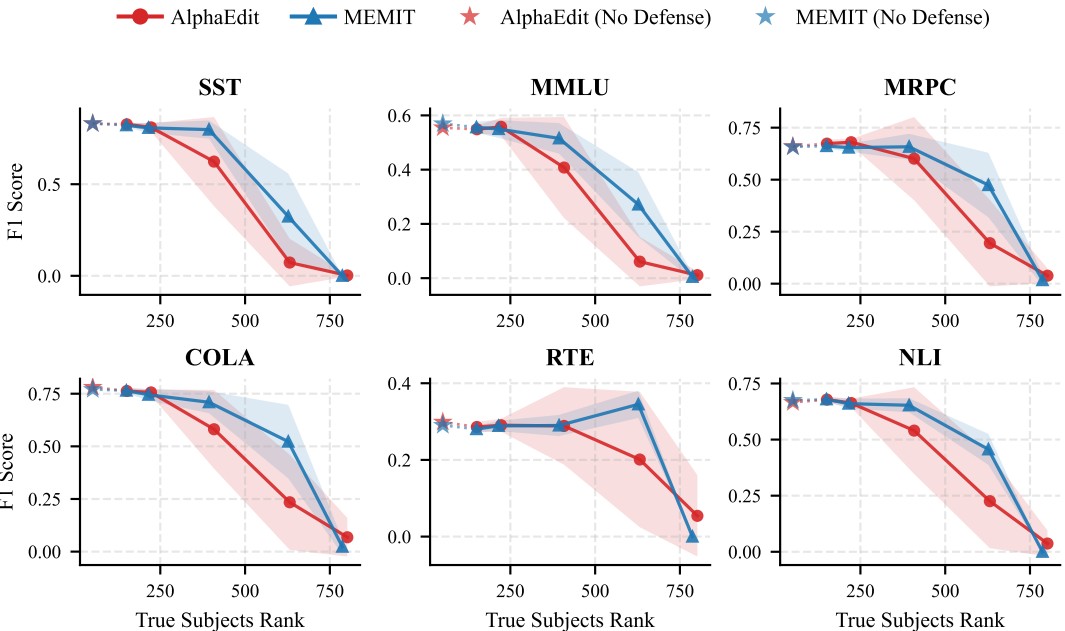

*Figure 6.* Protection-utility trade-off for Llama3-8B-Instruct on CounterFact ($N = 100$). Shaded regions indicate standard deviation over 5 runs.

*Table 8.* Impact of Camouflage Scale ($\alpha$) on ROME Editing Performance (CounterFact).

| $\alpha$ | Rank | Efficacy | Generalization | Specificity | Fluency | Consistency |
|---|---|---|---|---|---|---|
| 0 | $1 \pm 0.00$ | $1.00 \pm 0.00$ | $0.80 \pm 0.24$ | $0.22 \pm 0.39$ | $6.31 \pm 0.10$ | $0.44 \pm 0.16$ |
| 3 | $1.20 \pm 0.40$ | $1.00 \pm 0.00$ | $0.40 \pm 0.37$ | $0.26 \pm 0.35$ | $6.32 \pm 0.14$ | $0.34 \pm 0.12$ |
| 5 | $12.60 \pm 3.72$ | $1.00 \pm 0.00$ | $1.00 \pm 0.00$ | $0.26 \pm 0.14$ | $6.35 \pm 0.08$ | $0.45 \pm 0.12$ |
| 7 | $13.90 \pm 1.35$ | $1.00 \pm 0.00$ | $0.90 \pm 0.20$ | $0.16 \pm 0.14$ | $6.30 \pm 0.16$ | $0.37 \pm 0.07$ |
| 9 | $17.20 \pm 11.41$ | $1.00 \pm 0.00$ | $0.60 \pm 0.49$ | $0.24 \pm 0.32$ | $6.36 \pm 0.12$ | $0.42 \pm 0.11$ |

*Table 9.* Impact of Camouflage Scale ($\alpha$) on ROME Editing Performance (zsRE).

| $\alpha$ | Rank | Efficacy | Generalization | Specificity |
|---|---|---|---|---|
| 0 | $1.00 \pm 0.00$ | $1.00 \pm 0.00$ | $1.00 \pm 0.00$ | $0.5 \pm 0.00$ |
| 3 | $2.00 \pm 1.54$ | $0.93 \pm 0.13$ | $0.87 \pm 0.16$ | $0.46 \pm 0.14$ |
| 5 | $9.20 \pm 2.23$ | $0.97 \pm 0.07$ | $0.90 \pm 0.13$ | $0.46 \pm 0.09$ |
| 7 | $25.30 \pm 29.81$ | $0.93 \pm 0.13$ | $0.93 \pm 0.13$ | $0.30 \pm 0.16$ |
| 9 | $30.40 \pm 20.21$ | $0.97 \pm 0.07$ | $0.97 \pm 0.07$ | $0.25 \pm 0.15$ |

*Table 10.* Performance comparison on zsRE dataset across different camouflage scales.

| Method | $\alpha$ | Rank | Efficacy | Generalization | Specificity |
|---|---|---|---|---|---|
| **MEMIT** | 0 | $47.42 \pm 1.57$ | $0.91 \pm 0.01$ | $0.80 \pm 0.02$ | $0.32 \pm 0.02$ |
| | 1 | $133.87 \pm 1.12$ | $0.91 \pm 0.03$ | $0.81 \pm 0.04$ | $0.32 \pm 0.02$ |
| | 2 | $157.23 \pm 10.54$ | $0.93 \pm 0.01$ | $0.86 \pm 0.03$ | $0.31 \pm 0.01$ |
| | 3 | $162.45 \pm 6.54$ | $0.88 \pm 0.05$ | $0.79 \pm 0.05$ | $0.29 \pm 0.01$ |
| | 4 | $175.71 \pm 6.38$ | $0.59 \pm 0.33$ | $0.51 \pm 0.29$ | $0.21 \pm 0.11$ |
| **AlphaEdit** | 0 | $47.04 \pm 1.82$ | $0.90 \pm 0.03$ | $0.81 \pm 0.03$ | $0.32 \pm 0.03$ |
| | 1 | $132.31 \pm 2.95$ | $0.95 \pm 0.01$ | $0.85 \pm 0.03$ | $0.33 \pm 0.01$ |
| | 2 | $153.18 \pm 6.23$ | $0.96 \pm 0.00$ | $0.89 \pm 0.02$ | $0.34 \pm 0.02$ |
| | 3 | $163.01 \pm 9.66$ | $0.89 \pm 0.10$ | $0.82 \pm 0.10$ | $0.30 \pm 0.07$ |
| | 4 | $176.38 \pm 11.12$ | $0.86 \pm 0.13$ | $0.79 \pm 0.14$ | $0.27 \pm 0.05$ |

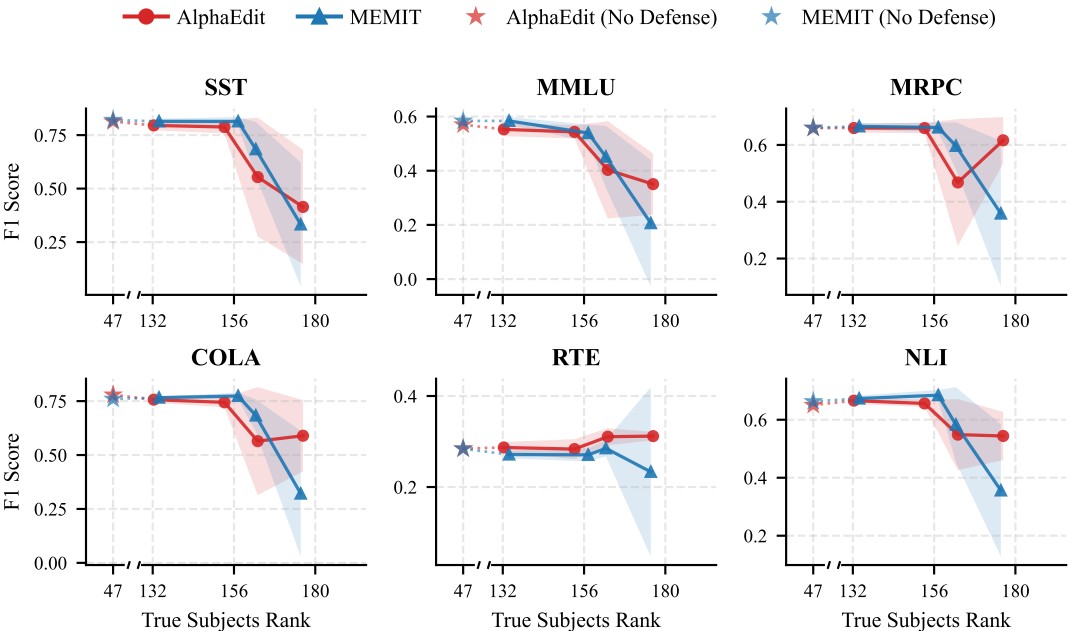

*Figure 7.* Protection-utility trade-off for Llama3-8B-Instruct on zsRE ($N = 100$). Shaded regions indicate standard deviation over 5 runs.

**Impact of Prompt Template on Subject Inference.** Table 11 evaluates the impact of five different prompt templates on subject inference attacks across three models. The results demonstrate that our white-box attack consistently maintains a recall exceeding 0.94 across all templates and datasets. In contrast, the gray-box attack performs poorly and exhibits substantial volatility on Llama3-8B-Instruct and Qwen2.5-7B-Instruct. This indicates that our white-box attack is highly robust to variations in prompt templates.

**Impact of Candidate Subject Pool Size on Subject Inference.** Table 12 illustrates the performance of our subject inference attack across varying candidate subject pool sizes (sampled from the real-world IMDb name dataset). Specifically, across multiple models and dataset settings, as we scale the candidate set from $2 * 10^3$ to $10^7$, the recall of the subject inference drops by no more than 3%. These results further demonstrate the robustness of our subject inference attack.

**Impact of Sequential Editing Number on Subject Inference.** Table 13 illustrates the performance of our subject inference attack in sequential editing tasks (a new model version is released after multiple edits). Specifically, we set the number of sequential edits $s$ from 2 to 10 and report the Top-$(s \times N)$ recall based on the cumulative parameter updates, where $N$ represents the number of facts per batch edit. The results show that our attack consistently achieves a subject recall of over 95% in sequential editing scenarios, which demonstrates the effectiveness of our attack.

**Impact of Candidate Template Pool Size on Prompt Recovery.** Table 14 illustrates the performance of our prompt recovery attack across varying candidate prompt template pool sizes (1000 ground truth templates and 9000 templates generated by Gemini 3.1 Pro). Specifically, across multiple models and dataset settings, as we scale the candidate set from $10^3$ to $10^4$, the Top-20 recall of true templates drops by no more than 2%. These results further demonstrate the robustness of our prompt recovery attack.

**Performance Comparison between Prompt Recovery Baseline and Our Method.** Table 15 illustrates the performance comparison between our prompt recovery attack and the naive method (see Appendix E.1 for details). Specifically, on the CounterFact dataset, our attack outperforms the baseline by more than 13% in terms of Top-1 recall. For Top-5 and Top-20 recall, our method maintains an advantage of approximately 2%. These results further demonstrate the effectiveness of our prompt recovery attack.

**Performance of Subspace Camouflage against Subject Inference Baseline.** Table 16 illustrates the performance of our subspace camouflage in defending against gray-box subject inference attacks (logit-based, see Appendix E.1 for details). Specifically, on the CounterFact dataset, as the camouflage scale increases from 0 to 7, the subject recall drops from 0.66 to 0.00, while the average rank of the ground-truth subject increases from 61 to 934. These results further demonstrate the

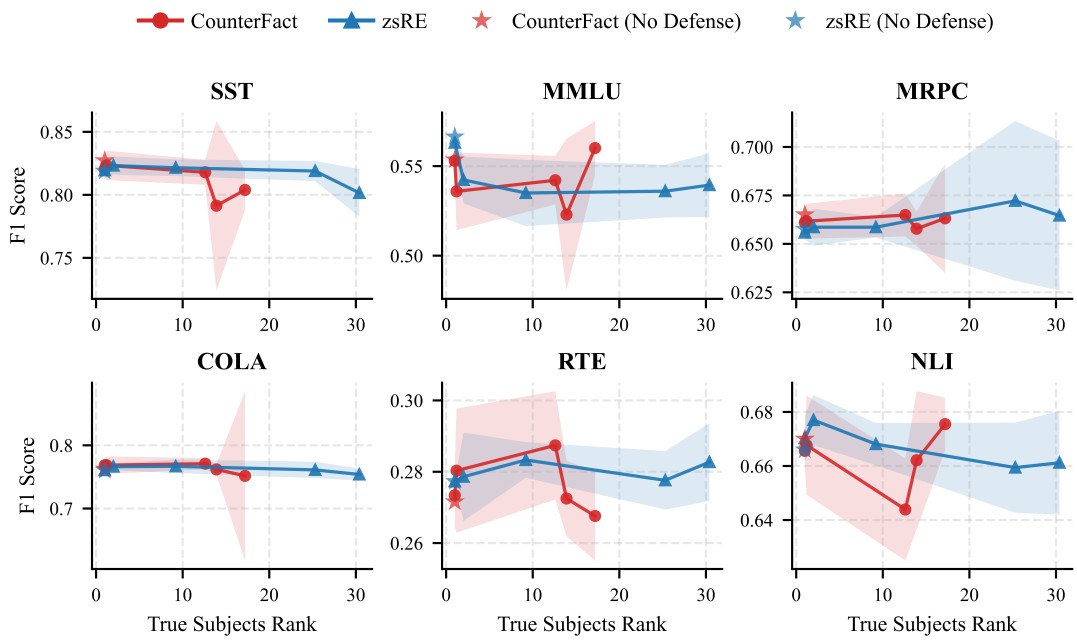

*Figure 8.* Protection-utility trade-off for Llama3-8B-Instruct on ROME ($N = 1$). Shaded regions indicate standard deviation over 5 runs.

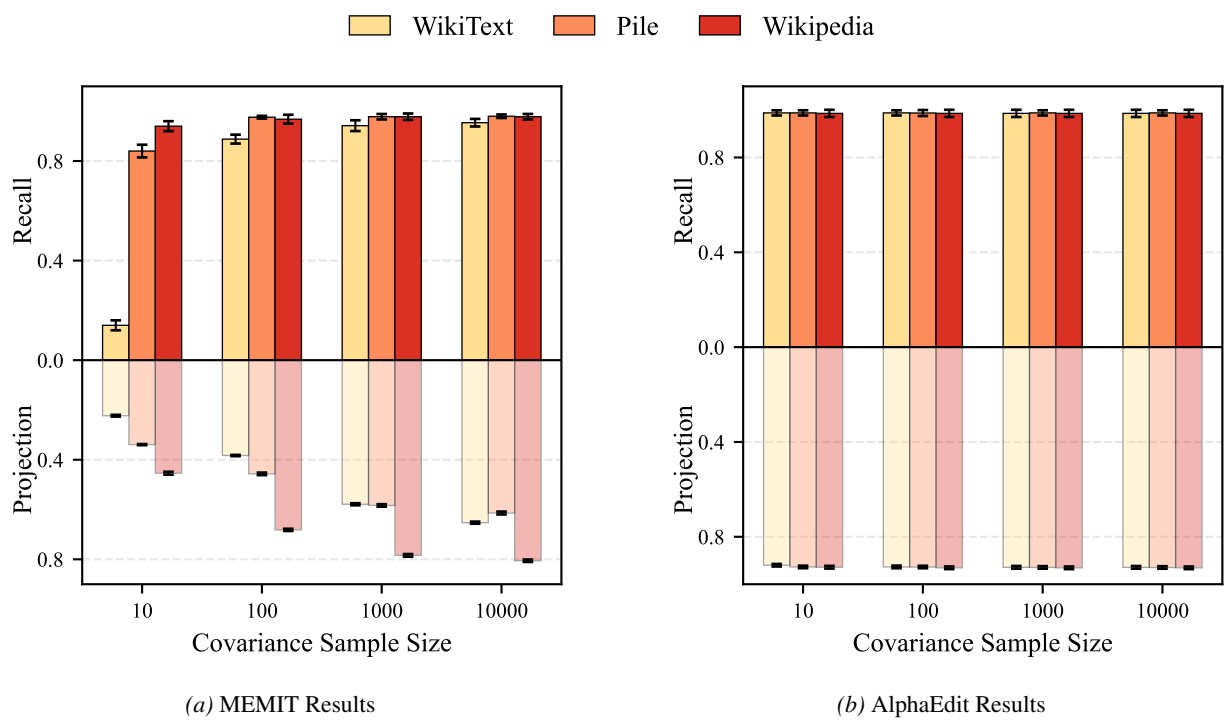

*(a)* MEMIT Results

*(b)* AlphaEdit Results

*Figure 9.* Performance comparison on Llama3-8B-Instruct under different covariance estimations.

*Table 11.* Performance comparison of subject inference attack ($N = 100$) for batch-edit tasks across different prompt templates.

| Model | Template Type | CounterFact | | | | zsRE | | | |
| | | MEMIT | | AlphaEdit | | MEMIT | | AlphaEdit | |
| | | Ours (White-box) | Gray-box | Ours (White-box) | Gray-box | Ours (White-box) | Gray-box | Ours (White-box) | Gray-box |
|---|---|---|---|---|---|---|---|---|---|
| GPT-J | Template 1 | $0.95 \pm 0.01$ | $0.88 \pm 0.03$ | $0.96 \pm 0.01$ | $0.86 \pm 0.04$ | $0.99 \pm 0.01$ | $0.92 \pm 0.02$ | $0.99 \pm 0.01$ | $0.96 \pm 0.01$ |
| | Template 2 | $0.93 \pm 0.02$ | $0.83 \pm 0.02$ | $0.95 \pm 0.02$ | $0.85 \pm 0.03$ | $0.99 \pm 0.01$ | $0.93 \pm 0.01$ | $0.99 \pm 0.01$ | $0.94 \pm 0.01$ |
| | Template 3 | $0.95 \pm 0.01$ | $0.86 \pm 0.03$ | $0.96 \pm 0.01$ | $0.84 \pm 0.02$ | $0.99 \pm 0.01$ | $0.93 \pm 0.01$ | $0.99 \pm 0.01$ | $0.95 \pm 0.03$ |
| | Template 4 | $0.96 \pm 0.02$ | $0.81 \pm 0.02$ | $0.95 \pm 0.02$ | $0.82 \pm 0.03$ | $0.99 \pm 0.01$ | $0.92 \pm 0.02$ | $0.99 \pm 0.01$ | $0.93 \pm 0.02$ |
| | Template 5 | $0.94 \pm 0.02$ | $0.80 \pm 0.03$ | $0.96 \pm 0.02$ | $0.83 \pm 0.02$ | $0.99 \pm 0.01$ | $0.92 \pm 0.02$ | $0.99 \pm 0.01$ | $0.93 \pm 0.01$ |
| Llama-3 -Instruct | Template 1 | $0.99 \pm 0.01$ | $0.68 \pm 0.04$ | $0.99 \pm 0.01$ | $0.45 \pm 0.08$ | $0.99 \pm 0.01$ | $0.69 \pm 0.04$ | $0.99 \pm 0.01$ | $0.43 \pm 0.04$ |
| | Template 2 | $0.98 \pm 0.01$ | $0.64 \pm 0.04$ | $0.99 \pm 0.01$ | $0.48 \pm 0.06$ | $0.99 \pm 0.01$ | $0.63 \pm 0.03$ | $0.99 \pm 0.01$ | $0.47 \pm 0.06$ |
| | Template 3 | $0.99 \pm 0.01$ | $0.62 \pm 0.05$ | $0.99 \pm 0.01$ | $0.52 \pm 0.07$ | $0.99 \pm 0.01$ | $0.66 \pm 0.05$ | $0.99 \pm 0.01$ | $0.48 \pm 0.04$ |
| | Template 4 | $0.99 \pm 0.01$ | $0.61 \pm 0.05$ | $0.98 \pm 0.02$ | $0.49 \pm 0.07$ | $0.98 \pm 0.01$ | $0.62 \pm 0.04$ | $0.98 \pm 0.02$ | $0.52 \pm 0.04$ |
| | Template 5 | $0.98 \pm 0.01$ | $0.59 \pm 0.04$ | $0.98 \pm 0.01$ | $0.53 \pm 0.06$ | $0.99 \pm 0.01$ | $0.64 \pm 0.05$ | $0.99 \pm 0.01$ | $0.49 \pm 0.05$ |
| Qwen-2.5 -Instruct | Template 1 | $0.94 \pm 0.03$ | $0.59 \pm 0.11$ | $0.95 \pm 0.03$ | $0.51 \pm 0.06$ | $0.98 \pm 0.01$ | $0.58 \pm 0.03$ | $0.99 \pm 0.01$ | $0.55 \pm 0.04$ |
| | Template 2 | $0.95 \pm 0.02$ | $0.57 \pm 0.14$ | $0.94 \pm 0.03$ | $0.56 \pm 0.12$ | $0.99 \pm 0.01$ | $0.56 \pm 0.06$ | $0.99 \pm 0.01$ | $0.54 \pm 0.07$ |
| | Template 3 | $0.95 \pm 0.03$ | $0.58 \pm 0.06$ | $0.95 \pm 0.03$ | $0.53 \pm 0.10$ | $0.98 \pm 0.01$ | $0.57 \pm 0.05$ | $0.99 \pm 0.01$ | $0.53 \pm 0.03$ |
| | Template 4 | $0.96 \pm 0.02$ | $0.57 \pm 0.07$ | $0.96 \pm 0.03$ | $0.57 \pm 0.08$ | $0.99 \pm 0.01$ | $0.53 \pm 0.05$ | $0.99 \pm 0.01$ | $0.58 \pm 0.05$ |
| | Template 5 | $0.95 \pm 0.02$ | $0.53 \pm 0.08$ | $0.96 \pm 0.02$ | $0.52 \pm 0.09$ | $0.99 \pm 0.01$ | $0.58 \pm 0.06$ | $0.99 \pm 0.01$ | $0.54 \pm 0.04$ |

*Table 12.* Results of subject inference ($N = 100$) using MEMIT across different subject candidate pool sizes. We report Top-100 recall.

| Dataset | Model | Num=2,000 | Num=10,000 | Num=100,000 | Num=1,000,000 | Num=10,000,000 |
|---|---|---|---|---|---|---|
| CounterFact | Llama-3-Instruct | 0.99 | 0.99 | 0.99 | 0.97 | 0.96 |
| | Qwen-2.5-Instruct | 0.94 | 0.94 | 0.94 | 0.93 | 0.93 |
| | GPT-J | 0.96 | 0.96 | 0.96 | 0.95 | 0.95 |
| zsRE | Llama-3-Instruct | 0.99 | 0.99 | 0.98 | 0.98 | 0.97 |
| | Qwen-2.5-Instruct | 0.98 | 0.98 | 0.97 | 0.96 | 0.96 |
| | GPT-J | 0.99 | 0.99 | 0.99 | 0.97 | 0.97 |

effectiveness of our defense.

**Hallucination of Decoy Subjects in Subspace Camouflage.** We further explore the impact of the camouflage coefficient on the hallucination of decoy subject-related knowledge. As shown in Table 17, on the CounterFact dataset, once $\alpha$ reaches or exceeds 3, a visible intensification of hallucinations regarding decoy subjects occurs. Specifically, the TFR declines from 0.51 to below 0.43, while fluency drops from 6.24 to less than 6.20. This trend reveals a distinct trade-off between the level of protection and the model's general performance.

# G. Theoretical Guarantees for Subject Recovery Algorithms

This section presents the underlying mathematical details and provides theoretical guarantees for the Subject Recovery Algorithms 1 and 3 under an idealized setting.

In this section, for a linear subspace $\mathcal{U} \subseteq \mathbb{R}^d$, we denote $\mathcal{U}_\perp$ as its orthogonal complement space, $\Pi_\mathcal{U}$ as the orthogonal projection matrix from $\mathbb{R}^d$ to $\mathcal{U}$.

## G.1. Preliminaries

This section introduces basic notations, definitions, and necessary assumptions.

We denote $\mathcal{S}_{\mathrm{ed}}$ as the edited subject set consisting of $N$ edited subjects. In the Subject Inference section, one needs to identify the size of the edited subject set $N$, then distinguish $N$ edited subjects $s_i \in \mathcal{S}_{\mathrm{ed}}$ from selected candidate subjects set $\mathcal{S}_{\mathrm{cand}}$. Here we fix the generic template $\mathcal{T}_{\mathrm{gen}}$, and denote $\mathcal{K}_{\mathrm{ed}} := \{\mathbf{k}_i : s_i \in \mathcal{S}_{\mathrm{ed}}\}$ as the key vector set for edited subjects $s_i \in \mathcal{S}_{\mathrm{ed}}$, $\mathcal{K}_{\mathrm{cand}} := \{\mathbf{k}_i^c : s_i^c \in \mathcal{S}_{\mathrm{cand}}\}$ as the key vector set for all candidate subjects, $\mathcal{K}_{\mathrm{non}} := \{\mathbf{k}_i^n : s_i^n \in \mathcal{S}_{\mathrm{non}}\}$ as the key vector set for non-edited subjects $s_i^n \in \mathcal{S}_{\mathrm{non}} := \mathcal{S}_{\mathrm{cand}} - \mathcal{S}_{\mathrm{cand}} \cap \mathcal{S}_{\mathrm{ed}}$. Here, the key vectors are of dimension $d_{\mathrm{in}}$. Notice

*Table 13.* Attack performance on sequential editing across different sequential numbers ($s$) at $N = 100$. We report Top-($s * N$) recall.

| Dataset | Algorithm | Model | s=2 | s=3 | s=4 | s=5 | s=6 | s=7 | s=8 | s=9 | s=10 |
|---|---|---|---|---|---|---|---|---|---|---|---|
| **CounterFact** | MEMIT | GPT-J | 0.97 | 0.97 | 0.97 | 0.97 | 0.97 | 0.97 | 0.97 | 0.97 | 0.97 |
| | | Llama-3-Instruct | 0.99 | 0.99 | 0.98 | 0.98 | 0.98 | 0.98 | 0.98 | 0.98 | 0.98 |
| | | Qwen-2.5-Instruct | 0.94 | 0.94 | 0.95 | 0.95 | 0.95 | 0.95 | 0.95 | 0.95 | 0.95 |
| | AlphaEdit | GPT-J | 0.96 | 0.97 | 0.97 | 0.97 | 0.97 | 0.97 | 0.97 | 0.97 | 0.97 |
| | | Llama-3-Instruct | 0.98 | 0.99 | 0.99 | 0.99 | 0.99 | 0.99 | 0.99 | 0.99 | 0.99 |
| | | Qwen-2.5-Instruct | 0.95 | 0.95 | 0.96 | 0.96 | 0.96 | 0.96 | 0.96 | 0.96 | 0.96 |
| **zsRE** | MEMIT | GPT-J | 0.99 | 1.00 | 1.00 | 1.00 | 1.00 | 1.00 | 1.00 | 1.00 | 1.00 |
| | | Llama-3-Instruct | 0.99 | 0.99 | 1.00 | 1.00 | 1.00 | 1.00 | 1.00 | 1.00 | 1.00 |
| | | Qwen-2.5-Instruct | 0.98 | 1.00 | 1.00 | 1.00 | 1.00 | 1.00 | 1.00 | 1.00 | 1.00 |
| | AlphaEdit | GPT-J | 1.00 | 1.00 | 1.00 | 1.00 | 1.00 | 1.00 | 1.00 | 1.00 | 1.00 |
| | | Llama-3-Instruct | 0.99 | 1.00 | 1.00 | 1.00 | 1.00 | 1.00 | 1.00 | 1.00 | 1.00 |
| | | Qwen-2.5-Instruct | 0.99 | 1.00 | 1.00 | 1.00 | 1.00 | 1.00 | 1.00 | 1.00 | 1.00 |

*Table 14.* Results of prompt recovery attack (Top-20 recall) using MEMIT across different prompt candidate pool sizes in $N = 100$.

| Dataset | Model | Num=1,000 | Num=3,000 | Num=5,000 | Num=7,000 | Num=10,000 |
|---|---|---|---|---|---|---|
| **CounterFact** | Llama-3-Instruct | 0.94 | 0.94 | 0.93 | 0.93 | 0.92 |
| | Qwen-2.5-Instruct | 0.97 | 0.96 | 0.96 | 0.97 | 0.95 |
| | GPT-J | 0.99 | 0.99 | 0.97 | 0.98 | 0.98 |
| **zsRE** | Llama-3-Instruct | 0.76 | 0.78 | 0.75 | 0.73 | 0.72 |
| | Qwen-2.5-Instruct | 0.63 | 0.58 | 0.61 | 0.55 | 0.52 |
| | GPT-J | 0.83 | 0.81 | 0.82 | 0.81 | 0.80 |

that the key matrix $\mathbf{K}$ is constructed by key vectors for edited subjects $\mathcal{S}_{\mathrm{ed}}$ under corresponding edit prompts, not the generic prompt $\mathcal{T}_{\mathrm{gen}}$.

We first assume that the edited subjects are contained in the candidate set and that they are not degenerate:

**Assumption G.1.** The candidate set contains all edited subjects, that is, $\mathcal{S}_{\mathrm{ed}} \subseteq \mathcal{S}_{\mathrm{cand}}$.

A natural corollary of Assumption G.1 is $\mathcal{K}_{\mathrm{cand}} = \mathcal{K}_{\mathrm{ed}} \dot{\cup} \mathcal{K}_{\mathrm{non}}$.

**Assumption G.2.** The key matrix $\mathbf{K} \in \mathbb{R}^{d_{\mathrm{in}} \times N}$, residual matrix $\mathbf{R} \in \mathbb{R}^{d_{\mathrm{out}} \times N}$ are both full column rank matrices:

$$\mathrm{rank}(\mathbf{K}) = \mathrm{rank}(\mathbf{R}) = N. \tag{21}$$

Moreover, given projection matrix $\mathbf{P}$ for AlphaEdit, $\mathrm{rank}(\mathbf{PK}) = N$.

Assumption G.2 indicates that the edited knowledge is not degenerate.

For both MEMIT and AlphaEdit, $\mathrm{rank}(\Delta\mathbf{W}) = N$ is implied by this assumption, see Lemma G.10 for full proof.

*Remark* G.3. Justification for Assumption G.2. Unless *both the prompts and subjects are exactly the same*, it is almost impossible for two (prompt, subject) groups to have the same key vectors/residual vectors. Then due to the high dimension of key/residual vectors (which are $d_{\mathrm{in}}$ and $d_{\mathrm{out}}$ respectively), it is also very unlikely for some key/residual vector to fall into the subspace formed by others, unless number of edited subjects $N$ is close to/larger than $\min(d_{\mathrm{in}}, d_{\mathrm{out}})$.

Then we further define the separation of correct key vectors and other candidate key vectors on a linear subspace:

**Definition G.4.** $\delta\theta$-separability of candidate key vectors.

Given linear subspace $\mathcal{U} \subset \mathbb{R}^{d_{\mathrm{in}}}$, $\mathcal{K}_{\mathrm{ed}}$ and $\mathcal{K}_{\mathrm{non}}$ are said to be $\delta\theta$-separable on $\mathcal{U}$ if for some $\delta\theta \in (0, \frac{\pi}{2})$,

$$\arccos\left(\frac{\|\Pi_{\mathcal{U}}\mathbf{k}_2\|_2}{\|\mathbf{k}_2\|_2}\right) - \arccos\left(\frac{\|\Pi_{\mathcal{U}}\mathbf{k}_1\|_2}{\|\mathbf{k}_1\|_2}\right) \geq \delta\theta, \ \forall \mathbf{k}_1 \in \mathcal{K}_{\mathrm{ed}}, \mathbf{k}_2 \in \mathcal{K}_{\mathrm{non}}. \tag{22}$$

Table 15. Performance comparison of prompt recovery attack between our method and baseline using MEMIT in $N = 100$.

| Dataset | Model | Method | Top-1 | Top-5 | Top-20 | Sim. |
|---|---|---|---|---|---|---|
| **CounterFact** | Llama-3-Instruct | Ours | **0.51** | **0.81** | **0.94** | **0.88** |
| | | Baseline | 0.35 | 0.79 | 0.92 | 0.87 |
| | Qwen-2.5-Instruct | Ours | **0.47** | **0.86** | **0.97** | **0.86** |
| | | Baseline | 0.31 | 0.84 | 0.94 | 0.84 |
| | GPT-J | Ours | **0.50** | **0.92** | **0.99** | **0.88** |
| | | Baseline | 0.37 | 0.88 | **0.99** | 0.86 |
| **zsRE** | Llama-3-Instruct | Ours | **0.33** | **0.57** | **0.76** | **0.83** |
| | | Baseline | 0.24 | 0.56 | 0.72 | 0.82 |
| | Qwen-2.5-Instruct | Ours | **0.08** | **0.29** | **0.55** | **0.78** |
| | | Baseline | 0.02 | 0.26 | 0.51 | 0.75 |
| | GPT-J | Ours | **0.32** | **0.61** | **0.83** | **0.82** |
| | | Baseline | 0.20 | 0.59 | 0.82 | 0.81 |

Table 16. Protection Performance against logit-based baseline (Llama3-8B-Instruct, MEMIT) at $N = 100$.

| Camouflage scale | CounterFact | | zsRE | |
|---|---|---|---|---|
| | **Recall** | **Mean rank** | **Recall** | **Mean rank** |
| 0 | 0.66 | 61 | 0.69 | 57 |
| 1 | 0.57 | 86 | 0.52 | 93 |
| 3 | 0.42 | 153 | 0.38 | 162 |
| 5 | 0.03 | 324 | 0.07 | 314 |
| 7 | 0.00 | 934 | 0.02 | 798 |

Here, $\arccos\left(\frac{\|\Pi_{\mathcal{U}}\mathbf{k}\|_2}{\|\mathbf{k}_{(\cdot)}\|_2}\right)$ is exactly the angle between vector $\mathbf{k}_{(\cdot)}$ and linear subspace $\mathcal{U}$.

Intuitively, $\mathcal{U}$ should be a linear subspace where edited subjects $\mathbf{k}_1 \in \mathcal{K}_{\text{ed}}$ approximately lies in (have large projection norm), while other candidates $\mathbf{k}_2 \in \mathcal{K}_{\text{non}}$ lie more in its orthogonal subspace (have small projection norm on $\mathcal{U}$). We observe such phenomena on both $\mathcal{U}_1 = \text{ColSpace}(\mathbf{K})$ and $\mathcal{U}_2 = \text{ColSpace}(\mathbf{PK})$, which are formalized into Assumptions G.5 and G.6 below:

**Assumption G.5.** Separation robustness of template.

Under the generic template $\mathcal{T}_{\text{gen}}$, $\mathcal{K}_{\text{ed}}$ and $\mathcal{K}_{\text{non}}$ are $\delta\theta$-separable on $\text{ColSpace}(\mathbf{K})$ for some $\delta\theta \in (0, \frac{\pi}{2})$.

**Assumption G.6.** Separation robustness of template, with projection matrix $\mathbf{P}$.

Under the generic template $\mathcal{T}_{\text{gen}}$, $\mathbf{P} \circ \mathcal{K}_{\text{ed}}$ and $\mathbf{P} \circ \mathcal{K}_{\text{non}}$ (here $\mathbf{P} \circ \mathcal{K}_{(\cdot)}$ means $\{\mathbf{Pk}_{(\cdot)} : \mathbf{k}_{(\cdot)} \in \mathcal{K}_{(\cdot)}\}$) are $\delta\theta$-separable on $\text{ColSpace}(\mathbf{PK})$ for some $\delta\theta \in (0, \frac{\pi}{2})$.

*Remark* G.7. Discussion on the generic template. If the template is the same as the one used in model editing, this assumption simply degenerates into requiring the other non-edited candidates not to lie in the linear subspace spanned by the edited subjects' key vectors. Based on the observation of Figure 2b, this assumption extracts the fact that even for the generic template, the key vectors for edited subjects and other candidate subjects are still separable.

To assure the validity of Eq. (4) for MEMIT, we need to further make the following assumption that the covariance matrix $\mathbf{C}$ is not degenerate (and thus invertible):

**Assumption G.8.** The covariance matrix $\mathbf{C}$ is positive definite: $\mathbf{C} \succ \mathbf{O}$.

*Table 17.* Impact of camouflage scale on True Fact Recall (TFR.) and Fluency on Llama-3-Instruct in $N = 100$. (Fluency measures the naturalness of generated text, while True Fact Recall measures the accuracy of the edited model in outputting the target ground truth when prompted with decoy subjects via rewrite prompt and paraphrase prompt.)

| Method | $\alpha$ | CounterFact | | zsRE | |
|---|---|---|---|---|---|
| | | **TFR.** | **Fluency** | **TFR.** | **Fluency** |
| | 0.0 | 0.51 | 6.24 | 0.53 | 6.23 |
| | 1.0 | 0.51 | 6.22 | 0.53 | 6.22 |
| MEMIT | 3.0 | 0.43 | 6.20 | 0.45 | 6.18 |
| | 5.0 | 0.36 | 6.19 | 0.38 | 6.17 |
| | 7.0 | 0.23 | 6.11 | 0.25 | 6.14 |
| | 0.0 | 0.52 | 6.18 | 0.54 | 6.21 |
| | 1.0 | 0.51 | 6.21 | 0.53 | 6.22 |
| AlphaEdit | 3.0 | 0.42 | 6.19 | 0.44 | 6.19 |
| | 5.0 | 0.22 | 6.05 | 0.24 | 6.16 |
| | 7.0 | 0.13 | 5.96 | 0.15 | 5.98 |

### G.2. Subject recovery for MEMIT, Algorithm 1

Before simplifying the update, we introduce the Woodbury matrix identity, which describes the inverse of a rank-$N$ correction to a matrix. The detailed derivation can be found in Eq. (24).

**Lemma G.9** (Woodbury matrix identity). *For an invertible matrix $\mathbf{A} \in \mathbb{R}^{d \times d}$, for all $\mathbf{U} \in \mathbb{R}^{d \times r}$ and $\mathbf{V} \in \mathbb{R}^{r \times d}$ such that $\mathbf{I}_r + \mathbf{VA}^{-1}\mathbf{U}$ is invertible, the following equality holds:*

$$(\mathbf{A} + \mathbf{UV})^{-1} = \mathbf{A}^{-1} - \mathbf{A}^{-1}\mathbf{U}(\mathbf{I}_r + \mathbf{VA}^{-1}\mathbf{U})^{-1}\mathbf{VA}^{-1}. \tag{23}$$

Now under Assumption G.8 we can rewrite the expression of $\Delta\mathbf{W}_{\mathrm{MEMIT}}$ through Lemma G.9:

$$
\begin{aligned}
\Delta\mathbf{W} := \Delta\mathbf{W}_{\mathrm{MEMIT}} &= \mathbf{RK}^\top \left(\mathbf{C} + \mathbf{KK}^\top\right)^{-1} \\
&= \mathbf{RK}^\top \left[\mathbf{C}^{-1} - \mathbf{C}^{-1}\mathbf{K}\left(\mathbf{I} + \mathbf{K}^\top\mathbf{C}^{-1}\mathbf{K}\right)^{-1}\mathbf{K}^\top\mathbf{C}^{-1}\right] \\
&= \mathbf{R}\left[\mathbf{K}^\top\mathbf{C}^{-1} - \mathbf{K}^\top\mathbf{C}^{-1}\mathbf{K}\left(\mathbf{I} + \mathbf{K}^\top\mathbf{C}^{-1}\mathbf{K}\right)^{-1}\mathbf{K}^\top\mathbf{C}^{-1}\right] \\
&= \mathbf{R}\left[(\mathbf{I} + \mathbf{K}^\top\mathbf{C}^{-1}\mathbf{K}) - \mathbf{K}^\top\mathbf{C}^{-1}\mathbf{K}\right]\left(\mathbf{I} + \mathbf{K}^\top\mathbf{C}^{-1}\mathbf{K}\right)^{-1}\mathbf{K}^\top\mathbf{C}^{-1} \\
&= \mathbf{R}\left(\mathbf{I} + \mathbf{K}^\top\mathbf{C}^{-1}\mathbf{K}\right)^{-1}\mathbf{K}^\top\mathbf{C}^{-1}.
\end{aligned} \tag{24}
$$

We also introduce a basic linear algebra lemma for the recovery of the number of edited knowledge $N$:

**Lemma G.10.** *Given three arbitrary matrices $\mathbf{M}_1 \in \mathbb{R}^{d_{\mathrm{out}} \times N}$, $\mathbf{M}_2 \in \mathbb{R}^{d_{\mathrm{in}} \times N}$, $\mathbf{M}_3 \in \mathbb{R}^{d_{\mathrm{in}} \times d_{\mathrm{in}}}$, where $d_{\mathrm{out}}, d_{\mathrm{in}} \geq N$, $\mathbf{M}_1, \mathbf{M}_2$ are full column rank and $\mathbf{M}_3$ is invertible, then $\mathrm{rank}\left(\mathbf{M}_1\mathbf{M}_2^\top\mathbf{M}_3^{-1}\right) = N$.*

*Remark* G.11. By setting $\mathbf{M}_1 = \mathbf{R}, \mathbf{M}_2 = \mathbf{K}, \mathbf{M}_3 = \mathbf{C} + \mathbf{KK}^\top$ (for MEMIT), or $\mathbf{M}_1 = \mathbf{R}, \mathbf{M}_2 = \mathbf{PK}, \mathbf{M}_3 = \mathbf{I} + \mathbf{KK}^\top\mathbf{P}$ (for AlphaEdit), we know that under Assumption G.2 (key matrix and residual matrix are both full column rank), the KSTER attack identifies the correct number of edited knowledge $N$ in Algorithm 1 (MEMIT) and 3 (AlphaEdit).

*Proof.* This result can be proved in two distinct ways:

**1. Linear Transformation approach.** Since $\mathbf{M}_3$ is invertible, $\mathrm{Im}\left(\mathbf{M}_3^{-1}\right) = \mathbb{R}^{d_{\mathrm{in}}}$. Since $\mathbf{M}_2$ is full column rank, $\mathbf{M}_2^\top$ is surjective onto $\mathbb{R}^N$. Thus the image space of $\mathbf{M}_1\mathbf{M}_2^\top\mathbf{M}_3^{-1}$ is:

$$\mathrm{Im}\left(\mathbf{M}_1\mathbf{M}_2^\top\mathbf{M}_3^{-1}\right) = \mathbf{M}_1\left(\mathbf{M}_2^\top\left(\mathrm{Im}\left(\mathbf{M}_3^{-1}\right)\right)\right) = \mathbf{M}_1\left(\mathbf{M}_2^\top\left(\mathbb{R}^{d_{\mathrm{in}}}\right)\right) = \mathbf{M}_1\left(\mathbb{R}^N\right) = \mathrm{ColSpace}(\mathbf{M}_1), \tag{25}$$

giving

$$\text{rank}\left(\mathbf{M}_1\mathbf{M}_2^\top \mathbf{M}_3^{-1}\right) = \dim\left(\text{ColSpace}(\mathbf{M}_1)\right) = N. \tag{26}$$

**2. SVD decomposition approach.** Since $\mathbf{M}_3$ is invertible, $\text{rank}\left(\mathbf{M}_1\mathbf{M}_2^\top \mathbf{M}_3^{-1}\right) = \text{rank}\left(\mathbf{M}_1\mathbf{M}_2^\top\right)$. Suppose the singular value decompositions of $\mathbf{M}_1, \mathbf{M}_2$ are $\mathbf{M}_1 = \mathbf{U}_{\mathbf{M}_1}\boldsymbol{\Sigma}_{\mathbf{M}_1}\mathbf{V}_{\mathbf{M}_1}^\top$ and $\mathbf{M}_2 = \mathbf{U}_{\mathbf{M}_2}\boldsymbol{\Sigma}_{\mathbf{M}_2}\mathbf{V}_{\mathbf{M}_2}^\top$ respectively, where $\mathbf{U}_{\mathbf{M}_1} \in \mathbb{R}^{d_{\text{out}} \times N}, \mathbf{U}_{\mathbf{M}_2} \in \mathbb{R}^{d_{\text{in}} \times N}$ are column-orthogonal matrices, $\mathbf{V}_{\mathbf{M}_1}, \mathbf{V}_{\mathbf{M}_2} \in \mathbb{R}^{N \times N}$ are orthogonal matrices, and $\boldsymbol{\Sigma}_{\mathbf{M}_1}, \boldsymbol{\Sigma}_{\mathbf{M}_2} \in \mathbb{R}^{N \times N}$ are diagonal matrices with positive entries. Thus $\mathbf{V}_{\mathbf{M}_1}^\top \mathbf{V}_{\mathbf{M}_2}$ is also an orthogonal matrix, giving

$$\text{rank}\left(\mathbf{M}_1\mathbf{M}_2^\top\right) = \text{rank}\left(\mathbf{U}_{\mathbf{M}_1}\boldsymbol{\Sigma}_{\mathbf{M}_1}\mathbf{V}_{\mathbf{M}_1}^\top \mathbf{V}_{\mathbf{M}_2}\boldsymbol{\Sigma}_{\mathbf{M}_2}\mathbf{U}_{\mathbf{M}_2}^\top\right) = \text{rank}\left(\boldsymbol{\Sigma}_{\mathbf{M}_1}\mathbf{V}_{\mathbf{M}_1}^\top \mathbf{V}_{\mathbf{M}_2}\boldsymbol{\Sigma}_{\mathbf{M}_2}\right) = N. \tag{27}$$

This completes the proof.

$\square$

**Theorem G.12.** *Recovery of linear subspace for MEMIT.*

*Under Assumptions G.2 and G.8, given update matrix $\Delta\mathbf{W} = \Delta\mathbf{W}_{\text{MEMIT}}$ and positive definite covariance matrix $\mathbf{C}$, the column space of $\mathbf{K}$ is exactly the column space of $\mathbf{V}_N$ ($\mathbf{V}_N$ is defined in Algorithm 1):*

$$\text{ColSpace}(\mathbf{V}_N) = \text{ColSpace}(\mathbf{K}). \tag{28}$$

*Proof.* From Assumption G.8, both $\mathbf{C}$ and $\mathbf{C} + \mathbf{KK}^\top$ are invertible, then $\text{rank}(\mathbf{M}) = \text{rank}(\Delta\mathbf{W}) = \text{rank}(\mathbf{RK}^\top)$. Thus, from Assumption G.2, $\text{rank}(\mathbf{M})$ is exactly the number of edited subjects. Thus, denote $\mathbf{U}_N$ as the first $N$ columns of $\mathbf{U}$, $\boldsymbol{\Sigma}_N$ as the $N \times N$ truncated $\boldsymbol{\Sigma}$, we have $\mathbf{M} = \mathbf{U}_N\boldsymbol{\Sigma}_N\mathbf{V}_N^\top$.

From equation (24), $\Delta\mathbf{W} = \mathbf{R}\left(\mathbf{I} + \mathbf{K}^\top\mathbf{C}^{-1}\mathbf{K}\right)^{-1}\left(\mathbf{C}^{-1}\mathbf{K}\right)^\top$, thus

$$\mathbf{M} := \Delta\mathbf{W}\mathbf{C} = \mathbf{R}\left(\mathbf{I} + \mathbf{K}^\top\mathbf{C}^{-1}\mathbf{K}\right)^{-1}\mathbf{K}^\top. \tag{29}$$

Then from the singular value decomposition $\mathbf{M} = \mathbf{U}_N\boldsymbol{\Sigma}_N\mathbf{V}_N^\top$,

$$\mathbf{V}_N = \mathbf{M}^\top\mathbf{U}_N\boldsymbol{\Sigma}_N^{-1} = \mathbf{K}\left(\mathbf{I} + \mathbf{K}^\top\mathbf{C}^{-1}\mathbf{K}\right)^{-1}\mathbf{R}^\top\mathbf{U}_N\boldsymbol{\Sigma}_N^{-1}. \tag{30}$$

Since $\left(\mathbf{I} + \mathbf{K}^\top\mathbf{C}^{-1}\mathbf{K}\right)^{-1}\mathbf{R}^\top\mathbf{U}_N\boldsymbol{\Sigma}_N^{-1}$ is of rank $N$ and thus invertible, $\text{ColSpace}(\mathbf{V}_N) = \text{ColSpace}(\mathbf{K})$, which completes the proof.

$\square$

**Theorem G.13.** *Guarantee of subject recovery for MEMIT.*

*Under Assumptions G.1, G.2, G.5, G.8, given the update matrix $\Delta\mathbf{W} = \Delta\mathbf{W}_{\text{MEMIT}}$ and covariance matrix $\mathbf{C}$, Algorithm 1 exactly outputs the $N$ correct edited subjects, with the top $N^{th}$ and $(N+1)^{th}$ score (namely $\rho_{(N)}^{c\downarrow}$ and $\rho_{(N+1)}^{c\downarrow}$, here the round brackets mean that the scores are reordered in descending order) separated by:*

$$\arccos\rho_{(N+1)}^{c\downarrow} - \arccos\rho_{(N)}^{c\downarrow} \geq \delta\theta. \tag{31}$$

*Proof.* From the definition of $\rho_i^c$, using the fact that $\mathbf{V}_N$ is a column orthogonal matrix,

$$\rho_i^c := \frac{\left\|\mathbf{V}_N^\top\mathbf{k}_i^c\right\|_2}{\left\|\mathbf{k}_i^c\right\|_2} = \frac{\left\|\mathbf{V}_N\mathbf{V}_N^\top\mathbf{k}_i^c\right\|_2}{\left\|\mathbf{k}_i^c\right\|_2} = \frac{\left\|\Pi_{\text{ColSpace}(\mathbf{V}_N)}\mathbf{k}_i^c\right\|_2}{\left\|\mathbf{k}_i^c\right\|_2}. \tag{32}$$

By Theorem G.12, $\text{ColSpace}(\mathbf{V}_N) = \text{ColSpace}(\mathbf{K})$. Then from Assumption G.5,

$$\arccos \rho_j^c - \arccos \rho_i^c \geq \delta\theta, \ \forall i, j \text{ s.t. } \mathbf{k}_i^c \in \mathcal{K}_{\text{ed}}, \mathbf{k}_j^c \in \mathcal{K}_{\text{non}}, \tag{33}$$

which is equivalent to $s_i^c \in \mathcal{S}_{\text{ed}}, s_j^c \in \mathcal{S}_{\text{non}}$.

From Assumption G.1, $\mathcal{S}_{\text{ed}} \subseteq \mathcal{S}_{\text{cand}}$, then the top $N$ scores $\rho_{(1,\cdots,N)}^{c\downarrow}$ all correspond to edited subjects, while the others correspond to non-edited subjects, giving

$$\arccos \rho_{(N+1)}^{c\downarrow} - \arccos \rho_{(N)}^{c\downarrow} \geq \delta\theta. \tag{34}$$

This completes the proof.

$\square$

### G.3. Robustness on perturbation of covariance matrix C, for MEMIT

This section studies the setting where the covariance matrix $\mathbf{C}$ used in the *KSTER* attack is not exactly available. Specifically, $\mathbf{C}$ may only be estimated by a finite number of samples $N_{\mathbf{C}}$, or it may be replaced by a covariance matrix estimated from another distribution (a different dataset). We uniformly denote the new covariance matrix as $\mathbf{C}'$, now $\mathbf{M}' = \Delta\mathbf{W}\mathbf{C}'$.

Following standard definitions, we define the maximum principal angle of two linear subspaces $\mathcal{U}_{1,2}$ of same dimension as $\theta_{\max}(\mathcal{U}_1, \mathcal{U}_2) := \arccos(\min_{\mathbf{u} \in \mathcal{U}_1, \|\mathbf{u}\|_2 = 1}(\|\Pi_{\mathcal{U}_2}\mathbf{u}\|_2))$, and the angle between a non-zero vector $\mathbf{v} \neq \mathbf{0}$ and a linear subspace $\mathcal{U}$ to be $\phi(\mathbf{v}, \mathcal{U}) := \arccos\left(\frac{\|\Pi_{\mathcal{U}}\mathbf{v}\|_2}{\|\mathbf{v}\|_2}\right)$.

**Lemma G.14.** *Given the angle between vector $k$ and linear subspace $\mathcal{U}_{1,2}$ ($\mathcal{U}_{1,2}$ have same dimension) to be $\phi_{1,2}$ respectively, the maximum principal angle between $\mathcal{U}_1$ and $\mathcal{U}_2$ to be $\theta$, then $\phi_2$ is bounded by:*

$$\max(\phi_1 - \theta, 0) \leq \phi_2 \leq \min(\phi_1 + \theta, \pi/2). \tag{35}$$

*Proof.* This is directly from the triangle inequality of angles in Euclidean space.

$\square$

**Lemma G.15.** *For linear subspaces $\mathcal{U}_{1,2} \subseteq \mathbb{R}^{d_{\text{in}}}$ of same dimension, with $\theta_{\max}(\mathcal{U}_1, \mathcal{U}_2) < \frac{\delta\theta}{2}$, if $\mathcal{K}_{\text{ed}}$ and $\mathcal{K}_{\text{non}}$ are $\delta\theta$-separable on $\mathcal{U}_1$, then they are $\delta\theta'$-separable on $\mathcal{U}_2$, where $\delta\theta' = \delta\theta - 2\theta_{\max}(\mathcal{U}_1, \mathcal{U}_2)$.*

*Proof.* $\forall \mathbf{k}_1 \in \mathcal{K}_{\text{ed}}$ and $\mathbf{k}_2 \in \mathcal{K}_{\text{non}}$, from Definition G.4, $\phi(\mathbf{k}_2, \mathcal{U}_1) - \phi(\mathbf{k}_1, \mathcal{U}_1) \geq \delta\theta$.

From Lemma G.14,

$$\phi(\mathbf{k}_1, \mathcal{U}_2) \leq \phi(\mathbf{k}_1, \mathcal{U}_1) + \theta_{\max}(\mathcal{U}_1, \mathcal{U}_2), \ \phi(\mathbf{k}_2, \mathcal{U}_2) \geq \phi(\mathbf{k}_2, \mathcal{U}_1) - \theta_{\max}(\mathcal{U}_1, \mathcal{U}_2). \tag{36}$$

Thus

$$\phi(\mathbf{k}_2, \mathcal{U}_2) - \phi(\mathbf{k}_1, \mathcal{U}_2) \geq \delta\theta - 2\theta_{\max}(\mathcal{U}_1, \mathcal{U}_2) = \delta\theta'. \tag{37}$$

This completes the proof.

$\square$

**Theorem G.16.** *Subject recovery under noisy covariance estimation.*

*Under Assumptions G.1, G.2, G.5, G.8, given the update matrix $\Delta\mathbf{W}$ and noisy estimation of covariance matrix $\mathbf{C}' = \Delta\mathbf{C} + \mathbf{C} \succ \mathbf{O}$, if $\|\Delta\mathbf{C}\|_2 < \frac{\sin\left(\frac{\delta\theta}{2}\right)}{\|\mathbf{C}^{-1}\mathbf{V}_N\|_2}$, Algorithm 1 exactly outputs the $N$ correct edited subjects, with the top $N^{th}$ and $(N+1)^{th}$ score (namely $\rho_{(N)}^{c\downarrow}$ and $\rho_{(N+1)}^{c\downarrow}$) are separated by:*

$$\arccos \rho^{c\downarrow}_{(N+1)} - \arccos \rho^{c\downarrow}_{(N)} \geq \delta\theta - 2\arcsin\left(\|\Delta\mathbf{C}\|_2 \left\|\mathbf{C}^{-1}\mathbf{V}_N\right\|_2\right). \tag{38}$$

*Proof.* Similar to the proof of Theorem G.12,

$$\mathbf{V}'_N = \mathbf{M}'^\top \mathbf{U}'_N \mathbf{\Sigma}'^{-1}_N = \mathbf{C}'\mathbf{C}^{-1}\mathbf{M}^\top \mathbf{U}'_N \mathbf{\Sigma}'^{-1}_N = \mathbf{C}'\mathbf{C}^{-1}\mathbf{K}\left(\mathbf{I} + \mathbf{K}^\top \mathbf{C}^{-1}\mathbf{K}\right)^{-1}\mathbf{R}^\top \mathbf{U}'_N \mathbf{\Sigma}'^{-1}_N, \tag{39}$$

giving

$$\mathrm{ColSpace}(\mathbf{V}'_N) = \mathrm{ColSpace}(\mathbf{C}'\mathbf{C}^{-1}\mathbf{K}) = \mathrm{ColSpace}(\mathbf{C}'\mathbf{C}^{-1}\mathbf{V}_N). \tag{40}$$

From Wedin's sin Theorem (Theorem 4.4, Chapter 5, Stewart & guang Sun (1990)),

$$\sin(\theta_{\max}(\mathrm{ColSpace}(\mathbf{C}'\mathbf{C}^{-1}\mathbf{V}_N), \mathrm{ColSpace}(\mathbf{V}_N))) \leq \frac{\left\|\Delta\mathbf{C}\mathbf{C}^{-1}\mathbf{V}_N\right\|_2}{\sigma_{\min}(\mathbf{V}_N)} = \left\|\Delta\mathbf{C}\mathbf{C}^{-1}\mathbf{V}_N\right\|_2 \leq \|\Delta\mathbf{C}\|_2\left\|\mathbf{C}^{-1}\mathbf{V}_N\right\|_2. \tag{41}$$

From Lemma G.15, for $\delta\theta' = \delta\theta - 2\arcsin\left(\|\Delta\mathbf{C}\|_2\left\|\mathbf{C}^{-1}\mathbf{V}_N\right\|_2\right) > 0$,

$$\arccos \rho^c_j - \arccos \rho^c_i \geq \delta\theta', \forall i, j \text{ s.t. } \mathbf{k}^c_i \in \mathcal{K}_{\mathrm{ed}}, \mathbf{k}^c_j \in \mathcal{K}_{\mathrm{non}}. \tag{42}$$

Following the same arguments as Theorem G.13, the proof is concluded.

$\square$

*Remark* G.17. Consider estimating $\mathbf{C}$ from $N_{\mathbf{C}}$ number of i.i.d. samples drawn from same distribution with $\mathbf{C}$. Now $\Delta\mathbf{C} = \hat{\mathbf{C}} - \mathbf{C}$, where $\hat{\mathbf{C}}$ is the estimated covariance matrix $\mathbf{C}$ is the ground truth. Under standard matrix concentration conditions, $\|\Delta\mathbf{C}\|_2$ decreases with $N_{\mathbf{C}}$, typically at an $N_{\mathbf{C}}^{-1/2}$-type rate up to dimension/effective-rank factors.

## G.4. Subject recovery for AlphaEdit, Algorithm 3

From Woodbury Matrix Identity (Lemma G.9),

$$\begin{aligned}
\Delta\mathbf{W} &\coloneqq \Delta\mathbf{W}_{\mathrm{AlphaEdit}} = \mathbf{R}\mathbf{K}^\top \mathbf{P}\left(\mathbf{I} + \mathbf{K}\mathbf{K}^\top \mathbf{P}\right)^{-1} \\
&= \mathbf{R}\mathbf{K}^\top \mathbf{P}\left[\mathbf{I}^{-1} - \mathbf{I}^{-1}\mathbf{K}\left(\mathbf{I} + \mathbf{K}^\top \mathbf{P}\mathbf{I}^{-1}\mathbf{K}\right)^{-1}\mathbf{K}^\top \mathbf{P}\mathbf{I}^{-1}\right] \\
&= \mathbf{R}\left[\mathbf{K}^\top \mathbf{P} - \mathbf{K}^\top \mathbf{P}\mathbf{K}\left(\mathbf{I} + \mathbf{K}^\top \mathbf{P}\mathbf{K}\right)^{-1}\mathbf{K}^\top \mathbf{P}\right] \\
&= \mathbf{R}\left[\left(\mathbf{I} + \mathbf{K}^\top \mathbf{P}\mathbf{K}\right) - \mathbf{K}^\top \mathbf{P}\mathbf{K}\right]\left(\mathbf{I} + \mathbf{K}^\top \mathbf{P}\mathbf{K}\right)^{-1}\mathbf{K}^\top \mathbf{P} \\
&= \mathbf{R}\left(\mathbf{I} + \mathbf{K}^\top \mathbf{P}\mathbf{K}\right)^{-1}\mathbf{K}^\top \mathbf{P}.
\end{aligned} \tag{43}$$

**Lemma G.18.** *Indistinguishability of the key matrix's projection on the orthogonal complement subspace.*

$\forall \mathbf{K} \in \mathbb{R}^{d_{\mathrm{in}} \times N}$, $\mathbf{R} \in \mathbb{R}^{d_{\mathrm{out}} \times N}$, $\forall \Delta\mathbf{K} \in \mathbb{R}^{d_{\mathrm{in}} \times N}$ *such that* $\mathbf{P}\Delta\mathbf{K} = \mathbf{O}$, *for AlphaEdit (at the first edit)* $\Delta\mathbf{W} \coloneqq \Delta\mathbf{W}_{\mathrm{AlphaEdit}}(\mathbf{K}, \mathbf{R})$, *we have* $\Delta\mathbf{W}_{\mathrm{AlphaEdit}}(\mathbf{K}, \mathbf{R}) = \Delta\mathbf{W}_{\mathrm{AlphaEdit}}(\mathbf{K} + \Delta\mathbf{K}, \mathbf{R})$.

*Remark* G.19. Explanation of Lemma G.18.

For a key matrix with decomposition $\mathbf{K} = \mathbf{P}\mathbf{K} + (\mathbf{I}-\mathbf{P})\mathbf{K}$, we have $\mathbf{P}[(\mathbf{I}-\mathbf{P})\mathbf{K}] = \mathbf{O}$. So here we can set $\Delta\mathbf{K} = (\mathbf{I}-\mathbf{P})\mathbf{K}$. Thus Lemma G.18 indicates that given $\Delta\mathbf{W}$, $\mathbf{P}$ and even $\mathbf{R}$, one cannot directly obtain any information of $(\mathbf{I} - \mathbf{P})\mathbf{K}$ without additional data. This is why we use the projected key vectors $\mathbf{k}'^c_i = \mathbf{P}\mathbf{k}^c_i$ rather than $\mathbf{k}^c_i$ itself to calculate the score function in Algorithm 3.

*Proof.* From the property of projection matrix, $\mathbf{P} = \mathbf{P}^2$, $\mathbf{P} = \mathbf{P}^\top$. Then we have

$$
\begin{aligned}
\Delta\mathbf{W}_{\text{AlphaEdit}}(\mathbf{K} + \Delta\mathbf{K}, \mathbf{R}) &= \mathbf{R}\left(\mathbf{I} + (\mathbf{K} + \Delta\mathbf{K})^\top \mathbf{P}\,(\mathbf{K} + \Delta\mathbf{K})\right)^{-1}(\mathbf{K} + \Delta\mathbf{K})^\top \mathbf{P} \\
&= \mathbf{R}\left(\mathbf{I} + (\mathbf{K} + \Delta\mathbf{K})^\top \mathbf{P}^2\,(\mathbf{K} + \Delta\mathbf{K})\right)^{-1}(\mathbf{K} + \Delta\mathbf{K})^\top \mathbf{P} \\
&= \mathbf{R}\left(\mathbf{I} + (\mathbf{P}\,(\mathbf{K} + \Delta\mathbf{K}))^\top (\mathbf{P}\,(\mathbf{K} + \Delta\mathbf{K}))\right)^{-1}(\mathbf{P}\,(\mathbf{K} + \Delta\mathbf{K}))^\top \\
&= \mathbf{R}\left(\mathbf{I} + (\mathbf{PK})^\top (\mathbf{PK})\right)^{-1}(\mathbf{PK})^\top \\
&= \mathbf{R}\left(\mathbf{I} + \mathbf{K}^\top \mathbf{PK}\right)^{-1}\mathbf{K}^\top \mathbf{P} = \Delta\mathbf{W}_{\text{AlphaEdit}}(\mathbf{K}, \mathbf{R}).
\end{aligned}
\tag{44}
$$

This completes the proof.

$\square$

From Lemma G.18, it is impossible to recover $(\mathbf{I} - \mathbf{P})\mathbf{K}$. The following Theorem then proves that the column space of $\mathbf{PK}$ can be recovered through $\Delta\mathbf{W}$ and $\mathbf{P}$:

**Theorem G.20.** *Recovery of linear subspace for AlphaEdit.*

*Under Assumption G.2, given update matrix $\Delta\mathbf{W} = \Delta\mathbf{W}_{\text{AlphaEdit}}$ and the null-space projection matrix $\mathbf{P}$, the column space of $\mathbf{PK}$ is exactly the column space of $\mathbf{V}_N$ ($\mathbf{V}_N$ is defined in Algorithm 3):*

$$
\text{ColSpace}(\mathbf{V}_N) = \text{ColSpace}(\mathbf{PK}).
\tag{45}
$$

*Proof.* From Eq. (43), $\Delta\mathbf{W} = \mathbf{R}\left(\mathbf{I} + \mathbf{K}^\top \mathbf{PK}\right)^{-1}(\mathbf{PK})^\top$, then similar to the proof of Theorem G.12, $\text{rank}(\Delta\mathbf{W}) = N$, $\Delta\mathbf{W} = \mathbf{M} = \mathbf{U}_N \mathbf{\Sigma}_N \mathbf{V}_N^\top$. Thus we have

$$
\mathbf{V}_N = \mathbf{M}^\top \mathbf{U}_N \mathbf{\Sigma}_\mathbf{N}^{-1} = \mathbf{PK}\left(\mathbf{I} + \mathbf{K}^\top \mathbf{PK}\right)^{-1}\mathbf{R}^\top \mathbf{U}_N \mathbf{\Sigma}_N^{-1}.
\tag{46}
$$

Since $\left(\mathbf{I} + \mathbf{K}^\top \mathbf{PK}\right)^{-1}\mathbf{R}^\top \mathbf{U}_N \mathbf{\Sigma}_N^{-1}$ is of rank $N$ and thus invertible, $\text{ColSpace}(\mathbf{V}_N) = \text{ColSpace}(\mathbf{PK})$, which completes the proof.

$\square$

**Theorem G.21.** *Guarantee of subject recovery for AlphaEdit.*

*Under Assumptions G.1, G.2, G.6, given the update matrix $\Delta\mathbf{W} = \Delta\mathbf{W}_{\text{AlphaEdit}}$ and the null-space projection matrix $\mathbf{P}$, Algorithm 3 exactly outputs the $N$ correct edited subjects, with the top $N^{th}$ and $(N+1)^{th}$ score (namely $\rho_{(N)}^{c\downarrow}$ and $\rho_{(N+1)}^{c\downarrow}$) separated by:*

$$
\arccos \rho_{(N+1)}^{c\downarrow} - \arccos \rho_{(N)}^{c\downarrow} \geq \delta\theta.
\tag{47}
$$

*Proof.* Similar to the proof of Theorem G.13.

$\square$

## H. Theoretical Guarantees for Mechanism and Properties of Subspace Camouflage

This section first establishes the proof of the solutions for the constraint problems in Theorem H.1 under appropriate assumptions, then shows that these solutions degenerate into the original weight updates of ROME, MEMIT, and AlphaEdit in Remark H.2. Finally, we provide Theorem H.3 on whether the subspace camouflage defense method is applied or not, and Theorem H.4 on the non-recoverability of the key matrix $\mathbf{K}$. These two properties exhibit the robustness of our defense

strategy against any subspace-based attack. Thus, this *subspace camouflage* defense strategy should be regarded as a general defense mechanism, rather than one specifically tailored to our *KSTER* attack.

We first recall the three constraint problems we establish for subspace camouflage in §5.1 (for MEMIT) and Appendix D (for ROME and AlphaEdit):

$$
\begin{cases}
\text{RowSpace}(\Delta \mathbf{W}_{\text{defense,MEMIT}} \mathbf{C}) & \subseteq \text{ColSpace}(\tilde{\mathbf{K}}) \\
\Delta \mathbf{W}_{\text{defense,MEMIT}} \mathbf{K} & = \Delta \mathbf{W}_{\text{MEMIT}} \mathbf{K}
\end{cases},
$$
$$
\begin{cases}
\text{RowSpace}(\Delta \mathbf{W}_{\text{defense,ROME}} \mathbf{C}) & \subseteq \text{Span}(\{\tilde{\mathbf{k}}_*\}) \\
\Delta \mathbf{W}_{\text{defense,ROME}} \mathbf{k}_* & = \Delta \mathbf{W}_{\text{ROME}} \mathbf{k}_*
\end{cases}, \qquad (48)
$$
$$
\begin{cases}
\text{RowSpace}(\Delta \mathbf{W}_{\text{defense,AlphaEdit}}) & \subseteq \text{ColSpace}(\mathbf{P}\tilde{\mathbf{K}}) \\
\Delta \mathbf{W}_{\text{defense,AlphaEdit}} \mathbf{K} & = \Delta \mathbf{W}_{\text{AlphaEdit}} \mathbf{K}
\end{cases},
$$

and their solutions are listed without proof:

$$
\Delta \mathbf{W}_{\text{defense,MEMIT}} = \Delta \mathbf{W}_{\text{MEMIT}} \mathbf{K} \left( \tilde{\mathbf{K}}^\top \mathbf{C}^{-1} \mathbf{K} \right)^{-1} \tilde{\mathbf{K}}^\top \mathbf{C}^{-1},
$$
$$
\Delta \mathbf{W}_{\text{defense,ROME}} = \frac{\Delta \mathbf{W}_{\text{ROME}} \mathbf{k}_* \tilde{\mathbf{k}}_*^\top \mathbf{C}^{-1}}{\tilde{\mathbf{k}}_*^\top \mathbf{C}^{-1} \mathbf{k}_*}, \qquad (49)
$$
$$
\Delta \mathbf{W}_{\text{defense,AlphaEdit}} = \Delta \mathbf{W}_{\text{AlphaEdit}} \mathbf{K} \left( \tilde{\mathbf{K}}^\top \mathbf{P} \mathbf{K} \right)^{-1} \tilde{\mathbf{K}}^\top \mathbf{P}.
$$

Below, we ignore the regularizer $\lambda$.

The first theorem proves the existence and uniqueness of their solutions under appropriate assumptions:

**Theorem H.1.** *Suppose Assumption G.8 holds. Further assume that both $\tilde{\mathbf{K}}^\top \mathbf{C}^{-1} \mathbf{K}$ and $\tilde{\mathbf{K}}^\top \mathbf{P} \mathbf{K}$ are invertible, and $\tilde{\mathbf{k}}_*^\top \mathbf{C}^{-1} \mathbf{k}_* \neq 0$. Then for the three constraint problems in (48) ((17) for ROME, (9) for MEMIT and (19) for AlphaEdit), their solutions exist and are unique, given by (49) ((18), (10) and (20) respectively).*

*Proof.* We begin with the proof for MEMIT. We first prove the necessity, then the sufficiency, by assuming the existence of the solution.

The first constraint of (9) is equivalent to the following statement: there exists a matrix $\mathbf{F}_{\text{MEMIT}} \in \mathbb{R}^{d_{\text{out}} \times N}$ such that

$$
\Delta \mathbf{W}_{\text{defense,MEMIT}} \mathbf{C} = \mathbf{F}_{\text{MEMIT}} \tilde{\mathbf{K}}^\top. \qquad (50)
$$

From Assumption G.8, $\Delta \mathbf{W}_{\text{defense,MEMIT}} = \mathbf{F}_{\text{MEMIT}} \tilde{\mathbf{K}}^\top \mathbf{C}^{-1}$. Then, by plugging into the second constraint, we obtain:

$$
\Delta \mathbf{W}_{\text{defense,MEMIT}} \mathbf{K} = \mathbf{F}_{\text{MEMIT}} \tilde{\mathbf{K}}^\top \mathbf{C}^{-1} \mathbf{K} = \Delta \mathbf{W}_{\text{MEMIT}} \mathbf{K}. \qquad (51)
$$

Since $\tilde{\mathbf{K}}^\top \mathbf{C}^{-1} \mathbf{K}$ is assumed to be invertible, by right multiplying $\left( \tilde{\mathbf{K}}^\top \mathbf{C}^{-1} \mathbf{K} \right)^{-1}$ we solve $\mathbf{F}_{\text{MEMIT}}$:

$$
\mathbf{F}_{\text{MEMIT}} = \Delta \mathbf{W}_{\text{MEMIT}} \mathbf{K} \left( \tilde{\mathbf{K}}^\top \mathbf{C}^{-1} \mathbf{K} \right)^{-1}. \qquad (52)
$$

Thus, we obtain the final result:

$$
\Delta \mathbf{W}_{\text{defense,MEMIT}} = \Delta \mathbf{W}_{\text{MEMIT}} \mathbf{K} \left( \tilde{\mathbf{K}}^\top \mathbf{C}^{-1} \mathbf{K} \right)^{-1} \tilde{\mathbf{K}}^\top \mathbf{C}^{-1}. \qquad (53)
$$

The proof of necessity for MEMIT is completed. The proof of sufficiency follows directly from verifying the two constraints.

Similar techniques can be applied to ROME and AlphaEdit:

For ROME we have some $\mathbf{f}_{\text{ROME}} \in \mathbb{R}^{d_{\text{out}}}$ such that $\Delta\mathbf{W}_{\text{defense,ROME}}\mathbf{C} = \mathbf{f}_{\text{ROME}}\tilde{\mathbf{k}}_*^\top$, combining with the second constraint we have $\mathbf{f}_{\text{ROME}} = \frac{\Delta\mathbf{W}_{\text{ROME}}\mathbf{k}_*}{\tilde{\mathbf{k}}_*^\top\mathbf{C}^{-1}\mathbf{k}_*}$, giving $\Delta\mathbf{W}_{\text{defense,ROME}} = \frac{\Delta\mathbf{W}_{\text{ROME}}\mathbf{k}_*\tilde{\mathbf{k}}_*^\top\mathbf{C}^{-1}}{\tilde{\mathbf{k}}_*^\top\mathbf{C}^{-1}\mathbf{k}_*}$. For AlphaEdit we have some $\mathbf{F}_{\text{AlphaEdit}} \in \mathbb{R}^{d_{\text{out}} \times N}$ such that $\Delta\mathbf{W}_{\text{defense,AlphaEdit}} = \mathbf{F}_{\text{AlphaEdit}}(\mathbf{P}\tilde{\mathbf{K}})^\top = \mathbf{F}_{\text{AlphaEdit}}\tilde{\mathbf{K}}^\top\mathbf{P}$, by combining with the second constraint the $\mathbf{F}_{\text{AlphaEdit}}$ is given by $\mathbf{F}_{\text{AlphaEdit}} = \Delta\mathbf{W}_{\text{AlphaEdit}}\mathbf{K}\left(\tilde{\mathbf{K}}^\top\mathbf{P}\mathbf{K}\right)^{-1}$, thus $\Delta\mathbf{W}_{\text{defense,AlphaEdit}} = \Delta\mathbf{W}_{\text{AlphaEdit}}\mathbf{K}\left(\tilde{\mathbf{K}}^\top\mathbf{P}\mathbf{K}\right)^{-1}\tilde{\mathbf{K}}^\top\mathbf{P}$. The proof of sufficiency for both ROME and AlphaEdit follows directly from verifying the constraints, respectively. This finally completes the proof.

$\square$

*Remark* H.2. We show that the new weight updates of our subspace camouflage $\Delta\mathbf{W}_{\text{defense},(\cdot)}$ degenerate into the original weight updates $\Delta\mathbf{W}_{(\cdot)}$ when $\alpha \to 0^+$ ($(\cdot)$ denotes the specific editing method), under the assumptions of Theorem H.1. Here assumption of invertibility becomes: there exists some $\epsilon > 0$ such that for any $\alpha' \in [0, \epsilon)$, $\left(\tilde{\mathbf{K}}(\alpha')^\top\mathbf{C}^{-1}\mathbf{K}\right)^{-1}$ and $\left(\tilde{\mathbf{K}}(\alpha')^\top\mathbf{P}\mathbf{K}\right)^{-1}$ are well-defined, $\tilde{\mathbf{k}}_*(\alpha)^\top\mathbf{C}^{-1}\mathbf{k}_* \neq 0$. (Recall Eq. (8) that $\tilde{\mathbf{K}}$ and $\mathbf{k}_*$ are functions of $\alpha$. )

By rewriting the solutions in (49):

$$\begin{aligned}
\Delta\mathbf{W}_{\text{defense,MEMIT}} &= \Delta\mathbf{W}_{\text{MEMIT}}\mathbf{K}\left(\tilde{\mathbf{K}}^\top\mathbf{C}^{-1}\mathbf{K}\right)^{-1}\tilde{\mathbf{K}}^\top\mathbf{C}^{-1} \\
&= \mathbf{R}\left(\mathbf{I} + \mathbf{K}^\top\mathbf{C}^{-1}\mathbf{K}\right)^{-1}\mathbf{K}^\top\mathbf{C}^{-1}\mathbf{K}\left(\tilde{\mathbf{K}}^\top\mathbf{C}^{-1}\mathbf{K}\right)^{-1}\tilde{\mathbf{K}}^\top\mathbf{C}^{-1}, \\
\Delta\mathbf{W}_{\text{defense,ROME}} &= \frac{\Delta\mathbf{W}_{\text{ROME}}\mathbf{k}_*\tilde{\mathbf{k}}_*^\top\mathbf{C}^{-1}}{\tilde{\mathbf{k}}_*^\top\mathbf{C}^{-1}\mathbf{k}_*} \\
&= \frac{\mathbf{r}_*\tilde{\mathbf{k}}_*^\top\mathbf{C}^{-1}}{\tilde{\mathbf{k}}_*^\top\mathbf{C}^{-1}\mathbf{k}_*}, \\
\Delta\mathbf{W}_{\text{defense,AlphaEdit}} &= \Delta\mathbf{W}_{\text{AlphaEdit}}\mathbf{K}\left(\tilde{\mathbf{K}}^\top\mathbf{P}\mathbf{K}\right)^{-1}\tilde{\mathbf{K}}^\top\mathbf{P} \\
&= \mathbf{R}\left(\mathbf{I} + \mathbf{K}^\top\mathbf{P}\mathbf{K}\right)^{-1}\mathbf{K}^\top\mathbf{P}\mathbf{K}\left(\tilde{\mathbf{K}}^\top\mathbf{P}\mathbf{K}\right)^{-1}\tilde{\mathbf{K}}^\top\mathbf{P}.
\end{aligned} \tag{54}$$

If $\alpha \to 0^+$, then $\tilde{\mathbf{K}} \to \mathbf{K}$, $\tilde{\mathbf{k}}_* \to \mathbf{k}_*$, consequently we have:

$$\begin{aligned}
\Delta\mathbf{W}_{\text{defense,MEMIT}} &\to \mathbf{R}\left(\mathbf{I} + \mathbf{K}^\top\mathbf{C}^{-1}\mathbf{K}\right)^{-1}\mathbf{K}^\top\mathbf{C}^{-1}\mathbf{K}\left(\mathbf{K}^\top\mathbf{C}^{-1}\mathbf{K}\right)^{-1}\mathbf{K}^\top\mathbf{C}^{-1} \\
&= \mathbf{R}\left(\mathbf{I} + \mathbf{K}^\top\mathbf{C}^{-1}\mathbf{K}\right)^{-1}\mathbf{K}^\top\mathbf{C}^{-1} = \Delta\mathbf{W}_{\text{MEMIT}}, \\
\Delta\mathbf{W}_{\text{defense,ROME}} &\to \frac{\mathbf{r}_*\mathbf{k}_*^\top\mathbf{C}^{-1}}{\mathbf{k}_*^\top\mathbf{C}^{-1}\mathbf{k}_*} = \Delta\mathbf{W}_{\text{ROME}}, \\
\Delta\mathbf{W}_{\text{defense,AlphaEdit}} &\to \mathbf{R}\left(\mathbf{I} + \mathbf{K}^\top\mathbf{P}\mathbf{K}\right)^{-1}\mathbf{K}^\top\mathbf{P}\mathbf{K}\left(\mathbf{K}^\top\mathbf{P}\mathbf{K}\right)^{-1}\mathbf{K}^\top\mathbf{P} \\
&= \mathbf{R}\left(\mathbf{I} + \mathbf{K}^\top\mathbf{P}\mathbf{K}\right)^{-1}\mathbf{K}^\top\mathbf{P} = \Delta\mathbf{W}_{\text{AlphaEdit}}.
\end{aligned} \tag{55}$$

This completes our illustration of degeneration.

For notation simplicity, we write the original (undefended) weight updates $\Delta\mathbf{W}_{(\cdot)} = \Delta\mathbf{W}_{(\cdot)}(\mathbf{K}, \mathbf{R})$ as functions of the key matrix $\mathbf{K}$ and the residual matrix $\mathbf{R}$, and the camouflaged weight updates $\Delta\mathbf{W}_{\text{defense},(\cdot)} = \Delta\mathbf{W}_{\text{defense},(\cdot)}\left(\mathbf{K}, \tilde{\mathbf{K}}, \mathbf{R}\right)$ as functions of the key matrix $\mathbf{K}$, the aggregated camouflage key matrix $\tilde{\mathbf{K}}$ and the residual matrix $\mathbf{R}$. Here, $(\cdot)$ denotes the specific editing method, which can be ROME, MEMIT, or AlphaEdit. For ROME, $\mathbf{K}$ and $\mathbf{R}$ reduce to $\mathbf{k}_*$ and $\mathbf{r}_*$ respectively.

The following theorem shows that if $\mathbf{R}$ is unknown, one cannot identify that whether the subspace camouflage defense strategy has been applied to the weight change. Specifically, given the camouflaged weight update $\Delta\mathbf{W}_{\text{defense},(\cdot)}$, one can construct a new residual matrix $\mathbf{R}(\cdot)'$, such that $\Delta\mathbf{W}_{\text{defense},(\cdot)}$ is mathematically equivalent to the original weight update $\Delta\mathbf{W}_{(\cdot)}$ that takes $\tilde{\mathbf{K}}$ and $\mathbf{R}'_{(\cdot)}$ as input.

**Theorem H.3.** *Indistinguishability of subspace camouflage and original editing.*

*Suppose Assumption G.8 holds. Given any residual matrix $\mathbf{R} \in \mathbb{R}^{d_{\text{out}} \times N}$, true key matrix $\mathbf{K} \in \mathbb{R}^{d_{\text{in}} \times N}$ and aggregated camouflage key matrix $\tilde{\mathbf{K}} \in \mathbb{R}^{d_{\text{in}} \times N}$ such that both $\tilde{\mathbf{K}}^\top \mathbf{C}^{-1} \mathbf{K}$ and $\tilde{\mathbf{K}}^\top \mathbf{P} \mathbf{K}$ are invertible, there exists $\mathbf{R}'_{(\cdot)}$ such that: taking $\tilde{\mathbf{K}}$ as the new key matrix and $\mathbf{R}'_{(\cdot)}$ as the new residual matrix, the new update weight of the original algorithm (ROME, MEMIT or AlphaEdit) $\Delta\mathbf{W}_{(\cdot)}\left(\tilde{\mathbf{K}}, \mathbf{R}'_{(\cdot)}\right)$ exactly equals to the camouflaged weight $\Delta\mathbf{W}_{\text{defense},(\cdot)}$. Specifically, $\mathbf{r}'_{\text{ROME}}$, $\mathbf{R}'_{\text{MEMIT}}$ and $\mathbf{R}'_{\text{AlphaEdit}}$ are given by:*

$$
\begin{aligned}
\mathbf{r}'_{\text{ROME}} &= \frac{\tilde{\mathbf{k}}_*^\top \mathbf{C}^{-1} \tilde{\mathbf{k}}_*}{\tilde{\mathbf{k}}_*^\top \mathbf{C}^{-1} \mathbf{k}_*} \cdot \mathbf{r}_*, \\
\mathbf{R}'_{\text{MEMIT}} &= \mathbf{R} \left(\mathbf{I} + \mathbf{K}^\top \mathbf{C}^{-1} \mathbf{K}\right)^{-1} \mathbf{K}^\top \mathbf{C}^{-1} \mathbf{K} \left(\tilde{\mathbf{K}}^\top \mathbf{C}^{-1} \mathbf{K}\right)^{-1} \left(\mathbf{I} + \tilde{\mathbf{K}}^\top \mathbf{C}^{-1} \tilde{\mathbf{K}}\right), \\
\mathbf{R}'_{\text{AlphaEdit}} &= \mathbf{R} \left(\mathbf{I} + \mathbf{K}^\top \mathbf{P} \mathbf{K}\right)^{-1} \mathbf{K}^\top \mathbf{P} \mathbf{K} \left(\tilde{\mathbf{K}}^\top \mathbf{P} \mathbf{K}\right)^{-1} \left(\mathbf{I} + \tilde{\mathbf{K}}^\top \mathbf{P} \tilde{\mathbf{K}}\right).
\end{aligned}
\tag{56}
$$

*Proof.* The invertibility of $\tilde{\mathbf{K}}^\top \mathbf{P} \mathbf{K}$ implies that both $\tilde{\mathbf{K}}$ and $\mathbf{P}\mathbf{K}$ have full column rank. Thus by Eq. (24) and Eq. (43), $\Delta\mathbf{W}_{\text{MEMIT}}\left(\tilde{\mathbf{K}}, \mathbf{R}'\right) = \mathbf{R}' \left(\mathbf{I} + \tilde{\mathbf{K}}^\top \mathbf{C}^{-1} \tilde{\mathbf{K}}\right)^{-1} \tilde{\mathbf{K}}^\top \mathbf{C}^{-1}$, $\Delta\mathbf{W}_{\text{AlphaEdit}}\left(\tilde{\mathbf{K}}, \mathbf{R}'\right) = \mathbf{R}' \left(\mathbf{I} + \tilde{\mathbf{K}}^\top \mathbf{P} \tilde{\mathbf{K}}\right)^{-1} \tilde{\mathbf{K}}^\top \mathbf{P}$.

From Eq. (54), we reorganize the new weight updates of our subspace camouflage:

$$
\begin{aligned}
\Delta\mathbf{W}_{\text{defense},\text{ROME}} &= \frac{\mathbf{r}_* \tilde{\mathbf{k}}_*^\top \mathbf{C}^{-1}}{\tilde{\mathbf{k}}_*^\top \mathbf{C}^{-1} \mathbf{k}_*} \\
&= \left[\frac{\tilde{\mathbf{k}}_*^\top \mathbf{C}^{-1} \tilde{\mathbf{k}}_*}{\tilde{\mathbf{k}}_*^\top \mathbf{C}^{-1} \mathbf{k}_*} \cdot \mathbf{r}_*\right] \cdot \frac{\tilde{\mathbf{k}}_*^\top \mathbf{C}^{-1}}{\tilde{\mathbf{k}}_*^\top \mathbf{C}^{-1} \tilde{\mathbf{k}}_*} \\
&= \Delta\mathbf{W}_{\text{ROME}}\left(\tilde{\mathbf{k}}_*, \frac{\tilde{\mathbf{k}}_*^\top \mathbf{C}^{-1} \tilde{\mathbf{k}}_*}{\tilde{\mathbf{k}}_*^\top \mathbf{C}^{-1} \mathbf{k}_*} \cdot \mathbf{r}_*\right),
\end{aligned}
\tag{57}
$$

$$
\begin{aligned}
\Delta\mathbf{W}_{\text{defense},\text{MEMIT}} &= \mathbf{R} \left(\mathbf{I} + \mathbf{K}^\top \mathbf{C}^{-1} \mathbf{K}\right)^{-1} \mathbf{K}^\top \mathbf{C}^{-1} \mathbf{K} \left(\tilde{\mathbf{K}}^\top \mathbf{C}^{-1} \mathbf{K}\right)^{-1} \tilde{\mathbf{K}}^\top \mathbf{C}^{-1} \\
&= \left[\mathbf{R} \left(\mathbf{I} + \mathbf{K}^\top \mathbf{C}^{-1} \mathbf{K}\right)^{-1} \mathbf{K}^\top \mathbf{C}^{-1} \mathbf{K} \left(\tilde{\mathbf{K}}^\top \mathbf{C}^{-1} \mathbf{K}\right)^{-1} \left(\mathbf{I} + \tilde{\mathbf{K}}^\top \mathbf{C}^{-1} \tilde{\mathbf{K}}\right)\right] \\
&\quad \cdot \left(\mathbf{I} + \tilde{\mathbf{K}}^\top \mathbf{C}^{-1} \tilde{\mathbf{K}}\right)^{-1} \tilde{\mathbf{K}}^\top \mathbf{C}^{-1} \\
&= \Delta\mathbf{W}_{\text{MEMIT}}\left(\tilde{\mathbf{K}}, \mathbf{R} \left(\mathbf{I} + \mathbf{K}^\top \mathbf{C}^{-1} \mathbf{K}\right)^{-1} \mathbf{K}^\top \mathbf{C}^{-1} \mathbf{K} \left(\tilde{\mathbf{K}}^\top \mathbf{C}^{-1} \mathbf{K}\right)^{-1} \left(\mathbf{I} + \tilde{\mathbf{K}}^\top \mathbf{C}^{-1} \tilde{\mathbf{K}}\right)\right),
\end{aligned}
\tag{58}
$$

$$
\begin{aligned}
\Delta\mathbf{W}_{\text{defense},\text{AlphaEdit}} &= \mathbf{R} \left(\mathbf{I} + \mathbf{K}^\top \mathbf{P} \mathbf{K}\right)^{-1} \mathbf{K}^\top \mathbf{P} \mathbf{K} \left(\tilde{\mathbf{K}}^\top \mathbf{P} \mathbf{K}\right)^{-1} \tilde{\mathbf{K}}^\top \mathbf{P} \\
&= \left[\mathbf{R} \left(\mathbf{I} + \mathbf{K}^\top \mathbf{P} \mathbf{K}\right)^{-1} \mathbf{K}^\top \mathbf{P} \mathbf{K} \left(\tilde{\mathbf{K}}^\top \mathbf{P} \mathbf{K}\right)^{-1} \left(\mathbf{I} + \tilde{\mathbf{K}}^\top \mathbf{P} \tilde{\mathbf{K}}\right)\right] \\
&\quad \cdot \left(\mathbf{I} + \tilde{\mathbf{K}}^\top \mathbf{P} \tilde{\mathbf{K}}\right)^{-1} \tilde{\mathbf{K}}^\top \mathbf{P} \\
&= \Delta\mathbf{W}_{\text{AlphaEdit}}\left(\tilde{\mathbf{K}}, \mathbf{R} \left(\mathbf{I} + \mathbf{K}^\top \mathbf{P} \mathbf{K}\right)^{-1} \mathbf{K}^\top \mathbf{P} \mathbf{K} \left(\tilde{\mathbf{K}}^\top \mathbf{P} \mathbf{K}\right)^{-1} \left(\mathbf{I} + \tilde{\mathbf{K}}^\top \mathbf{P} \tilde{\mathbf{K}}\right)\right).
\end{aligned}
\tag{59}
$$

This completes our proof.

$\square$

Finally, the following theorem shows that our subspace camouflage defense strategy leaks virtually no further information about $\mathbf{K}$ beyond what is revealed by $\tilde{\mathbf{K}}$, under the assumption that $\mathbf{R}$ is unknown. Specifically, given $\Delta \mathbf{W}_{\text{defense},(\cdot)} \left( \mathbf{K}, \tilde{\mathbf{K}}, \mathbf{R} \right)$, for almost all $\mathbf{K}'$ of the same shape as $\tilde{\mathbf{K}}$ (detailed in Remark H.5), we can construct an $\mathbf{R}'_{(\cdot)}$ such that $\Delta \mathbf{W}_{\text{defense},(\cdot)} \left( \mathbf{K}', \tilde{\mathbf{K}}, \mathbf{R}'_{(\cdot)} \right) = \Delta \mathbf{W}_{\text{defense},(\cdot)} \left( \mathbf{K}, \tilde{\mathbf{K}}, \mathbf{R} \right)$:

**Theorem H.4.** *Non-recoverability of the true key matrix for subspace camouflage.*

*Suppose Assumption G.8 holds. Given any residual matrix $\mathbf{R} \in \mathbb{R}^{d_{\text{out}} \times N}$, true key matrix $\mathbf{K} \in \mathbb{R}^{d_{\text{in}} \times N}$ and aggregated camouflage key matrix $\tilde{\mathbf{K}} \in \mathbb{R}^{d_{\text{in}} \times N}$, suppose both $\tilde{\mathbf{K}}^\top \mathbf{C}^{-1} \mathbf{K}$ and $\tilde{\mathbf{K}}^\top \mathbf{P} \mathbf{K}$ are invertible. Then $\forall \mathbf{K}' \in \mathbb{R}^{d_{\text{in}} \times N}$ such that both $\tilde{\mathbf{K}}^\top \mathbf{C}^{-1} \mathbf{K}'$ and $\tilde{\mathbf{K}}^\top \mathbf{P} \mathbf{K}'$ are invertible, there exists $\mathbf{R}'_{(\cdot)}$ such that $\Delta \mathbf{W}_{\text{defense},(\cdot)} \left( \mathbf{K}', \tilde{\mathbf{K}}, \mathbf{R}'_{(\cdot)} \right) = \Delta \mathbf{W}_{\text{defense},(\cdot)} \left( \mathbf{K}, \tilde{\mathbf{K}}, \mathbf{R} \right)$, where $\mathbf{r}'_{\text{ROME}}$, $\mathbf{R}'_{\text{MEMIT}}$ and $\mathbf{R}'_{\text{AlphaEdit}}$ are given by:*

$$
\mathbf{r}'_{\text{ROME}} = \frac{\tilde{\mathbf{k}}_*^\top \mathbf{C}^{-1} \mathbf{k}'_*}{\tilde{\mathbf{k}}_*^\top \mathbf{C}^{-1} \mathbf{k}_*} \cdot \mathbf{r}_*,
$$

$$
\mathbf{R}'_{\text{MEMIT}} = \mathbf{R} \left( \mathbf{I} + \mathbf{K}^\top \mathbf{C}^{-1} \mathbf{K} \right)^{-1} \mathbf{K}^\top \mathbf{C}^{-1} \mathbf{K} \left( \tilde{\mathbf{K}}^\top \mathbf{C}^{-1} \mathbf{K} \right)^{-1} \left( \tilde{\mathbf{K}}^\top \mathbf{C}^{-1} \mathbf{K}' \right) \left( \mathbf{K}'^\top \mathbf{C}^{-1} \mathbf{K}' \right)^{-1} \left( \mathbf{I} + \mathbf{K}'^\top \mathbf{C}^{-1} \mathbf{K}' \right),
$$

$$
\mathbf{R}'_{\text{AlphaEdit}} = \mathbf{R} \left( \mathbf{I} + \mathbf{K}^\top \mathbf{P} \mathbf{K} \right)^{-1} \mathbf{K}^\top \mathbf{P} \mathbf{K} \left( \tilde{\mathbf{K}}^\top \mathbf{P} \mathbf{K} \right)^{-1} \left( \tilde{\mathbf{K}}^\top \mathbf{P} \mathbf{K}' \right) \left( \mathbf{K}'^\top \mathbf{P} \mathbf{K}' \right)^{-1} \left( \mathbf{I} + \mathbf{K}'^\top \mathbf{P} \mathbf{K}' \right).
$$

$$
(60)
$$

*Proof.* Since $\tilde{\mathbf{K}}^\top \mathbf{C}^{-1} \mathbf{K}'$ is invertible, $\mathbf{C}^{-1} \mathbf{K}'$ has full column rank, thus $\mathbf{K}'^\top \mathbf{C}^{-1} \mathbf{K}' = \left( \mathbf{C}^{-1} \mathbf{K}' \right)^\top \mathbf{C} \left( \mathbf{C}^{-1} \mathbf{K}' \right) \succ \mathbf{O}$, further giving that both $\mathbf{K}'^\top \mathbf{C}^{-1} \mathbf{K}'$ and $\mathbf{I} + \mathbf{K}'^\top \mathbf{C}^{-1} \mathbf{K}'$ are invertible. Similarly, from $\tilde{\mathbf{K}}^\top \mathbf{P} \mathbf{K}'$ is invertible, $\mathbf{P} \mathbf{K}'$ is full column rank; since $\mathbf{P}$ is an orthogonal projection matrix, $\mathbf{P}^2 = \mathbf{P} = \mathbf{P}^\top$, thus $\mathbf{K}'^\top \mathbf{P} \mathbf{K}' = \mathbf{K}'^\top \mathbf{P}^\top \mathbf{P} \mathbf{K}' = \left( \mathbf{P} \mathbf{K}' \right)^\top \left( \mathbf{P} \mathbf{K}' \right) \succ \mathbf{O}$, giving that both $\mathbf{K}'^\top \mathbf{P} \mathbf{K}'$ and $\mathbf{I} + \mathbf{K}'^\top \mathbf{P} \mathbf{K}'$ are also invertible.

By reorganizing the camouflaged weight updates $\Delta \mathbf{W}_{\text{defense},(\cdot)} \left( \mathbf{K}, \tilde{\mathbf{K}}, \mathbf{R} \right)$ in Eq. (54):

$$
\begin{aligned}
\Delta \mathbf{W}_{\text{defense,ROME}} \left( \mathbf{k}_*, \tilde{\mathbf{k}}_*, \mathbf{r}_* \right) &= \frac{\mathbf{r} \tilde{\mathbf{k}}_*^\top \mathbf{C}^{-1}}{\tilde{\mathbf{k}}_*^\top \mathbf{C}^{-1} \mathbf{k}_*} \\
&= \left[ \frac{\tilde{\mathbf{k}}_*^\top \mathbf{C}^{-1} \mathbf{k}'_*}{\tilde{\mathbf{k}}_*^\top \mathbf{C}^{-1} \mathbf{k}_*} \cdot \mathbf{r}_* \right] \cdot \frac{\tilde{\mathbf{k}}_*^\top \mathbf{C}^{-1}}{\tilde{\mathbf{k}}_*^\top \mathbf{C}^{-1} \mathbf{k}'_*} \\
&= \Delta \mathbf{W}_{\text{defense,ROME}} \left( \mathbf{k}'_*, \tilde{\mathbf{k}}_*, \mathbf{r}'_{\text{ROME}} \right),
\end{aligned}
\qquad (61)
$$

$$
\begin{aligned}
&\Delta \mathbf{W}_{\text{defense,MEMIT}} \left( \mathbf{K}, \tilde{\mathbf{K}}, \mathbf{R} \right) \\
=&\mathbf{R} \left( \mathbf{I} + \mathbf{K}^\top \mathbf{C}^{-1} \mathbf{K} \right)^{-1} \mathbf{K}^\top \mathbf{C}^{-1} \mathbf{K} \left( \tilde{\mathbf{K}}^\top \mathbf{C}^{-1} \mathbf{K} \right)^{-1} \tilde{\mathbf{K}}^\top \mathbf{C}^{-1} \\
=& \left[ \mathbf{R} \left( \mathbf{I} + \mathbf{K}^\top \mathbf{C}^{-1} \mathbf{K} \right)^{-1} \mathbf{K}^\top \mathbf{C}^{-1} \mathbf{K} \left( \tilde{\mathbf{K}}^\top \mathbf{C}^{-1} \mathbf{K} \right)^{-1} \left( \tilde{\mathbf{K}}^\top \mathbf{C}^{-1} \mathbf{K}' \right) \left( \mathbf{K}'^\top \mathbf{C}^{-1} \mathbf{K}' \right)^{-1} \left( \mathbf{I} + \mathbf{K}'^\top \mathbf{C}^{-1} \mathbf{K}' \right) \right] \\
& \cdot \left( \mathbf{I} + \mathbf{K}'^\top \mathbf{C}^{-1} \mathbf{K}' \right)^{-1} \mathbf{K}'^\top \mathbf{C}^{-1} \mathbf{K}' \left( \tilde{\mathbf{K}}^\top \mathbf{C}^{-1} \mathbf{K}' \right)^{-1} \tilde{\mathbf{K}}^\top \mathbf{C}^{-1} \\
=& \Delta \mathbf{W}_{\text{defense,MEMIT}} \left( \mathbf{K}', \tilde{\mathbf{K}}, \mathbf{R}'_{\text{MEMIT}} \right),
\end{aligned}
\qquad (62)
$$

$$\Delta\mathbf{W}_{\text{defense,AlphaEdit}}\left(\mathbf{K},\tilde{\mathbf{K}},\mathbf{R}\right)$$

$$=\mathbf{R}\left(\mathbf{I}+\mathbf{K}^{\top}\mathbf{P}\mathbf{K}\right)^{-1}\mathbf{K}^{\top}\mathbf{P}\mathbf{K}\left(\tilde{\mathbf{K}}^{\top}\mathbf{P}\mathbf{K}\right)^{-1}\tilde{\mathbf{K}}^{\top}\mathbf{P}$$

$$=\left[\mathbf{R}\left(\mathbf{I}+\mathbf{K}^{\top}\mathbf{P}\mathbf{K}\right)^{-1}\mathbf{K}^{\top}\mathbf{P}\mathbf{K}\left(\tilde{\mathbf{K}}^{\top}\mathbf{P}\mathbf{K}\right)^{-1}\left(\tilde{\mathbf{K}}^{\top}\mathbf{P}\mathbf{K}'\right)\left(\mathbf{K}'^{\top}\mathbf{P}\mathbf{K}'\right)^{-1}\left(\mathbf{I}+\mathbf{K}'^{\top}\mathbf{P}\mathbf{K}'\right)\right] \tag{63}$$

$$\cdot\left(\mathbf{I}+\mathbf{K}'^{\top}\mathbf{P}\mathbf{K}'\right)^{-1}\mathbf{K}'^{\top}\mathbf{P}\mathbf{K}'\left(\tilde{\mathbf{K}}^{\top}\mathbf{P}\mathbf{K}'\right)^{-1}\tilde{\mathbf{K}}^{\top}\mathbf{P}$$

$$=\Delta\mathbf{W}_{\text{defense,AlphaEdit}}\left(\mathbf{K}',\tilde{\mathbf{K}},\mathbf{R}'_{\text{AlphaEdit}}\right).$$

This completes the proof.

$\square$

*Remark* H.5. Discussion on the mild requirements of $\mathbf{K}'$.

In Theorem H.4, we require both $\tilde{\mathbf{K}}^{\top}\mathbf{C}^{-1}\mathbf{K}'$ and $\tilde{\mathbf{K}}^{\top}\mathbf{P}\mathbf{K}'$ to be invertible. Since both $\tilde{\mathbf{K}}^{\top}\mathbf{C}^{-1}\mathbf{K}$ and $\tilde{\mathbf{K}}^{\top}\mathbf{P}\mathbf{K}$ are invertible, $\tilde{\mathbf{K}}^{\top}\mathbf{C}^{-1}$ and $\tilde{\mathbf{K}}^{\top}\mathbf{P}$ have full row rank. Thus, the corresponding two violation sets $\left\{\mathbf{K}': \det\left(\tilde{\mathbf{K}}^{\top}\mathbf{C}^{-1}\mathbf{K}'\right)=0\right\}$ and $\left\{\mathbf{K}': \det\left(\tilde{\mathbf{K}}^{\top}\mathbf{P}\mathbf{K}'\right)=0\right\}$ are zero sets of polynomials, and thus have *Lebesgue measure zero*. Therefore, the invertibility conditions fail only for a negligible portion of all possible $\mathbf{K}'$, which justifies the claim about "almost all" $\mathbf{K}'$.

