# OpenReview forum: "Reverse-Engineering Model Editing on Language Models"
_ICML.cc/2026/Conference — ICML 2026 regular_

### Official Review · Reviewer_2fzm · 2026-03-02

**Soundness:** 3
**Presentation:** 3
**Significance:** 2
**Originality:** 2
**Overall Recommendation:** 3
**Confidence:** 3

**Summary:**

This study's principal domain comprises the security and privacy of large language models, specifically focusing on reverse-engineering attacks against locate-then-edit model editing techniques. The paper introduces KSTER, a two-stage attack that inverts parameter updates to recover edited subjects and prompts, achieving high success rates across multiple LLMs and editing methods. It also proposes subspace camouflage, a defense that injects semantic decoys to obscure the editing fingerprint while preserving editing efficacy. Overall, the authors present an important domain of research revealing that model editing can leak sensitive data, and they offer both an attack to audit edits and a defense to mitigate risks, emphasizing the need for secure editing mechanisms.

**Compliance With Llm Reviewing Policy:**

Affirmed.

**Final Justification:**

The paper addresses a relevant and non-trivial problem, and the proposed approach is technically sound in several aspects. The rebuttal clarifies some implementation details and partially addresses concerns about the experimental setup. However, I remain concerned that the core methodology relies on assumptions or design choices that are not sufficiently justified or grounded in established principles. This raises issues about reproducibility, generalizability, and potential bias in the evaluation. As it stands, these limitations reduce my confidence in the validity of the claims and the overall impact of the work. Therefore, I will keep my score as it is.

**Key Questions For Authors:**

1. Your attack operates in a white-box setting requiring access to the weight update $ΔW$. How do you envision this attack being executed against a deployed, closed-source LLM where only API access is available? If the primary threat is from open-source models where weights are public, does that fundamentally change the significance and scope of the vulnerability you are highlighting?

2. For the prompt recovery task, how does your entropy-based scoring function compare against a simpler baseline that directly queries the post-edit model with a candidate prompt and measures the log-probability of the target object? Could this simpler method achieve comparable or better results without requiring access to the pre-edit model's outputs, and if not, what specific advantage does your method provide?

3. You explicitly leave multi-time (sequential) editing for future work, noting that parameter updates become an "entangled superposition" that makes recovery harder. Do you have any preliminary theoretical or empirical evidence to suggest whether the attack could be extended to disentangle these updates, or does the superposition fundamentally break the linear subspace assumption your method relies on?

4. Your "subspace camouflage" defense is evaluated against your own KSTER attack. Have you considered or tested whether this defense would be effective against a different type of reverse-engineering attack—for example, one that targets the residual matrix $R$ or analyzes the difference in activations rather than the weight subspace? In other words, is the defense robust to unknown attack strategies, or does it only obscure the specific fingerprint your method exploits?

5. You identify two primary failure modes: "Semantic Generalization Error" and "Optimization Constraint Conflict." Could you provide quantitative estimates of how frequently each failure mode occurs across your experiments? Understanding the prevalence of these failures would help gauge the overall reliability of your attack in practice.

**Limitations:**

yes

**Strengths And Weaknesses:**

Strengths:
+ The attack's theoretical grounding in linear algebra provides a clean and generalizable framework for understanding and exploiting the algebraic structure of parameter updates.

+ The discovery of distinct failure modes—semantic ambiguity and optimization conflicts—offers valuable diagnostic insights into the fundamental limitations of current editing algorithms.

Weaknesses:
- The attack assumes a white-box setting where the adversary has full access to the model's internal weights and the exact parameter update $ΔW$. In many real-world scenarios, edited models are deployed as APIs or black-box services, making the direct acquisition of weight deltas implausible. While the authors compare against a gray-box baseline, the core attack's reliance on internal algebraic structures means its practical impact is confined to scenarios where model weights are openly shared or leaked, which significantly narrows the scope of the claimed threat.

- The evaluation of the prompt recovery attack lacks a critical baseline. The authors do not compare their entropy-based scoring function against a simpler method, such as directly querying the post-edit model with candidate prompts and measuring the probability of the target object. This is a more straightforward attack that an adversary would likely attempt first. Without demonstrating that the proposed method outperforms or provides unique value beyond this simple baseline, the contribution of the second stage of the attack is unclear.

- The paper explicitly restricts its analysis to single-time and independent batch edits, deferring multi-time (sequential) editing to future work. This is a major limitation, as real-world deployment of model editing would almost certainly involve a sequence of edits over time. The parameter updates in such a scenario become an entangled superposition, and the paper acknowledges that this would make accurate recovery "substantially harder." By failing to address this more realistic and complex setting, the work's conclusions about the severity of the leakage risk are significantly weakened.

- The proposed "subspace camouflage" defense is evaluated primarily by measuring its effectiveness against the authors' own proposed attack (KSTER). While this is a standard first step, it raises concerns about overfitting to a specific attack methodology. The defense works by obfuscating the subspace that this particular attack targets. It is unclear whether the defense would be equally robust against a different, unforeseen reverse-engineering technique that might exploit another aspect of the parameter update, such as the residual matrix $R$ or the interaction between the update and the unedited weights.

---

> ### Author Rebuttal · Authors · 2026-03-31
>
> Thanks for your valuable feedback. Our detailed responses are provided below and full experimental results are available at https://anonymous.4open.science/r/EditAttackFile-CD44/additional_experiments.md.
>
> ## W1&Q1: Concern regarding the applicability of the attack to closed-source models.
>
> Thank you for raising this insight. For attackers that only access APIs,  current research [1] can only recover the final MLP layer of a black-box model. How to recover specific layer weights falls under model stealing attacks, which is beyond the scope of our research.
>
> We want to respectfully point out that, to the best of our knowledge, __our work is the first study of reverse-engineering edited data__, and we establish a unified framework that reveals the inherent vulnerability of the located-then-edit paradigm. This work exposes the major threats of such techniques in two reasonable scenarios:
>
> (1) External threats: Recent research [2,3] employs model editing to erase sensitive data from public models; however, once the weights are released, $\Delta W$ leaks the very data it aims to hide.
>
> (2) Insider threat: In corporate environments with restricted data access, a malicious insider with only weight permissions can bypass access controls and extract confidential data through the attack.
>
>
> ## W2&Q2: Comparison with the Probability-Based Baseline.
>
> Thank you for this constructive suggestion. We supplement your intuitive approach in our baselines. As shown in Table 1, although this baseline achieves similar recovery performance to our method on the Top-5 and Top-20 metrics, __our method outperforms the baseline by over 10% on the Top-1 metric__. We will include these comparison results in the revision.
>
> **Table 1: Results of prompt recovery attack (baseline) using MEMIT in the Counterfact dataset (num_edit=100).**
>
> | Model | Method |  Top-1   |  Top-5   |  Top-20  |   Sim.   |
> | :--- | :--- |:--------:|:--------:|:--------:|:--------:|
> | `Llama3-8B-Instruct` | **Ours** | **0.51** | **0.81** | **0.94** | **0.88** |
> | | Baseline |   0.35   |   0.79   |   0.92   |   0.87   |
> | `Qwen2.5-7B-Instruct`| **Ours** | **0.47** | **0.86** | **0.97** | **0.86** |
> | | Baseline |   0.31   |   0.84   |   0.94   |   0.84   |
> | `GPT-J-6B` | **Ours** | **0.50** | **0.92** | **0.99** | **0.88** |
> | | Baseline |   0.37   |   0.88   | **0.99** |   0.86   |
>
> ## W3&Q3: Concern regarding the applicability of the attack to sequential editing.
>
> Thank you for your comment. We would like to clarify a misunderstanding regarding "sequential editing" and "entangled superposition." In the real-world setting (where developers release new model versions as new data arrives), __our attack covers this scenario by computing parameter updates between adjacent model versions__. The "entangled superposition" only arises when a system faces a massive batch of data that cannot be processed all at once (e.g., OOM). In this scenario, the obtained $\Delta W$ is aggregated from multiple update matrices. This represents an extreme edge case, and we leave it as future work.
>
> ## W4&Q4: Concern regarding the generalizability of our defense to unforeseen attacks.
>
> Thank you for raising this concern. We will provide a theoretical analysis of our defense against unforeseen attacks in the revision. Specifically, we have proven that: without additional knowledge of the exact residual matrix $R$ (rather than merely its row space via SVD), an attacker can neither (i) detect whether the model has deployed this defense, nor (ii) recover any meaningful information about the key matrix $K$. It should be noted that estimating $R$ to bypass the defense is infeasible, because constructing $R$ must rely on the target and original values before and after editing. This means that the difficulty of extracting $R$ is no less than breaking the defense. Therefore, the update cannot form meaningful interaction with unedited weights in the absence of knowledge of $K$ and $R$.
>
> ## Q5: Regarding the estimation of two failure rates in the current editing methods.
>
> We appreciate your focus on this issue, and we consistently maintain that this is a critical bottleneck in the current locate-then-edit paradigm. We provide the two failure rates on Llama3-8B-Instruct in Table 2.
>
> **Table 2: Results of two failure rates on Llama3-8B-Instruct at num_edit=100.**
>
> | Dataset | Method | OCC (%) | SGE (%) |
> | :--- | :--- | :---: | :---: |
> | **MCF** | MEMIT | 3.6 | 14.6 |
> | | AlphaEdit | 3.8 | 14.2 |
> | **zsRE** | MEMIT | 12.8 | 21.4 |
> | | AlphaEdit | 11.8 | 22.2 |
> ___
>
> We are eager to know whether we have addressed your questions, and we would be happy to discuss further at any time to assist with your re-evaluation.
>
> Best,
>
> Authors of Paper 20446
>
> _[1] Stealing Part of a Production Language Model. 2024_
>
> _[2] Investigating Model Editing for Unlearning in Large Language Models. 2025_
>
> _[3] Editing as unlearning: Are knowledge editing methods strong baselines for large language model unlearning? 2026_

---

> > ### Author Rebuttal · Reviewer_2fzm · 2026-04-03
> >
> > Thank you for the detailed rebuttal. I found the response helpful, and it addresses several of my concerns in a meaningful way.
> >
> > Most notably, the addition of the probability-based baseline comparison (Table 1) convincingly demonstrates that your entropy-based method provides nontrivial gains on Top-1 accuracy, and the theoretical analysis of the subspace camouflage defense offers a principled argument for its robustness beyond the specific KSTER attack. The clarification of threat models (external release and insider access) also helps contextualize the white-box assumption.
> >
> > That said, my main concerns are only partially resolved. I still have residual concerns about the handling of sequential editing and the empirical validation of the defense’s generalizability. Specifically:
> > - Sequential editing (W3/Q3): You reinterpret “sequential editing” as computing differences between adjacent released model versions, which indeed captures a practical scenario. However, the original concern was about a single model undergoing multiple edits without intermediate version releases (e.g., a deployed editing system that accumulates edits over time). In that case, the attacker only sees the final model weights and the original weights, and $\Delta W$ becomes an “entangled superposition” of many edits. You label this an “extreme edge case,” but it is not uncommon—for example, a knowledge base being continuously updated. Do you have any preliminary experiments or theoretical bounds on how the attack’s accuracy degrades as the number of sequential edits grows? Without this, the claim that the leakage risk generalizes to realistic deployment remains under-supported.
> > - Defense robustness (W4/Q4): Your theoretical analysis shows that without knowledge of the exact residual matrix $R$, an attacker cannot recover meaningful information. However, the original question asked about a different type of reverse-engineering attack—for instance, one that targets activation differences or gradient information rather than the weight subspace. Does your defense also obscure such alternative attack surfaces? A brief empirical test against a simple heuristic (e.g., analyzing output logits or hidden state differences before and after editing) would significantly strengthen the claim of general robustness. If no such experiment has been conducted, please clarify what assumptions a future adaptive attacker would need to violate your defense.

---

> > > ### Author Response · Authors · 2026-04-03
> > >
> > > Thank you for your detailed feedback! We reply to your concerns below.
> > >
> > > ## Sequential editing:
> > >
> > > Thank you for emphasizing this point. We supplement experiments on sequential editing in Table 1. Surprisingly, we observe that attack performance remains stable as the number of sequential edits increases (without any modifications to score functions), which exhibits the robustness of our attack. Thank you again for your insightful question that strengthens our contribution, and we will add this result to our updated version.
> > >
> > > **Table 1:  Attack performance on sequential editing across different sequential numbers (seq_num) at num_edits=100. We report Top-(seq_num*num_edits) recall.**
> > >
> > >
> > > | Dataset | Algorithm | Model | seq_num=2 | seq_num=3 | seq_num=4 | seq_num=5 | seq_num=6 | seq_num=7 | seq_num=8 | seq_num=9 | seq_num=10 |
> > > | :--- | :--- | :---: | :-------: | :-------: | :-------: | :-------: | :-------: | :-------: | :-------: | :-------: | :-------: |
> > > | mcf | MEMIT | `GPT-J-6B` | 0.97 | 0.97 | 0.97 | 0.97 | 0.97 | 0.97 | 0.97 | 0.97 | 0.97 |
> > > | mcf | MEMIT | `Llama3-8B-Instruct` | 0.99 | 0.99 | 0.98 | 0.98 | 0.98 | 0.98 | 0.98 | 0.98 | 0.98 |
> > > | mcf | MEMIT | `Qwen2.5-7B-Instruct` | 0.94 | 0.94 | 0.95 | 0.95 | 0.95 | 0.95 | 0.95 | 0.95 | 0.95 |
> > > | mcf | AlphaEdit | `GPT-J-6B` | 0.96 | 0.97 | 0.97 | 0.97 | 0.97 | 0.97 | 0.97 | 0.97 | 0.97 |
> > > | mcf | AlphaEdit | `Llama3-8B-Instruct` | 0.98 | 0.99 | 0.99 | 0.99 | 0.99 | 0.99 | 0.99 | 0.99 | 0.99 |
> > > | mcf | AlphaEdit | `Qwen2.5-7B-Instruct` | 0.95 | 0.95 | 0.96 | 0.96 | 0.96 | 0.96 | 0.96 | 0.96 | 0.96 |
> > > | zsre | MEMIT | `GPT-J-6B` | 0.99 | 1.00 | 1.00 | 1.00 | 1.00 | 1.00 | 1.00 | 1.00 | 1.00 |
> > > | zsre | MEMIT | `Llama3-8B-Instruct` | 0.99 | 0.99 | 1.00 | 1.00 | 1.00 | 1.00 | 1.00 | 1.00 | 1.00 |
> > > | zsre | MEMIT | `Qwen2.5-7B-Instruct` | 0.98 | 1.00 | 1.00 | 1.00 | 1.00 | 1.00 | 1.00 | 1.00 | 1.00 |
> > > | zsre | AlphaEdit | `GPT-J-6B` | 1.00 | 1.00 | 1.00 | 1.00 | 1.00 | 1.00 | 1.00 | 1.00 | 1.00 |
> > > | zsre | AlphaEdit | `Llama3-8B-Instruct` | 0.99 | 1.00 | 1.00 | 1.00 | 1.00 | 1.00 | 1.00 | 1.00 | 1.00 |
> > > | zsre | AlphaEdit | `Qwen2.5-7B-Instruct` | 0.99 | 1.00 | 1.00 | 1.00 | 1.00 | 1.00 | 1.00 | 1.00 | 1.00 |
> > >
> > >
> > > ## Defense robustness:
> > >
> > > Thank you for the insightful question. Regarding alternative attacks, we evaluate our defense using the subject inference baseline introduced in Appendix E.1 (gray-box setting). Specifically, this baseline infers the edited subject by calculating the JS divergence of output logits before and after editing. As shown in Table 2, our defense achieves strong effectiveness against such attacks, validating the robustness of our method. Thank you again and we will add this result to our updated version.
> > >
> > > **Table 2: Protection Performance against logit-based baseline (Llama3-8B-Instruct, MEMIT) at num_edits=100.**
> > >
> > > | Dataset | Camouflage scale | Recall | Mean rank |
> > > | :--- | :---: | :---: | :---: |
> > > | mcf | 0 | 0.66 | 61 |
> > > | mcf | 1 | 0.57 | 86 |
> > > | mcf | 3 | 0.42 | 153 |
> > > | mcf | 5 | 0.03 | 324 |
> > > | mcf | 7 | 0.00 | 934 |
> > > | zsRE | 0 | 0.69 | 57 |
> > > | zsRE | 1 | 0.52 | 93 |
> > > | zsRE | 3 | 0.38 | 162 |
> > > | zsRE | 5 | 0.07 | 314 |
> > > | zsRE | 7 | 0.02 | 798 |
> > >
> > > ___
> > > We are eager to know whether we have addressed your questions, and we would be happy to discuss further at any time to assist with your re-evaluation.
> > >
> > > Best,
> > >
> > > Authors of Paper 20446
> > > ___
> > >
> > >
> > > ## Update in response to the final justification:
> > >
> > > Dear Reviewer 2fzm,
> > >
> > > Thank you for your feedback.
> > >
> > > Regarding the concerns about generalizability, reproducibility, and potential bias raised in your final justification, we would like to respectfully point out that __our latest response above has covered these concerns__:
> > >
> > > * __Generalizability__: As requested in your rebuttal acknowledgement, __Table 1 validates the robustness of our attack in the sequential editing scenario__ (multiple edits aggregated before release), while __Table 2 shows the effectiveness of our defense against alternative threats__, such as logit-based attacks.
> > >
> > > * __Reproducibility__: We have provided an anonymous link to our codebase in the initial submission, and the code details for the additional experiments will be open-sourced upon publication.
> > >
> > > * __Potential Bias__: We have evaluated our approach on standard benchmarks (mcf, zsRE) without cherry-picking the results.
> > >
> > >
> > > We sincerely appreciate your feedback and trust that these results can resolve your remaining concerns.
> > >
> > > Best,
> > >
> > > Authors of Paper 20446

---

### Official Review · Reviewer_uAeZ · 2026-03-09

**Soundness:** 3
**Presentation:** 3
**Significance:** 3
**Originality:** 3
**Overall Recommendation:** 4
**Confidence:** 4

**Summary:**

This paper studies a new privacy risk in locate-then-edit model editing methods (ROME/MEMIT/AlphaEdit) for LLMs: the weight update itself can leak the identity of edited subjects and the semantic prompt used for editing. The authors propose KSTER and a defense.

**Compliance With Llm Reviewing Policy:**

Affirmed.

**Key Questions For Authors:**

1.	In practice, attackers may not know the target layers, and hyperparameters. How do you justify it?

**Limitations:**

1. The assumption is too ideal.

**Strengths And Weaknesses:**

Strengths
1. Practical, effective two-stage attack: author demonstrate the privacy risk of memory editing and propose a simple, practical attack.
 2. The defense is strong and the experiment is solid.

Weaknesses
1. The attack presumes that the candidate pools contain the true subjects and prompts; this is a strong assumption in open-world settings.
2. White-box attacker assumption and reliance on ΔW and knowledge of the editor/hyperparameters (including C) may not always hold.

---

> ### Author Rebuttal · Authors · 2026-03-31
>
> We appreciate your constructive feedback and the recognition of our work. Detailed responses to your questions are provided below.
>
> ## W1: Concern regarding the strong assumption of candidate pools in open-world settings.
>
> Thank you for your comment. Respectfully, we argue that the reliance on candidate pools is a reasonable assumption in most scenarios.  Attackers usually utilize prior knowledge (such as public entity lists or individual registries) to construct candidate pools for attacks. To enhance its feasibility, we supplement the experiments on how candidate pool size impacts attack performance in Tables 1 and 2. The results show that our attack remains robust as the search space increases up to $10^7$ subjects and $10^4$ prompt templates.
>
> **Table 1: Results of subject inference attack using MEMIT across different subject candidate pool sizes (N) in the Counterfact dataset (num_edit=100).**
>
> | Model| N=2,000 | N=10,000 | N=100,000 | N=1,000,000 | N=10,000,000 |
> | :--- | :---: | :---: | :---: | :---: | :---: |
> | `Llama3-8B-Instruct` |  0.99   |   0.99   |   0.99    |    0.97     |     0.96     |
> | `Qwen2.5-7B-Instruct` |  0.94   |   0.94   |   0.94    |    0.93     |     0.93     |
> | `GPT-J-6B` |  0.96   |   0.96   |   0.96    |    0.95     |     0.95     |
>
> **Table 2: Results of prompt recovery attack (Top-20) using MEMIT across different prompt candidate pool sizes (N) in the Counterfact dataset (num_edit=100).**
>
> | Model | N=1,000 | N=3,000 | N=5,000 | N=7,000 | N=10,000 |
> | :--- |:-------:|:-------:|:-------:|:-------:|:--------:|
> | `Llama3-8B-Instruct` |  0.94   |  0.94   |  0.93   |  0.93   |   0.92   |
> | `Qwen2.5-7B-Instruct` |  0.97   |  0.96   |  0.96   |  0.97   |   0.95   |
> | `GPT-J-6B` |  0.99   |  0.99   |  0.97   |  0.98   |   0.98   |
>
> For more detailed results, please refer to Tables 1 and 2 provided in https://anonymous.4open.science/r/EditAttackFile-CD44/additional_experiments.md.
>
> ## W2&Q1: Concern regarding the white-box assumption and the knowledge of hyperparameters.
>
> Thank you for raising this interesting question. We want to respectfully point out that, to the best of our knowledge, our work is the first study of reverse-engineering edited data, and we establish a unified attack framework that reveals the inherent vulnerability of the located-then-edit paradigm. This work provides us with critical insight into the security risks of such techniques [1,2].
>
> Regarding hyperparameters, our attack remains effective without prior knowledge of these settings. Specifically, attackers can directly determine the target FFN layers via causal tracing [3]. Regarding the reliance of $C$, as shown in Figure 5 of Appendix E.5 (see Figure 1 in https://anonymous.4open.science/r/EditAttackFile-CD44/additional_experiments.md), we estimated $C$ across diverse datasets (Wikitext, the Pile, and Wikipedia) and varying sample sizes (from 10 to 10,000). The results show the strong robustness of our attack to the estimation of $C$, where the ground-truth $C$ is estimated from 100,000 samples in Wikipedia.
>
> ___
>
> Once again, we deeply appreciate your thoughtful and encouraging feedback. We look forward to your further discussions.
>
> Best,
>
> Authors of Paper 20446
>
> _[1] Investigating Model Editing for Unlearning in Large Language Models. 2025_
>
> _[2] Editing as unlearning: Are knowledge editing methods strong baselines for large language model unlearning? 2026_
>
> _[3] Locating and Editing Factual Associations in GPT. 2022_

---

> > ### Author Rebuttal · Reviewer_uAeZ · 2026-04-03
> >
> > The review addresses most of my concerns. I believe this work clears the bar for acceptance, especially if the results presented during the rebuttal are incorporated into the paper.

---

> > > ### Author Response · Authors · 2026-04-04
> > >
> > > Dear Reviewer uAeZ,
> > >
> > > We greatly appreciate your positive feedback and constructive suggestions, which have been instrumental in improving the quality of our work.
> > >
> > > As suggested, we will incorporate all experimental results presented during this rebuttal into the revision.
> > >
> > > If you have any additional questions or concerns that we can clarify or address, we would be happy to provide further information to ensure all aspects of our work are clear.
> > >
> > > Thank you once again for your valuable time and effort in reviewing our submission!
> > >
> > > Best regards,
> > >
> > > Authors

---

### Official Review · Reviewer_9DhC · 2026-03-09

**Soundness:** 3
**Presentation:** 3
**Significance:** 3
**Originality:** 3
**Overall Recommendation:** 5
**Confidence:** 3

**Summary:**

This paper investigates the privacy implications of "locate-then-edit" algorithms, specifically ROME, MEMIT, and AlphaEdit. The authors identify a side-channel vulnerability arising from the algebraic structure of the parameter updates ($\Delta W$): the low-rank nature of these updates preserves a spectral signature aligned with the key vectors of the edited subjects. To exploit this, the authors propose a two-stage reverse-engineering framework, KSTER. The first stage applies singular value decomposition (SVD) to the update matrix to infer the edited subject from a candidate pool, while the second stage utilizes an entropy-based metric to recover the associated prompt template. Finally, the paper introduces "Subspace Camouflage," a defense mechanism that injects semantic decoys into the update calculation to obfuscate the spectral fingerprint. Experiments are conducted on GPT-J, Llama-3, and Qwen-2.5.

**Compliance With Llm Reviewing Policy:**

Affirmed.

**Final Justification:**

My changed my score from Borderline Reject to Accept. The authors' rebuttal addressed my primary concerns regarding the soundness and practical significance of the attack. By providing the requested end-to-end extraction results, the authors demonstrated that the attack can realistically recover the sensitive object (e.g., achieving a 35% Top-1 exact match on Llama-3-Instruct), validating that the identified side-channel is a credible threat in an open-weight setting. The new experiments show robust recall against a 10M-candidate pool, addressing my concerns about scalability. While the new experiments also confirmed my hypothesis that the proposed "Subspace Camouflage" defense degrades local utility for the decoy subjects (a limitation that must be explicitly caveated in the final version) this does not negate the paper's core contribution. Ultimately, the originality of identifying this fingerprint in locate-then-edit updates, combined with the clarity of the presentation, makes this a valuable contribution that outweighs shortcomings of the proposed defense.

**Key Questions For Authors:**

1. How does the attack perform at _end-to-end extraction_, including recovery of the sensitive object $o$?

2. How does the candidate pool assumption scale to realistic domains, like deletion of sensitive information about private individuals, where the number of unique names/subjects could be of order a hundred million?

3. Regarding the "Subspace Camouflage" defense, can you evaluate whether injecting semantic decoys introduces hallucinations or factual errors specifically regarding the _decoy_ subjects themselves? This is where one might expect the most severe impact on utility.

**Limitations:**

The paper covers some limitations (e.g., reliance on predefined candidate pools, single-time editing), but the following limitations are under-emphasized or omitted:
- There is no evaluation of end-to-end attack effectiveness including the subject + prompt template + sensitive object.
- While the need for a candidate pool is mentioned, the authors don't discuss how this limits applicability in practice.

**Strengths And Weaknesses:**

# Strengths

**Originality.**

The paper introduces a novel perspective on model editing attacks by analyzing the algebraic structure of the parameter update matrix ($\Delta W$). The insight that the low-rank nature of ROME/MEMIT updates leaves a recoverable spectral signature is a creative application of linear algebra to model auditing. This distinguishes the work from prior studies (e.g., Patil et al., 2024) that primarily relied on input-output behaviors or logit distributions.

**Soundness.**
The theoretical derivation linking the row space of the update matrix to the subject key vectors ($\mathbf{k}_*$) appears technically sound. The analysis in Section 4.2 provides a solid justification for why the spectral attack is effective in the idealized experimental setting.

**Significance.**
The paper effectively highlights the "Streisand Effect" in model editing – specifically, that the differential update highlights exactly what the developer intended to hide. This is a valid and important caution for practitioners considering these tools for data deletion.

# Weaknesses

**Soundness**
- The paper explicitly omits the final step of the attack – recovering the sensitive object $o$ – claiming in Section 4.3 that it is "easy" once the subject and prompt are known. I’m doubtful of this claim since: (1) Modern instruction-tuned models (e.g., Llama-3-Instruct, used in experiments) are aligned to refuse PII extraction. Even if the attacker recovers the prompt "Alice's SSN is...", the pre-edit model may refuse to answer. (2) The attack recovers a semantically similar prompt, not necessarily the exact prompt. It is not clear whether a semantically similar prompt can trigger extraction of the memorized object.

- It is assumed that the attacker has access to a _candidate pool_ containing the ground-truth subjects and prompt templates. While this may be feasible for subjects that are common named entities (e.g., "Paris"), it seems impractical for subjects that are the names of private individuals (arguably the motivating use case). Moreover, the experiments only seem to cover small candidate pools (< 2000 candidates). It is unclear if the spectral signature is distinct enough to distinguish between highly similar candidates in a large search space.

- The evaluation checks if the defense breaks unrelated knowledge, but it doesn't check if the decoys themselves cause issues. For example, if I use "London" as a decoy to hide an edit about "Paris," does the model start getting confused about London?

**Significance**

- The paper frames ROME/MEMIT as privacy/unlearning tools. However, these methods were designed for factual association updates and error correction (Meng et al., 2022; 2023), not scrubbing sensitive PII. Characterizing the mathematical transparency of the update mechanism as a vulnerability misaligns with the original design of ROME/MEMIT.

- The proposed Subspace Camouflage defense offers no formal guarantees. It relies on obscuring the signal of the ground truth subject, which might be vulnerable to adaptive attacks (e.g., that filter outliers).

**Presentation.**

Section 3.1.1 may be challenging for readers who are not familiar with ROME/MEMIT. Concepts like $K_p$ (preserved knowledge) are initially introduced without clear definition. The paper discusses editing FFNs, but the notation sometimes conflates this with general parameter updates.

**Originality.**

The "Subject Invariance" observation (Section 4.1) is presented as a key empirical basis, yet this phenomenon is established in prior work. The original ROME paper (Meng et al., 2022) identified the subject's last token as a causal locus for factual recall.

# Minor points

- In impact statement: the claim that the attack functions as a generic "security ruler" is an overstatement, as the attack is tailored specifically to the algebraic structure of locate-then-edit algorithms and may not generalize.

- It is assumed that the attacker can estimate the covariance matrix used during the edit (often derived from Wikipedia). If the defender uses a private dataset to calculate $C$, the attacker's estimate will be off. The paper doesn't test how sensitive the attack is to a misspecified $C$.

---

> ### Author Rebuttal · Authors · 2026-03-31
>
> Thanks for your comments. We reply to the raised issues below, and full experimental results are available at https://anonymous.4open.science/r/EditAttackFile-CD44/additional_experiments.md.
>
> ## W1&Q1: Concern regarding the E2E extraction experiment.
> Thank you for raising this interesting question.
>
> First, the core contribution of this work is to design a unified framework for reverse-engineering edited subjects and prompt templates. Extracting $o$ using the given prompt falls under jailbreak/extraction attacks, which is beyond the scope of our research.
>
> Second, our focus remains on the security of editing methods, which should not be conflated with downstream alignment strategies. In practice, $o$ may not be PII that triggers safety alignment, but rather specific domain facts [1,2]; therefore, we do not assume additional defense. Below, we supplement an end-to-end extraction experiment.
>
> | Model | Top-1 | Top-5 | Top-20 |
> | :--- |:-----:|:-----:|:------:|
> | `Llama3-8B-Instruct` | 0.35  | 0.64  |  0.78  |
>
> ## W2&Q2: Concern regarding the attack‘s robustness across candidate size.
>
> Thank you for your comment.
>
> First, we maintain that privacy should encompass all public entities, as their leakage can pose potential security risks.
>
> Second, we argue that the reliance on candidate pools is a reasonable assumption, as attackers can usually utilize prior knowledge to construct candidate pools.
>
> To enhance its feasibility, we supplement experiments of how candidate pool size impacts attack performance below; the results show that our attack remains robust as the search space increases up to $10^7$ subjects.
>
> | Model  | N=2,000 | N=10,000 | N=100,000 | N=1,000,000 | N=10,000,000 |
> | :--- | :---: | :---: | :---: | :---: | :---: |
> | `Llama3-8B-Instruct` |  0.99   |   0.99   |   0.99    |    0.97     |     0.96     |
>
>
> ## W3&Q3: Concern regarding the factual errors of decoy subjects in our defense.
>
> Thank you for this insightful observation. We evaluate the target recall for decoy subjects (using `rewrite_prompt` and `paraphrase_prompt`) on Llama3-8B-Instruct edited via MEMIT. Results show that recall remains stable for coefficients up to 1 before declining. Our defense serves as a moderate method with an acceptable utility tradeoff, with detailed limitations discussed in the revision.
>
> | Scale | True Fact Recall |
> | :---: | :---: |
> | **0.0** | 0.51 |
> | **1.0** | 0.51 |
> | **3.0** | 0.43 |
>
> ## W4: Concern regarding the difference between the design intent and the use of editing methods.
>
> Although ROME/MEMIT were designed for factual correction, recent work [1,2] has explored how to leverage model editing for privacy protection.
> Therefore, when these methods are used to erase sensitive or harmful data, the "editing intent" and "original outputs" constitute private data. Our attack serves as a warning, reminding the community to apply model editing to these domains prudently.
>
> ## W5: Concern regarding the generalizability of our defense.
>
> Thank you for raising this concern. We will provide a theoretical analysis of our defense against adaptive attacks in the revision. Specifically, we have proven that: without additional knowledge of the residual matrix $R$ (rather than merely its row space via SVD), an attacker can neither (i) detect whether the model has deployed this defense, nor (ii) recover any meaningful information about the key matrix $K$. Therefore, attempting to filter outliers via statistical methods is ineffective.
>
> ## W6: Concern regarding the notations, definitions, and background.
> Thank you for your detailed feedback. We will state that $W$ denotes the FFN weight matrix at the beginning of Section 3.1 in the revision. Additionally, we will supplement a brief background on the ROME/MEMIT mechanisms and a definition for the preserved knowledge $K_p$.
>
> ## W7: Concern regarding the novelty of the "Subject Invariance."
>
> Thank you for your comment. Respectfully, we argue that our discovery is distinct from ROME. During the causal tracing phase, ROME only corrupted the subject's embedding, without modifying the remaining context. In contrast, "Subject Invariance" reveals that even when the surrounding context changes, the activation direction at the subject's last token remains highly consistent.
>
> ## MP2: Concern regarding the sensitivity of the attack to $C$.
>
> Thank you for your insight. As shown in Fig. 5 of Appendix E.5, we estimated $C$ across diverse datasets (Wikitext, the Pile, and Wikipedia) and varying sample sizes (from 10 to 10,000). The results show the strong robustness of our attack to the estimation of $C$.
>
> ---
>
> We are eager to know whether we have addressed your questions, and we would be happy to discuss further at any time to assist with your re-evaluation.
>
> Best,
>
> Authors of Paper 20446
>
> _[1] Investigating Model Editing for Unlearning in Large Language Models. 2025_
>
> _[2] Editing as unlearning: Are knowledge editing methods strong baselines for large language model unlearning? 2026_

---

> > ### Author Rebuttal · Reviewer_9DhC · 2026-04-03
> >
> > Thank you for the detailed rebuttal and additional experimental results. The new results address my primary concerns regarding the soundness of the attack, though they also confirm my hypotheses regarding the limitations of the proposed defense.
> >
> > **W1&Q1.**
> > I appreciate the inclusion of the E2E extraction results, which strengthen the paper's claims. While the rebuttal argues that object extraction is out of scope, the paper's abstract explicitly states the side-channel enables attackers to "recover the edited data," which logically includes the sensitive object. Achieving a 35% Top-1 exact match on Llama-3-Instruct validates this as a practical threat. I request that these E2E results be included in the main text to support the privacy implications of the attack.
> >
> > **W2&Q2.**
> > The scaling experiments show a strong 0.96 Top-100 recall at 10M candidates, validating the precision of the spectral signature. For the final version, please explain how this 10M candidate set was generated. The difficulty of this retrieval task depends on the semantic density of the candidate space (e.g., distinguishing a target from random tokens versus a dense set of similar entities). Providing a measure of candidate similarity would help contextualize this result.
> >
> > **W3&Q3.**
> > Table 5 confirms my initial concern regarding local utility degradation. At a camouflage scale of $\alpha = 5$ (which the text identifies as necessary for strong protection), the true fact recall for the decoy subjects decreases significantly (e.g., dropping from 0.52 to 0.22 for AlphaEdit). This indicates the defense damages the factual integrity of the decoy subjects used to mask the edit. This localized impact on utility should be clearly discussed as a limitation of the Subspace Camouflage defense.
> >
> > I am increasing my score to Accept, assuming the final version incorporates the E2E extraction results and clearly discusses the local utility tradeoffs of the defense.

---

> > > ### Author Response · Authors · 2026-04-04
> > >
> > > Dear Reviewer 9DhC,
> > >
> > > We greatly appreciate the positive feedback and constructive suggestions, which have been instrumental in improving the quality of this work.
> > >
> > > In the revision, we will:
> > >
> > > 1. Add the E2E extraction experiments to the main text.
> > >
> > > 2. Introduce the setup of the candidate subject pool (sampled from real human names in the IMDB dataset) and provide a similarity analysis.
> > >
> > > 3. Discuss the limitations of the Subspace Camouflage defense.
> > >
> > > If there are any additional questions or concerns that can be clarified or addressed, we would be happy to provide further information to ensure all aspects of the research are clear.
> > >
> > > Thank you once again for your valuable time and effort in reviewing this submission!
> > >
> > > Best regards,
> > >
> > >
> > > Authors

---

### Official Review · Reviewer_7N5U · 2026-03-12

**Soundness:** 3
**Presentation:** 3
**Significance:** 3
**Originality:** 3
**Overall Recommendation:** 5
**Confidence:** 3

**Summary:**

This paper investigates a previously underexplored privacy vulnerability in the locate-then-edit paradigm of model editing for large language models. The authors demonstrate that the parameter updates produced by editing methods such as ROME, MEMIT, and AlphaEdit inadvertently serve as a side channel, enabling an attacker to recover both the edited subjects and their associated prompts. The proposed attack framework, KSTER, operates in two stages: first, it applies spectral analysis to the weight update matrix to identify which subjects were edited; second, it uses an entropy-reduction-based scoring metric over candidate prompts to recover the semantic context of the edit. The paper provides theoretical analysis (via the Woodbury matrix identity) showing that the row space of the update matrix encodes a fingerprint of edited subjects. Additionally, the authors propose a defense called subspace camouflage, which injects semantic decoys into the key matrix to obfuscate the spectral fingerprint while preserving editing utility. Experiments are conducted on three LLMs (GPT-J, Llama3-8B-Instruct, Qwen2.5-7B-Instruct), three editing methods, and two benchmark datasets (CounterFact, zsRE), with both white-box and gray-box attack settings evaluated.

**Compliance With Llm Reviewing Policy:**

Affirmed.

**Final Justification:**

I maintain my original recommendation of acceptance

**Key Questions For Authors:**

- How does the attack perform when the candidate pool becomes significantly larger while the number of edits $N$ remains fixed? If the candidate space grows substantially, does the recovery accuracy degrade significantly? This would help clarify the practical threat level of the attack.

- Can you provide theoretical or empirical insights for the subject invariance property (Section 4.1)? Understanding why this phenomenon occurs, and under what conditions it might break down, would strengthen the analysis.

- In Section 3.1, the notation $D^t$ is introduced, but the superscript $t$ is not clearly defined. Could the authors clarify what $t$ represents (e.g., a target edit, time step, or another concept)?

**Limitations:**

Yes

**Strengths And Weaknesses:**

**Strengths**

The paper addresses an important and timely question: whether model editing methods, which are increasingly proposed as tools for privacy protection and knowledge correction, themselves introduce privacy risks. This is a meaningful contribution to the responsible deployment of LLMs. The finding that the parameter difference $\Delta W$ serves as a side channel is novel and has practical implications for model editing practitioners.

The paper is generally well-written and well-structured with rigorous theoretical analysis.

**Weaknesses**

- The proposed attack relies on a predefined candidate pool of subjects and prompts with structured format (i.e., knowledge represented as triples $(s,r,o)$). This assumption may limit the practicality of the attack in real-world scenarios involving free-form natural language edits, where candidate prompts and subjects are not constrained to such structured formats.

- There are some areas where clarity could be improved. The threat model (Section 3.2) introduces both white-box and gray-box settings but the paper primarily focuses on white-box; the gray-box results are somewhat buried and underdeveloped. A clearer upfront statement about this emphasis would help set reader expectations.

- The assumption line 146-152 that the attacker has access to $C$ and a joint candidate pool S_cand × R_cand containing the ground-truth edits deserves more prominent discussion, as it substantially constrains the threat model.

- The subject invariance observation (Figure 2) is presented empirically, but the paper would benefit from a clearer explanation of why this invariance emerges after model editing. For example, it would be helpful to understand whether this behavior is specific to the FFN layers targeted by current editing methods, or whether it reflects a more general property of transformer representations.

---

> ### Author Rebuttal · Authors · 2026-03-31
>
> Thank you for your constructive comments and the recognition of our work! We reply to the raised questions below.
>
> ## W1: Concern regarding the reliance on structured candidate pools (triples).
>
> Thank you for raising this insight. Our reliance on triples is a setting made for evaluation purposes.
>
> First, to evaluate the effectiveness of our attack in locate-then-edit methods, we must align with their standard benchmarks (e.g., CounterFact, zsRE) and expected input formats.
>
> Second, in practice, locate-then-edit methods can be extended to free-form text through a "triple extractor." For example, in the recent evaluation benchmark __AKEW__ [1], standard practice is to use an LLM to extract triples from Wikipedia, and then perform editing. Therefore, by integrating an extraction module, our attack method can be extended to free-form text, and leave it as future work.
>
> ## W2: Concern regarding the white-box setting.
>
> Thank you for your concern. Our primary focus is indeed on the white-box setting. We want to respectfully point out that, to the best of our knowledge, __our work is the first study of reverse-engineering edited data__, and we establish a unified attack framework that reveals the inherent vulnerability of the located-then-edit paradigm, which holds greater theoretical significance than the gray-box setting. We will emphasize this at the beginning of Section 3.2 in the revision.
>
> ## W3&Q1: Concern regarding more discussion about $C$ and the candidate pools.
>
> Thank you for your comment. Respectfully, we argue that the reliance on candidate pools is a reasonable assumption, as attackers can usually construct a feasible candidate space using public trends or domain knowledge. To enhance its feasibility, we supplement the experiments of how candidate pool size impacts attack performance in Tables 1 and 2; the results show that our attack remains robust as the search space increases up to $10^7$ subjects and $10^4$ prompt templates.
>
> Regarding access to $C$, as shown in Fig. 5 of Appendix E.5, we estimated $C$ across diverse datasets (Wikitext, the Pile, and Wikipedia) and varying sample sizes (from 10 to 10,000). The results show the strong robustness of our attack to the estimation of $C$ (where the ground-truth $C$ is estimated from 100,000 samples in Wikipedia).
>
> **Table 1: Results of subject inference attack using MEMIT across different subject candidate pool sizes (N) in the Counterfact dataset (num_edit=100).**
>
>
> | Model  | N=2,000 | N=10,000 | N=100,000 | N=1,000,000 | N=10,000,000 |
> | :--- | :--------: | :--------: | :--------: | :--------: | :--------: |
> | `Llama3-8B-Instruct` |  0.99   |   0.99   |   0.99    |    0.97     |     0.96     |
> | `Qwen2.5-7B-Instruct` |  0.94   |   0.94   |   0.94    |    0.93     |     0.93     |
> | `GPT-J-6B` |  0.96   |   0.96   |   0.96    |    0.95     |     0.95     |
>
>
> **Table 2: Results of prompt recovery attack (Top-20) using MEMIT across different prompt candidate pool sizes (N) in the Counterfact dataset (num_edit=100).**
>
>
> | Model  | N=1,000 | N=3,000 | N=5,000 | N=7,000 | N=10,000 |
> | :--- | :--------: | :--------: | :--------: | :--------: | :--------: |
> | `Llama3-8B-Instruct` |  0.94   |  0.94   |  0.93   |  0.93   |   0.92   |
> | `Qwen2.5-7B-Instruct` |  0.97   |  0.96   |  0.96   |  0.97   |   0.95   |
> | `GPT-J-6B` |  0.99   |  0.99   |  0.97   |  0.98   |   0.98   |
>
> For more detailed results, please refer to Table 1, Table 2, and Figure 1 provided in https://anonymous.4open.science/r/EditAttackFile-CD44/additional_experiments.md.
>
> ## W4&Q2 Concern regarding the subject invariance observation.
>
> Thank you for raising this interesting question. We supplement activation alignment experiments on Llama3-8B-Instruct across different down_proj layers (Figure 2, https://anonymous.4open.science/r/EditAttackFile-CD44/additional_experiments.md). The results show alignment rate decreases across layers 0–11, with irregular fluctuations in layers 12–31.
>
> The main focus of this work is to design and evaluate attacks based on this observation, and a deep dissection of its underlying mechanism lies beyond the scope of this research. We plan to add relevant discussions in the revision and leave it as future work. Thank you again for your inspiring suggestion.
>
> ## Q3: Concern regarding the notation $D^t$.
>
> Thank you for your careful review. In Section 3.1, the superscript $t$ in $D^t$ refers to the knowledge **t**riple $(s, r, o)$ that is to be edited. We will add the corresponding definition in the revision.
>
> ---
>
> Once again, we deeply appreciate your thoughtful and encouraging feedback. We look forward to your further discussions!
>
> Best,
>
> Authors of Paper 20446
>
> _[1] AKEW: Assessing Knowledge Editing in the Wild. 2024_

---

> > ### Author Rebuttal · Reviewer_7N5U · 2026-04-03
> >
> > Thank you for your response. I have no further questions.

---

> > > ### Author Response · Authors · 2026-04-04
> > >
> > > Dear Reviewer 7N5U,
> > >
> > > We greatly appreciate your positive feedback and constructive suggestions, which have been instrumental in improving the quality of our work.
> > >
> > > If you have any additional questions or concerns that we can clarify or address, we would be happy to provide further information to ensure all aspects of our work are clear.
> > >
> > > Thank you once again for your valuable time and effort in reviewing our submission!
> > >
> > > Best regards,
> > >
> > > Authors

---

### Decision · Program_Chairs · 2026-04-30

**Decision:**

Accept (regular)

**Comment:**

The paper discovers a side-channel to the locate-then-edit methods where one can, from the weight update recorder subject(s) that were edited, and often also the associated prompt semantics.

The reviewers appreciate the paper's novel attack surface, the clear algebraic insights and theory and the broad empirical evaluation. The rebuttal further strengthened the original submission by extending the baselines, including larger candidate pools, sequential editing, and further analyses.

Points that remain open are the reliance on the white-box setup, the utility trade-off introduced by the defense (which should be highlighted more in a discussion in the paper), and the framing around privacy and memorization which seems unsuited for the provided contribution (which is finding a side-channel against concept editing methods).

The AC fully agrees with the issue with the framing of the paper (similar to the one expressed by: Reviewer 9DhC), which seems not in line fully with the contribution: The paper discusses "memorization" of sensitive information and privacy risks, and then discusses the locate-then-edit methods that are removing concepts or facts from the model. The AC would argue that the framing of sensitive and private data here is not too adequate, it seems that the paper analyzes facts rather than personal information. This becomes especially obvious as the methods that are evaluated were primarily introduced for: fact editing, knowledge correction, updating specific model outputs, and as all evaluation is conducted on CounterFact (Meng et al., 2022), and zsRE (Levy et al., 2017), non-private, non-sensitive datasets, where it is not known that the data is actually "memorized". Instead having learned the facts would rather be a generalization, i.e., the model's ability to learn from the training data. Memorization, from a privacy perspective, is, in the AC's take rather to be connected with memorization of individual training data points that can then be verbatim extracted, e.g. as shown by Carlini et al. [A]. Therefore, shall the paper still be nominated for acceptance, the paper should change the framing and focus on the attack and the facts, but remove the "memorization" and "privacy" spin of the paper.

While the authors argue that two papers (which they cite as [1,2] in their answer to Reviewer 9DhC) have used the concept-editing methods for privacy, at least to the AC's research, these papers were not accepted at A* conferences (or in case of [2] accepted at all), and using them as motivation (without making this explicit in the original submission) seems very weak.

While the overall attack results are also valuable without the memorization/privacy framing, adding this framing makes the paper actually weaker and not well positioned for what it is.

In the current form, the paper blurs factual/concept editing with privacy-preserving deletion/unlearning, and it uses “memorization” in a way that is often not the right concept for what is actually being attacked.


[A] Carlini, Nicholas, Florian Tramer, Eric Wallace, Matthew Jagielski, Ariel Herbert-Voss, Katherine Lee, Adam Roberts et al. "Extracting training data from large language models." In 30th USENIX security symposium (USENIX Security 21), pp. 2633-2650. 2021.